# THE ROLE OF REPRESENTATION TRANSFER IN MULTITASK IMITATION LEARNING

## ABSTRACT

Transferring representation for multitask imitation learning has the potential to provide improved sample efficiency on learning new tasks, when compared to learning from scratch. In this work, we provide a statistical guarantee indicating that we can indeed achieve improved sample efficiency on the target task when a representation is trained using sufficiently diverse source tasks. Our theoretical results can be readily extended to account for commonly used neural network architectures such as multilayer perceptrons and convolutional networks with realistic assumptions. Inspired by the theory, we propose a practical metric that estimates the notion of task diversity. We conduct empirical analyses that align with our theoretical findings on five simulated environments—in particular leveraging more data from source tasks can improve sample efficiency on learning in the new task. Our experiments further demonstrate that our proposed task diversity metric is positively correlated to the imitation performance.

## 1 INTRODUCTION

Imitation learning (IL) is a common approach to learn sequential decision making agents—it involves imitating the expert through matching distributions induced by expert transitions (Osa et al., 2018). However, current methods require thousands of expert transitions even in simple tasks (Mandlekar et al., 2022; Jang et al., 2021; Ablett et al., 2023). Acquiring large amounts of data can be expensive and even infeasible in domains including robotic and healthcare applications. To address this challenge, empirical research has proposed transferring part of the agent trained from one or more tasks to a target task with the goal of improving sample efficiency on the target task (Brohan et al., 2023; Hansen et al., 2022; Jang et al., 2021; Li et al., 2022). In this paper we show how much sample efficiency on the target task improves, compared to training an agent from scratch, when we transfer a pretrained representation to the target task via multitask imitation learning (MTIL). We investigate the following questions:

*(i)* Compared to learning from scratch, can we reduce the number of expert transitions on target task by leveraging expert transitions from source tasks?

*(ii)* If so, given a particular set of source tasks, can we characterize its impact on the reduction of required expert transitions on the target task?

We address these two questions through both theoretical and empirical analyses:

**Theoretical Contributions** Although Arora et al. (2020) has investigated the benefits of representation transfer, their result does not relate the target task and the source tasks. We cannot guarantee that transferring the representation to a particular target task can yield any benefit. Inspired by Tripuraneni et al. (2020), we provide an analysis that relates the source and target tasks via the notion of task diversity. This line of work (Arora et al., 2020; Tripuraneni et al., 2020; Maurer et al., 2016) heavily relies on Gaussian complexity. In this paper, we instead use Rademacher complexity and provide a tighter bound by a log factor than using Gaussian complexity. Informally, our main result indicates a sample-complexity bound to capture the benefits of transferring representation in MTIL:

$$\text{Imitation Gap} \leq \mathcal{O}\left(\frac{1}{(1-\gamma)^2}\sqrt{\frac{1}{\sigma}\left(\mathfrak{R}_{NT}(\Phi) + \frac{1}{\sqrt{NT}}\right)} + \frac{1}{\sqrt{M}}\right),$$

where $\gamma$ is the discount factor, $\mathfrak{R}_{NT}(\Phi)$ is the Rademacher complexity of the representation class, $\sigma$ captures the task diversity of the $T$ source tasks, each with $N$ expert transitions, and $M$ is the number of expert transitions from the target task. Our result is due to the objective of behavioural cloning, where the method aims to minimize the Kullback–Leibler (KL) divergence between the expert and the learner (Ghasemipour et al., 2019; Xu et al., 2020). The consequence is that we can connect our result with deep-learning theory, where the commonly used neural networks are quantified directly with Rademacher complexity (Bartlett et al., 2021).

**Empirical Contributions**  While there are existing works on representation learning in MTIL, we focus on understanding how task diversity plays a role in the success of representation transfer (Xu et al., 2018; Arora et al., 2020; Zhang et al., 2023; Guo et al., 2023). First, building upon our theoretical contribution, we propose a new metric that measures task diversity using the KL-divergence of the expert and the trained policies. Our metric is asymmetrical, that is a desired property for transfer learning (Hanneke & Kpotufe, 2019). Furthermore, our metric is practical to obtain and we empirically show that it is positively correlated to whether the source tasks can yield benefits when transferring the learned representation to the target task. Secondly, we conduct experiments to validate our theory and in addition to discrete action space environments, we extend our experiments to the continuous action space environments. Confirming our theoretical findings, we empirically demonstrate that transferring representations from source tasks to target tasks is a valid approach and generally performs better as we increase the number of source tasks and data.

## 2 BACKGROUND

### 2.1 MARKOV DECISION PROCESSES

Sequential decision making problems can be formulated as Markov Decision Processes (MDPs). An infinite-horizon MDP is a tuple $\mathcal{M} = \langle \mathcal{S}, \mathcal{A}, r, P, \rho, \gamma \rangle$, where $\mathcal{S}$ is the finite state space, $\mathcal{A}$ is the finite action space, $r : \mathcal{S} \times \mathcal{A} \to [0, 1]$ is the bounded reward function, $P \in \Delta_{\mathcal{S} \times \mathcal{A}}^{\mathcal{S}}$ is the transition distribution over the states for each state-action pair, $\rho \in \Delta^{\mathcal{S}}$ is the initial state distribution over the states, and $\gamma \in [0, 1)$ is the discount factor. The agent interacts with the environment through a policy $\pi \in \Pi$, which we assume to be stationary and Markovian (i.e. $\Pi = \Delta_{\mathcal{S}}^{\mathcal{A}}$.)

Through the interconnection between the policy $\pi \in \Pi$ and the environment $\pi$ induces a random infinite-length trajectory $S_0, A_0, S_1, A_1, \ldots$, where $S_0 \sim \rho$, $A_h \sim \pi(\cdot|S_h)$, and $S_{h+1} \sim P(\cdot|S_h, A_h)$. The corresponding discounted stationary state(-action) distribution for policy $\pi$, which describes the "frequency" of visiting state $s$ (and action $a$) under $\pi$, can be written as $\nu_\pi(s) = (1 - \gamma) \sum_{h=0}^{\infty} \gamma^h \mathbb{P}(S_h = s; \pi)$ (and $\mu_\pi(s, a) = (1 - \gamma) \sum_{h=0}^{\infty} \gamma^h \mathbb{P}(S_h = s, A_h = a; \pi)$.)

For any $\pi \in \Pi$, the corresponding value function $v^\pi : \mathcal{S} \to \mathbb{R}$ is defined as $v^\pi(s) = \mathbb{E}_{\pi, \rho, P} \left[ \sum_{h=0}^{\infty} \gamma^h r(S_h, A_h) | S_0 = s \right]$. Then, the optimal policy $\pi^*$ is such that $\pi^*(s) = \arg\max_{\pi \in \Pi} v^\pi(s)$, for any $s \in \mathcal{S}$ (randomly breaking ties in the case of two or more maxima.) In general, to make the problem more tractable, the goal is to obtain an $\varepsilon$-optimal policy. That is, for any $s \in \mathcal{S}$, we have that $v^\pi(s) \geq v^{\pi^*}(s) - \varepsilon$.

### 2.2 IMITATION LEARNING

In imitation learning, the learner has no access to the reward function $r$ and the learner is instead given expert transitions from an expert policy $\pi^*$. The goal is to find an $\varepsilon$-optimal policy $\pi \in \Pi$ using the expert transitions, which is given as a set of $N$ state-action pairs $\{(s_n, a_n)\}_{n=1}^N$, where $(s_n, a_n) \overset{\text{i.i.d.}}{\sim} \mu_{\pi^*}$, for $n \in [N]$. In this case, the quantity $\varepsilon$ is also known as the imitation gap (Swamy et al., 2021). Behavioural cloning (Pomerleau, 1988) treats this problem as a supervised learning problem and aims to obtain $\varepsilon$-optimal policy through minimizing the risk. The risk is defined as $\ell(\pi) = \mathbb{E}_{(s,a)\sim\mu_{\pi^*}} [\ell(\pi(s), a)]$, where $\ell(\cdot, \cdot) : \Delta^{\mathcal{A}} \times \mathcal{A} \to \mathbb{R}^+$ is a loss function. This paper considers the log loss $\ell(\pi(s), a) = -\log \pi(a|s)$, which is a surrogate of the 0-1 loss. The log loss is also equivalent to the KL-divergence between the expert and the learner when the expert is deterministic.

## 2.3 MODEL COMPLEXITY

Analyzing the statistical convergence bounds of learning problems often involves the complexity of the learner's function class. In this paper we use the Rademacher complexity to measure the complexity of a function class.

**Definition 1** *(Rademacher Complexity.) For a vector-valued function $\mathcal{F} = \{f : \mathcal{X} \to \mathbb{R}^K\}$, and $N$ data points $X = (x_1, \ldots, x_N)$, where $x_n \in \mathcal{X}$ for $n \in [N]$, the empirical Rademacher complexity of $\mathcal{F}$ is defined as*

$$\hat{\mathfrak{R}}_X(\mathcal{F}) = \mathbb{E}_\varepsilon \left[ \sup_{f \in \mathcal{F}} \frac{1}{N} \sum_{n=1}^{N} \sum_{k=1}^{K} \varepsilon_{n,k} f_k(x_n) \right],$$

*where $\varepsilon_{n,k}$ are sampled i.i.d. from the Rademacher random variable and $f_k(\cdot)$ is the $k$'th component of $f(\cdot)$. Fix a data distribution $\mathcal{D}_\mathcal{X}$ over $\mathcal{X}$. The corresponding Rademacher complexity of $\mathcal{F}$ is defined as $\mathfrak{R}_N(\mathcal{F}) = \mathbb{E}_\mathbf{X} \left[ \hat{\mathfrak{R}}_\mathbf{X}(\mathcal{F}) \right]$, where the expectation is taken over the distribution $\mathcal{D}_\mathcal{X}$. In the case of $K = 1$, we recover the scalar Rademacher complexity.*

Intuitively, the Rademacher complexity of $\mathcal{F}$ measures the expressiveness of $\mathcal{F}$ over all datasets $\mathbf{X}$ through fitting random noise.

# 3 MULTITASK IMITATION LEARNING (MTIL) WITH REPRESENTATION TRANSFER

We consider the transfer-learning setting where we are given expert transitions of $T$ source tasks. We define each task $t$ as an MDP with different transition distributions and reward functions, but with the same state and action spaces. When the context is unclear, we use subscript, e.g. $\pi_t$, to denote the task-specific objects. Our goal is to leverage the expert transitions of $T$ source tasks to learn an $\varepsilon$-optimal policy on a new target task $\tau$ with better sample efficiency on expert transitions of the target task, when compared to learning from scratch. To achieve this, we first learn a shared representation from the source tasks and transfer it to the target task. During the transfer, we fix the representation and only learn the task-specific mapping, similar to a finetuning procedure (Howard & Ruder, 2018; Shachaf et al., 2021; Liu et al., 2022).

Formally, we consider softmax parameterized policies of the form $\pi^{f,\phi}(s) = \mathrm{softmax}((f \circ \phi)(s))$, where $f \in \mathcal{F}$ is the task-specific mapping and $\phi \in \Phi$ is the representation. Our multitask imitation learning (MTIL) approach is a two-phase procedure for transfer learning: (1) learn a representation $\hat{\phi}$ from the source tasks, and (2) learn a task-specific mapping $\hat{f}$ such that $\pi^{\hat{f},\hat{\phi}}$ performs well on the target task. We call the phases respectively the training phase and the transfer phase.

In the training phase, for each task $t$ of the $T$ source tasks, we are given $N$ state-action pairs $\{(s_{t,n}, a_{t,n})\}_{n=1}^{N}$. Let $\boldsymbol{f} = (f_1, \ldots, f_T)$, where $f_t \in \mathcal{F}$ is the task-specific mapping for task $t$, for $t \in [T]$, which we write $\boldsymbol{f} \in \mathcal{F}^{\otimes T}$ for conciseness. Then, we define the **empirical training risk** as

$$\hat{R}_{train}(\boldsymbol{f}, \phi) := \frac{1}{NT} \sum_{t=1}^{T} \sum_{n=1}^{N} \ell(\pi^{f_t, \phi}(s_{t,n}), a_{t,n}), \tag{1}$$

and the corresponding minimizer of equation 1 is $\hat{\phi} = \arg\min_{\phi \in \Phi} \min_{\boldsymbol{f} \in \mathcal{F}^{\otimes T}} \hat{R}_{train}(\boldsymbol{f}, \phi)$.

In the transfer phase, we are given $M$ state-action pairs $\{(s_m, a_m)\}_{m=1}^{M}$ for a target task $\tau$. With the same loss function $\ell$ as equation 1, we define the **empirical test risk** as

$$\hat{R}_{test}(f_\tau, \phi) := \frac{1}{M} \sum_{m=1}^{M} \ell(\pi^{f_\tau, \phi}(s_m), a_m), \tag{2}$$

where $f_\tau \in \mathcal{F}$ is the task-specific mapping for task $\tau$. We obtain a task-specific mapping that minimizes equation 2 based on the representation $\hat{\phi}$ obtained from the training phase. That is, $\hat{f}_\tau = \arg\min_{f \in \mathcal{F}} \hat{R}_{test}(f, \hat{\phi})$.

Note that taking the expectation of the empirical training and test risks yields their corresponding expected (population) risks: $R_{phase}(\cdot) = \mathbb{E}\hat{R}_{phase}(\cdot)$, for $phase \in \{train, test\}$, where the expectation is over the randomness of the state-action pairs. Let $\pi_\tau^*$ be the optimal policy for the target task $\tau$. Then, we aim to analyze the performance of $(\hat{f}_\tau, \hat{\phi})$ based on the excess risk on the target task $\tau$, which we call the **transfer risk**:

$$R_{transfer}(\hat{f}_\tau, \hat{\phi}) = R_{test}(\hat{f}_\tau, \hat{\phi}) - R_{test}(f_\tau^*, \phi^*), \tag{3}$$

The transfer risk quantifies the generalization error of the learner. Since our loss function is a log loss, which corresponds to the KL-divergence between the expert and the learner, we can convert the transfer risk into the imitation gap. Consequently, when the set of source tasks is diverse, we can establish the sample complexity bound of achieving $\varepsilon$-close to the policy. The diversity is measured with a positive constant $\sigma$, where small $\sigma$ corresponds to less diversity while large $\sigma$ corresponds to high diversity.

For our theorem, we assume policy realizability and regularity conditions on bounding the the norms of the representation and the parameters. Policy realizability enforces that there exists a policy $\pi \in \Pi$ such that with probability at least $1 - \zeta \in [0.5, 1]$, $\pi$ takes the expert action given any state. The consequnce is eliminating a quantity in the risk analysis and quantifying the best possible error achieved by any policy $\pi \in \Pi$. The regularity conditions are required to apply McDiarmid's inequality to upper bound the risks by the Rademacher complexity. We refer the readers to Appendix C.1 for the exact details. We now state our main theorem and defer the analysis to appendix A:

**Theorem 1** *(Transfer Imitation Learning Imitation Gap Bound.) Let $\hat{\phi}$ be the ERM of $\hat{R}_{train}$ defined in equation 1 and let $\hat{f}_\tau$ be the ERM of $\hat{R}_{test}$ defined in equation 2 by fixing $\hat{\phi}$. Let $\sigma > 0$. Suppose the source tasks are $\sigma$-diverse. Under assumptions in Appendix C.1, we have that with probability $1 - 2\delta$,*

$$Imitation\ Gap = \|v^{\pi_\tau^*} - v^{\text{softmax}(\hat{f}_\tau \circ \hat{\phi})}\|_\infty \leq \frac{2\sqrt{2}}{(1 - \gamma)^2}\sqrt{\varepsilon_{gen} + 2\zeta}, \tag{4}$$

*where*

$$\varepsilon_{gen} = \mathcal{O}\left(\frac{1}{\sigma}\left(\mathfrak{R}_{NT}(\Phi) + \frac{1}{\sqrt{NT}}\right) + \frac{1}{\sqrt{M}}\right)$$

*is the generalization error of the transfer risk and $\zeta \in (0, 0.5)$ is a constant related to the policy realizability.*

Theorem 1 indicates that we can obtain $\varepsilon$-optimal policy through our MTIL procedure. Notably, for a fixed performance, MTIL can trade-off the number of target data $M$ at the cost of the number of source tasks $T$ and number of training data per task $N$. Furthermore, if the task diversity is high (i.e. large $\sigma$), then we require less source tasks and source data to obtain the $\varepsilon$-optimal policy.

**Remark 1** By slightly modifying our proof, we can obtain the sample complexity bound for single-task imitation learning (i.e. behavioural cloning (BC)). In particular, by removing the training phase and optimizing both $\phi$ and $f$ in the transfer phase, we can quantify the generalization error of the transfer risk to be $\varepsilon_{gen} = \mathcal{O}(\mathfrak{R}_M(\ell \circ \mathcal{F} \circ \Phi) + 1/\sqrt{M})$. Notice that the Rademacher complexity term dominates the generalization error when the policy is highly expressive (e.g. a neural network). Consequently, if the representation class $\Phi$ is expressive, MTIL can improve upon BC by pretraining the representation using existing source data, thereby reducing the amount of target data required for obtaining an $\varepsilon$-optimal policy.

**Remark 2** In practice, our policies are neural networks (Fujimoto et al., 2018; Haarnoja et al., 2018; Yarats et al., 2020)—we can consider the last layer as the task-specific mapping $f$ and the remaining layers as the representation $\phi$. These neural networks include multilayer perceptrons and convolutional neural networks with Lipschitz activation functions (e.g. ReLU, tanh, sigmoid, max-pooling, etc.). Their Rademacher complexities can be upper bounded based on the amount of training data, number of layers, number of hidden units, and their parameter norms (Neyshabur et al., 2015; Sokolic et al., 2016; Golowich et al., 2018; Truong, 2022b). Due to lemma 4 of Bartlett & Mendelson (2002), our result provides a tighter bound than the Gaussian complexity used in existing works by $\mathcal{O}(\ln NT)$ (Arora et al., 2020; Tripuraneni et al., 2020; Maurer et al., 2016).

## 3.1 ESTIMATING TASK DIVERSITY

Based on our theory, we propose to use the approximate KL-divergences between the expert and the trained source policies to measure the task diversity $\sigma$ (see appendix B for more details):

$$\hat{\sigma} = \frac{\sum_{t=1}^{T} \mathbb{E}_{s_t} \left[ D_{KL}(\pi_t^*(s_t) \| \mathrm{softmax}(\hat{f}_t \circ \hat{\phi})(s_t)) \right]}{T \min_{t \in [T]} \mathbb{E}_{s_\tau} \left[ D_{KL}(\pi_\tau^*(s_\tau) \| \mathrm{softmax}(\hat{f}_t \circ \hat{\phi})(s_\tau)) \right]}. \tag{5}$$

Intuitively, equation 5 measures whether any of the trained source policies already performs well in the target task. Suppose not, then $\hat{\sigma}$ tends to underestimate the task diversity since it is still possible for the learned representation to be beneficial on the target task. Otherwise, it is likely that the task diversity is high since there is already a known source policy that can closely imitate the expert.

We note that equation 5 is asymmetrical (i.e. swapping $\tau$ with one of $t \in [T]$ can yield different diversity estimate.) This is a desirable property since policy transfer generally is not symmetrical (see example 3 of (Hanneke & Kpotufe, 2019)). We further demonstrate the asymmetry in figure 1 and appendix D, where the expert from each environment variation exhibits different robustness—one can stay performant in the target environment while another can degrade in performance.

## 4 EXPERIMENTS

We aim to answer the following questions in our experiments regarding MTIL: *(i)* Can we achieve similar or better imitation performance as behavioural cloning (BC) with less target data? *(ii)* How does the number of source tasks or number of source data impact imitation performance? *(iii)* Does the number of target data have stronger influence on imitation performance compared to the number of source tasks and source data? *(iv)* Can our proposed estimate inform us whether the representation transfer to the target task will be successful?

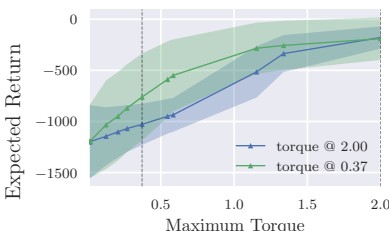

We implement our MTIL procedure with multitask behavioural cloning (MTBC). Algorithmic details of MTBC are provided in appendix E. In short, MTBC is first pretrained with $N$ source data from $T$ source tasks, then finetuned with $M$ target data, whereas BC is trained with only $M$ target data. We analyze the performance of MTBC compared to BC learned from scratch on five simulated environments: frozen lake, pendulum, cartpole, cheetah, and walker. For pendulum, cheetah, and walker, we also provide a discrete action space variant for further analyses. We include continous action spaces to determine if the theoretical findings for discrete action spaces described in section 3

Figure 1: An example demonstrating that policy transfer is asymmetrical. Given two expert policies, $\pi_{0.37}$ and $\pi_{2.0}$, in two pendulum environments with maximum torques 0.37 and 2.0 respectively, we can see that generally $\pi_{0.37}$ is more robust to varying environment variations and is even comparable to $\pi_{2.0}$.

carry over the continuous space. The detailed descriptions of the implementation and the varying environmental parameters are found in appendix F. For the analysis below, we determined that each environment requires $|\mathcal{D}|$ expert transitions to achieve near-expert performance when training BC from scratch (see appendix F.3 for more details).

**Main Results** We first investigate questions *(i)* and *(ii)*. We fix the number of target data $M = |\mathcal{D}|$ while varying the number of source data $N$ and the number of source tasks $T$. Figures 2 and 3 indicate that as we increase $N$ and $T$, MTBC generally improves and can outperform BC with the same number of target data. For continuous and discrete cheetah and walker, which are more complex environments, MTBC outperforms BC while MTBC and BC have comparable performance in simpler environments. Nevertheless, MTBC does not reach expert performance.

In the continuous cartpole and pendulum environments, MTBC achieves higher returns with increasing $N$. In contrast, in frozen lake, cheetah, and walker environments, MTBC achieves higher returns with increasing $T$. We hypothesize that this is because in the former tasks, the agent requires

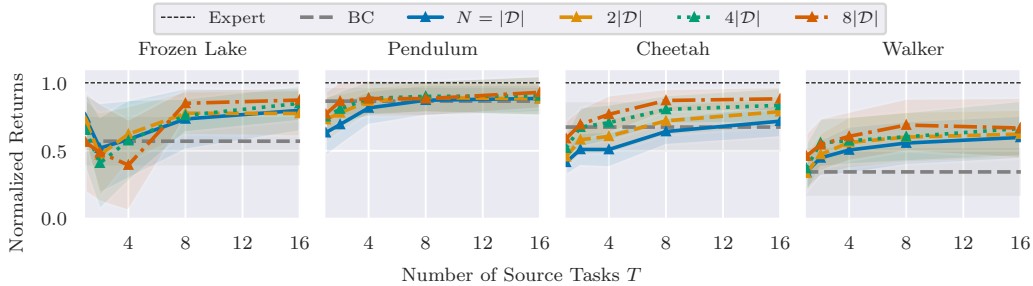

Figure 2: The performance of MTBC generally improves with increasing $N$ and $T$ in discrete tasks. Each solid line colour corresponds to a particular $N$. The solid line corresponds to the mean and the shaded region is 1 standard error from the mean.

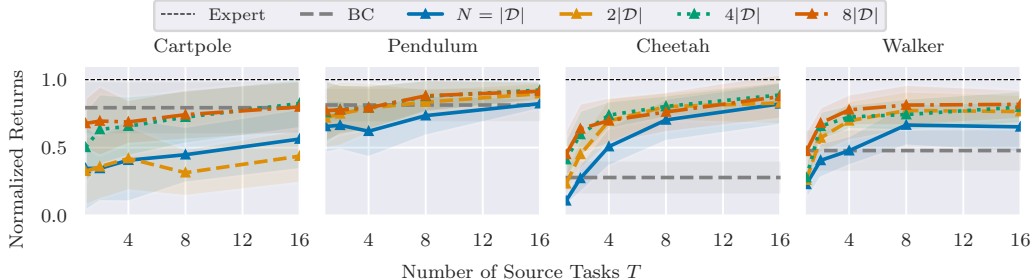

Figure 3: The performance of MTBC generally improves with increasing $N$ and $T$ in continuous tasks. Each solid line colour corresponds to a particular $N$. The solid line corresponds to the mean and the shaded region is 1 standard error from the mean.

full-trajectories to swing the link up and stabilize it, while in the latter tasks, the agent only requires sub-trajectories that are long enough to recover the gait motions. Furthermore, both cheetah and walker tasks have more environmental parameters—it is probable to require more source tasks to learn a representation that is robust in all environment variations. Finally, our results indicate that our theoretical findings on the discrete action space carry over to the continuous action space.

**Varying Amount of Target Data**  Our theoretical analysis suggests that $M$ influences the policy performance independently of $N$ and $T$. We test if varying the amount of test data $M$ has a stronger impact on imitation performance than varying $N$ or $T$, as in question *(iii)*. To do this, we fix the amount of source data $N = 8|\mathcal{D}|$ while varying the amount of target data $M$ by multiples of $|\mathcal{D}|$. For comparison we also include BC trained with $2|\mathcal{D}|$ target data. Figures 4 and 5 show that increasing the number of target data only marginally improves imitation performance. This result indicates that higher returns are mainly achieved by increasing the amount of source data $N$ and number of source tasks $T$. The latter might be due to the learned representation being more expressive than the task-specific mapping. Finally, our results again indicate that our theoretical findings on the discrete action space generally carry over to the continuous action space except for the cartpole task.

**Task-Diversity Metrics for Source and Target Tasks**  We analyze the impact of task diversity on the representation transferrability by investigating question *(iv)*. In practice, since we only have access to the expert transitions, we can approximate the task-diversity estimate defined in equation 5 by computing the empirical KL-divergences (**Approx. KL**):

$$\hat{\sigma} = \frac{M \sum_{t=1}^{T} \sum_{n=1}^{N} \left[ D_{KL}(\pi_t^*(s_{nt}) \| \text{softmax}(\hat{f}_t \circ \hat{\phi})(s_{nt})) \right]}{NT \min_{t \in [T]} \sum_{m=1}^{M} \left[ D_{KL}(\pi_\tau^*(s_{m,\tau}) \| \text{softmax}(\hat{f}_t \circ \hat{\phi})(s_{m,\tau})) \right]}. \tag{6}$$

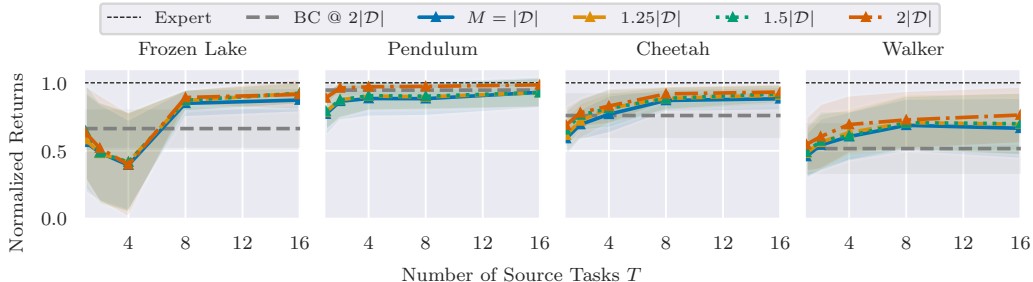

Figure 4: The performance of MTBC generally improves with increasing $M$ and $T$ in discrete tasks. Each solid line colour corresponds to a particular $N$. The solid line corresponds to the mean and the shaded region is 1 standard error from the mean.

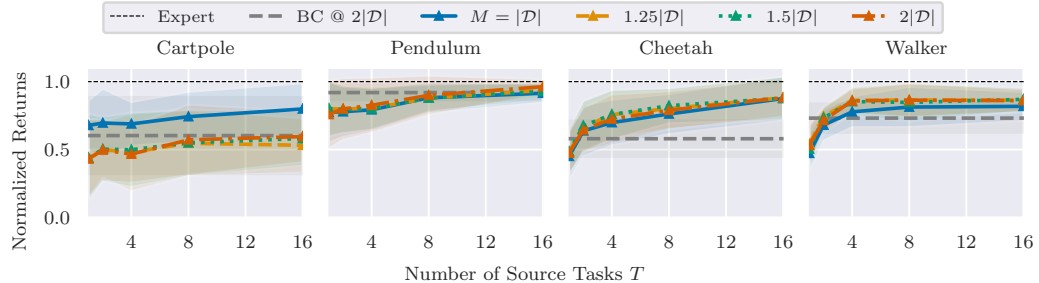

Figure 5: The performance of MTBC generally improves with increasing $M$ and $T$ in continuous tasks. Each solid line colour corresponds to a particular $N$. The solid line corresponds to the mean and the shaded region is 1 standard error from the mean.

In discrete action spaces we can estimate the KL-divergence using the cross-entropy loss, while in continuous action spaces we can estimate it using the mean-squared loss. To evaluate our proposed metric[1], we compare against:

(i) The Euclidean distance of the environmental parameters (**L2**). Let $e \in \mathbb{R}^E$ be an $E$-dimensional vector corresponding to the environmental parameters,

$$\hat{\sigma} = \frac{\sum_{t=1}^{T} \|e_\tau - e_t\|_2}{T \min_{t \in [T]} \|e_\tau - e_t\|_2}.$$

In general, L2 requires ground-truth environmental parameters, which may be unobtainable in practice and further assumes that environmental parameters can be represented as vectors. Euclidean distance might not be suitable in general since the environmental parameter space may lie in a different manifold. Furthermore, L2 is symmetrical which might be undesirable for some environments.

(ii) The difference between average rewards in the target data and the source data (**Data Perf.**). Let $\hat{r}_t$ be the average reward of the expert dataset for task $t$,

$$\hat{\sigma} = \frac{\sum_{t=1}^{T} \hat{r}_\tau - \hat{r}_t}{T \min_{t \in [T]} \hat{r}_\tau - \hat{r}_t}.$$

Data Perf. can capture the variation in task difficulty but it requires access to the rewards of the state-action pairs. This metric also does not reflect whether or not the target expert policy will perform similarly to the source expert policies. Data Perf. may be negative so it is difficult to identify the values to correlate to the task diversity described in the theory.

To analyze the suitability of each of the task diversity metrics described above, we compute the average correlation between the diversity metrics and the normalized returns. Tables 1 and 2 indicate

---

[1]We use the term "metric" loosely as our estimate is not an actual metric.

Table 1: The correlations between various task diversity metrics and the normalized returns in discrete tasks. Bolded text means highest mean.

|  | Frozen Lake | Pendulum | Cheetah | Walker |
|---|---|---|---|---|
| **Pearson** | | | | |
| L2 | $-0.008 \pm 0.023$ | $\mathbf{0.059 \pm 0.069}$ | $-0.008 \pm 0.080$ | $-0.016 \pm 0.096$ |
| Data Perf. | $0.035 \pm 0.223$ | $-0.027 \pm 0.023$ | $-0.057 \pm 0.041$ | $0.279 \pm 0.128$ |
| Approx. KL | $\mathbf{0.206 \pm 0.086}$ | $-0.117 \pm 0.023$ | $\mathbf{0.394 \pm 0.065}$ | $\mathbf{0.397 \pm 0.071}$ |
| **Spearman** | | | | |
| L2 | $-0.019 \pm 0.010$ | $-0.064 \pm 0.032$ | $0.091 \pm 0.024$ | $-0.008 \pm 0.155$ |
| Data Perf. | $0.035 \pm 0.028$ | $0.116 \pm 0.026$ | $0.053 \pm 0.071$ | $-0.159 \pm 0.083$ |
| Approx. KL | $\mathbf{0.163 \pm 0.036}$ | $\mathbf{0.168 \pm 0.031}$ | $\mathbf{0.369 \pm 0.059}$ | $\mathbf{0.228 \pm 0.038}$ |
| **Kendall** | | | | |
| L2 | $\mathbf{0.111 \pm 0.022}$ | $0.052 \pm 0.105$ | $-0.082 \pm 0.059$ | $0.010 \pm 0.044$ |
| Data Perf. | $-0.033 \pm 0.006$ | $\mathbf{0.137 \pm 0.068}$ | $0.046 \pm 0.023$ | $-0.355 \pm 0.053$ |
| Approx. KL | $0.038 \pm 0.037$ | $0.050 \pm 0.020$ | $\mathbf{0.474 \pm 0.070}$ | $\mathbf{0.222 \pm 0.040}$ |

Table 2: The correlations between various task diversity metrics and the normalized returns in continuous tasks. Bolded text means highest mean.

|  | Cartpole | Pendulum | Cheetah | Walker |
|---|---|---|---|---|
| **Pearson** | | | | |
| L2 | $-0.084 \pm 0.028$ | $-0.490 \pm 0.178$ | $0.055 \pm 0.029$ | $-0.131 \pm 0.090$ |
| Data Perf. | $\mathbf{0.227 \pm 0.105}$ | $\mathbf{0.672 \pm 0.119}$ | $0.107 \pm 0.021$ | $0.087 \pm 0.041$ |
| Approx. KL | $-0.038 \pm 0.033$ | $0.106 \pm 0.015$ | $\mathbf{0.589 \pm 0.083}$ | $\mathbf{0.381 \pm 0.069}$ |
| **Spearman** | | | | |
| L2 | $0.199 \pm 0.062$ | $\mathbf{0.449 \pm 0.134}$ | $0.076 \pm 0.026$ | $0.075 \pm 0.080$ |
| Data Perf. | $\mathbf{0.261 \pm 0.047}$ | $-0.434 \pm 0.119$ | $-0.078 \pm 0.030$ | $-0.007 \pm 0.005$ |
| Approx. KL | $0.095 \pm 0.036$ | $0.061 \pm 0.020$ | $\mathbf{0.433 \pm 0.068}$ | $\mathbf{0.405 \pm 0.060}$ |
| **Kendall** | | | | |
| L2 | $0.174 \pm 0.098$ | $\mathbf{0.331 \pm 0.198}$ | $0.062 \pm 0.025$ | $-0.109 \pm 0.052$ |
| Data Perf. | $\mathbf{0.331 \pm 0.064}$ | $-0.364 \pm 0.095$ | $0.153 \pm 0.066$ | $-0.186 \pm 0.046$ |
| Approx. KL | $0.008 \pm 0.025$ | $0.277 \pm 0.051$ | $\mathbf{0.259 \pm 0.040}$ | $\mathbf{0.362 \pm 0.058}$ |

that under three different correlation measures, Approx. KL is often more positively correlated to the normalized returns. In general, this suggests that we can consider Approx. KL to estimate the task diversity in the source tasks with respect to the target task. Our proposed metric also requires the least practical assumptions, only the state-action pairs, as opposed to the extra reward functions or environmental parameters as vectors.

To further analyze our results, we can see that for Spearman and Kendall correlations which can capture non-linear correlations, Approx. KL is positively correlated to the transfer performance on all environments. However, Approx. KL is negatively correlated to the transfer performance in cartpole and discrete pendulum environments under Pearson correlation. We also see that in some tasks Approx. KL is weakly correlated to the returns. Recall that the Approx. KL is an inverse function of the best-performing imitating source policies in the target task. Although this best-performing policy may perform poorly on the target task, it does not necessarily mean that the representation cannot be transferred successfully. For example, in discrete action space we can permute the ordering of the actions as different environmental variations. In this case, with a linear task-specific mapping, we can perfectly learn the policy for each variation by permuting its parameters to match the ordering of the actions.

## 5 RELATED WORK

**Multitask Imitation Learning (MTIL)** MTIL methods aim to find policies that can solve various tasks without incurring excessive cost on number of expert transitions. One class of MTIL methods assumes there exists a shared representation among tasks, learns it, and utilizes it to transfer between

related tasks (Arora et al., 2020; Singh et al., 2020; Brohan et al., 2023; Guo et al., 2023; Zhang et al., 2023). Another class of methods relaxes the shared representation assumption and allows for slight variations in the representation. Consequently, this class of methods requires extra test-time adaptation on the new tasks (Finn et al., 2017; Raghu et al., 2019; Oh et al., 2020; Collins et al., 2022). Finally, one class of methods assumes that the task reward functions are similar and aims to recover the target reward function with minimal expert transitions (Gleave & Habryka, 2018; Xu et al., 2019; Yu et al., 2019). While most theoretical work indicates that task diversity is critical to improving sample efficiency, the corresponding empirical work lacks the analyses on understanding task diversity or focuses on domain-specific properties (Singh et al., 2020; Xie et al., 2023). Reinforcing existing empirical results on MTIL, we show that increasing task diversity, amount of source tasks, and amount of data typically achieves higher performance.

**Sample Complexity of Multitask Learning**   Our work is heavily inspired by the theoretical analyses that obtain a sample complexity bound when transferring a shared reprsentation to a target task (Arora et al., 2020; Tripuraneni et al., 2020; Maurer et al., 2016). Existing works use Gaussian complexity to quantify the function class complexity to detemine the sample complexity bound; however, we use Rademacher complexity instead, which allows us to connect our analysis with deep-learning theory with improved complexity by a log factor (Bartlett et al., 2021; Truong, 2022b). Our analyses are currently limited to the discrete action space while another line of works focuses on continuous-control problem (Guo et al., 2023; Zhang et al., 2023), where they analyze systems with specific properties (e.g. linear systems.) Our work is similar in that we focus on connecting the source tasks and the target task via task diversity, and provide concrete sample complexity bounds based on the task diversity. However, we remain operating in the infinite-horizon finite MDP setting with non-linear representation class.

## 6   CONCLUSION AND FUTURE DIRECTIONS

We have theoretically shown a sample complexity bound to recover the expert policy when transferring representation from a set of source tasks to a target task. Our bound is tighter than existing results and is applicable for commonly used neural network architectures without making unrealistic assumptions. Based on our theoretical findings, we propose an estimate to quantify task diversity, which controls the number of source tasks and source data required to obtain near-expert policies. We conduct experiments to further support our theoretical findings. In particular, our empirical analyses show that when we pretrain a representation, increasing the amount source tasks and source data can reduce the amount of target data to imitate the expert policy, compared to training from scratch. The analyses further demonstrate that this trend remains even for Gaussian policies in MDPs with continuous action space. Finally, we demonstrate that our proposed estimate for task diversity is positively correlated with the performance in the target task, suggesting that we can use the estimate to quantify task diversity.

Our work has a few limitations and unveils few open questions that can be addressed in future work. First, our work assumes that there exists one shared representation. This assumption can be relaxed by considering the meta-learning formulation where we allow the representation per task to be in a neighbourhood of some reference representation (Collins et al., 2022). Secondly, we only focus on policies with softmax parameterization. While one can include the temperature hyperparameter in the softmax function to approximate the argmax function, there remains a gap in understanding other commonly used policy parameterizations theoretically (Mei et al., 2020; James & Abbeel, 2022). Third, while our metric can characterize the task diversity, it remains an open problem to develop efficient algorithms that leverage this metric to preemptively reduce transitions in practice. Finally, in our analysis we have considered the ERM but it remains an open question to analyze how to optimize for the representation—we have further assumed access to i.i.d. data on the discounted state-action distribution, which we believe can be extended to the non-i.i.d. setting (Mohri & Rostamizadeh, 2008; Truong, 2022a; Zhang et al., 2023).

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

## A  THEORETICAL ANALYSIS

Our goal is to analyze the sample complexity of obtaining an $\varepsilon$-optimal policy. The analysis contains the following two steps: (1) establishing a sample complexity bound for achieving near optimal risk when transferring the representation to a new target task, and (2) bounding the imitation gap of the transferred policy through the transfer risk.

The first step is heavily inspired by Tripuraneni et al. (2020) but we only leverage Rademacher complexity to quantify the sample complexity bound. Our result differs in that we use only Rademacher complexity throughout the analysis due to our assumptions. The second step leverages the fact that our loss function is a Kullback–Leibler (KL) divergence that allows us to upper bound the imitation gap through the transfer risk. In general, the results may include few extra constant terms including $C_\Phi, C_\mathcal{F}$, and $B$ that follow from our assumptions. We provide the detailed assumptions and proofs in appendix C.

First, we compare the training error between the learned representation and true representation. This error is quantified by their similarity—Tripuraneni et al. (2020) proposed to consider the task-average prediction difference:

**Definition 2** *(Task-average Representation Difference (Tripuraneni et al., 2020).) Fix a function class $\mathcal{F}$, $T$ functions $\boldsymbol{f} = (f_1, \ldots, f_T) \in \mathcal{F}^{\otimes T}$, a loss function $\ell(\cdot, \cdot)$, and data $(s_t, a_t) \sim \mu_{\pi_t^*}$. The task-average representation difference between $\phi, \phi' \in \Phi$ is defined as*

$$\bar{d}_\mathcal{F}(\phi'; \phi, \boldsymbol{f}) := \frac{1}{T} \sum_{t=1}^{T} \inf_{f' \in \mathcal{F}} \mathbb{E}_{(s_t, a_t)} \left[ \ell((f' \circ \phi')(s_t), a_t) - \ell((f_t \circ \phi)(s_t), a_t) \right]. \tag{7}$$

The error between the learned representation $\hat{\phi}$ and the true representation $\phi^*$ can be upper bounded by the task-average difference—the bound is dependent on the number of source tasks $T$, the amount of training data per task $N$, and the Rademacher complexity of the representation class.

**Theorem 2** *(Learned Representation Risk Bound.) Let $\hat{\phi}$ be the ERM of $\hat{R}_{train}$ defined in equation 1. Let $\phi^*$ be the true representation and $\boldsymbol{f}^* = (f_1^*, \ldots, f_T^*) \in \mathcal{F}^{\otimes T}$ be the $T$ true task-specific mappings. Under some assumptions, we have that with probability at least $1 - \delta$,*

$$\bar{d}_\mathcal{F}(\hat{\phi}; \phi^*, \boldsymbol{f}^*) \leq 8\sqrt{2} C_\mathcal{F} \mathfrak{R}_{NT}(\Phi) + 2B \sqrt{\frac{\log(2/\delta)}{2NT}}.$$

We now consider the test error when using the learned representation on the target task. This requires a different notion of similarity—Tripuraneni et al. (2020) proposed to consider the worst-case task-specific mapping in $\mathcal{F}$.

**Definition 3** *(Worst-case Representation Difference (Tripuraneni et al., 2020).) Fix a task $\tau$, a function class $\mathcal{F}$, a loss function $\ell(\cdot, \cdot)$, and data $(s, a) \sim \mu_{\pi_\tau^*}$. The worst-case representation difference between $\phi, \phi' \in \Phi$ is defined as*

$$d_{\tau, \mathcal{F}}(\phi'; \phi) := \sup_{f \in \mathcal{F}} \inf_{f' \in \mathcal{F}} \mathbb{E}_{(s, a)} \left[ \ell((f' \circ \phi')(s), a) - \ell((f \circ \phi)(s), a) \right]. \tag{8}$$

We can then use definition 3 to upper bound the generalization error for the transfer phase ERM estimator:

**Theorem 3** *(Transfer Risk Bound.) Let $\hat{f}_\tau$ be the ERM of $\hat{R}_{test}$ defined in equation 2 with some fixed $\hat{\phi} \in \Phi$. Under some assumptions, then with probability $1 - \delta$,*

$$R_{transfer}(\hat{f}_\tau, \hat{\phi}) \leq 8 C_\mathcal{F} C_\Phi \sqrt{\frac{|\mathcal{A}|}{M}} + 2B \sqrt{\frac{\log(2/\delta)}{2M}} + d_{\tau, \mathcal{F}}(\hat{\phi}; \phi^*). \tag{9}$$

We connect the training error of the learned representation $\hat{\phi}$ with the transfer risk via the notion of "task diversity" similar to the one defined by Tripuraneni et al. (2020):

**Definition 4** *($\sigma$-diversity.) Let $\sigma > 0$ and fix a task $\tau$. Fix a function class $\mathcal{F}$, $T$ functions $\boldsymbol{f} = (f_1, \ldots, f_T) \in \mathcal{F}^{\otimes T}$. The $T$ tasks are $\sigma$-diverse for representation $\phi$, if for all $\phi' \in \Phi$, we have that $d_{\tau, \mathcal{F}}(\phi'; \phi) \leq \bar{d}_\mathcal{F}(\phi'; \phi, \boldsymbol{f})/\sigma$.*

Intuitively, if the $T$ source tasks differ too much from the new task, then inequality only holds with small $\sigma$. In other words, there is little "diversity". Another perspective is that any $\phi'$ is overfitted to the $T$ tasks and is unable to generalize to the new task. Conseqeuntly, combining theorems 2 and 3, we get the following sample complexity bound:

**Corollary 1** *(Learned Representation Transfer Risk Bound.) Let $\hat{\phi}$ be the ERM of $\hat{R}_{train}$ defined in equation 1 and let $\hat{f}_\tau$ be the ERM of $\hat{R}_{test}$ defined in equation 2 by fixing $\hat{\phi}$. Suppose the source tasks are $\sigma$-diverse. Under some assumptions, we have that with probability $1 - 2\delta$, the transfer risk $R_{transfer}(\hat{f}_\tau, \hat{\phi})$ is upper bounded by:*

$$\mathcal{O}\left(C_\mathcal{F} C_\Phi \sqrt{\frac{|\mathcal{A}|}{M}} + B\sqrt{\frac{\log(2/\delta)}{M}} + \frac{1}{\sigma}\left(C_\mathcal{F}\mathfrak{R}_{NT}(\Phi) + B\sqrt{\frac{\log(2/\delta)}{NT}}\right)\right). \tag{10}$$

Corollary 1 tells us that so long as we can bound the task-average representation difference, we can achieve $\varepsilon$-error with sufficiently large number of demonstration data in the target task. Furthermore, if the source tasks are not diverse, then we will need more samples from them to reduce the error. Finally, since the log loss is the KL-divergence with deterministic experts, we can use the result from Xu et al. (2020) to obtain the imitation gap of the policy induced by the transfer-learning procedure:

**Theorem 4** *(Imitation Gap Bound (Xu et al., 2020).)* *Given any two policies $\pi, \pi'$ with $\mathbb{E}_{s\sim\nu_\pi}\left[D_{KL}(\pi(s)\|\pi'(s))\right] < \varepsilon$, we have that $\|v^\pi - v^{\pi'}\|_\infty \le \frac{2\sqrt{2}}{(1-\gamma)^2}\sqrt{\varepsilon}$.*

Thus, by theorem 1, for specific representation class, we can upper bound their Rademacher complexities and retrieve a concrete sample complexity bound to achieve $\varepsilon$-optimal policies with high probability.

**Remark 3** It is possible to extend this result to specific continuous state and action spaces. The former is trivial (e.g. our result holds in Euclidean space.) The latter may require more steps— however in many real-life applications, the action values are often bounded even if the action space is continuous. Consequently, we can use the standard covering argument (Wainwright, 2019) to quantize the action space, at the price of some approximation error.

# B  KL-DIVERGENCES AS A PROXY TO TASK DIVERSITY

In appendix A, we defined $\sigma$-diversity to describe task diversity. In practice, we are unable to directly obtain the representation differences, $\bar{d}_\mathcal{F}$ and $d_{\tau,\mathcal{F}}$. We note that in both definitions, the first term is the KL-divergence between the expert policy and the learner policy when we set $\ell$ to be the log-softmax loss, and the second term in equation 7 and equation 8 is a constant. Further notice that the second term implies the realizability of $f^*$ if we set $\phi = \phi^*$. Thus if the second term has no almost influence to the representation differences, we can almost recover the true $f^*$. Based on this insight, we define two risks that will enable us to obtain a diversity estimate:

**Definition 5** *(Task-average Train Risk.) Fix a function class $\mathcal{F}$ and a loss function $\ell(\cdot, \cdot)$, and data $(s_t, a_t) \sim \mu_{\pi_t^*}$. With task-specific mappings $\boldsymbol{f}' = (f_1', \dots, f_T') \in \mathcal{F}^{\otimes T}$, the task-average risk is defined as*

$$R_T(\phi', \boldsymbol{f}') := \frac{1}{T}\sum_{t=1}^T \mathbb{E}_{(s_t, a_t)}\left[\ell((f_t' \circ \phi')(s_t), a_t)\right]. \tag{11}$$

**Definition 6** *(Best-case Map-specific Test Risk.) Fix a function class $\mathcal{F}$, a loss function $\ell(\cdot, \cdot)$, and data $(s_t, a_t) \sim \mu_{\pi_t^*}$. With map-specific mappings $\boldsymbol{f}' = (f_1', \dots, f_T') \in \mathcal{F}^{\otimes T}$, the best-case map-specific risk is defined as*

$$R_\tau(\phi', \boldsymbol{f}') := \min_{t\in[T]} \mathbb{E}_{(s_\tau, a_\tau)}\left[\ell((f_t' \circ \phi')(s_\tau), a_\tau)\right]. \tag{12}$$

We can now define an approximation of task diversity:

**Definition 7** *($(\delta, \sigma)$-diversity.) Let $\sigma > 0, \delta \ge 0$, and fix a task $\tau$. Fix a function class $\mathcal{F}$, a true representation $\phi^* \in \Phi$ and $T$ functions $\boldsymbol{f} = (f_1, \dots, f_T) \in \mathcal{F}^{\otimes T}$. Given a representation $\phi' \in \Phi$ and $T$ functions $\boldsymbol{f}' = (f_1', \dots, f_T') \in \mathcal{F}^{\otimes T}$, the pair $(\phi', \boldsymbol{f}')$ is $(\delta, \sigma)$-diverse for the target task $\tau$, if:*

(i) *The $T$ tasks are $\sigma$-diverse for $\phi^*$,*

(ii) *the best-case map-specific test risk can be upper bounded by $1/\sigma$ the task-average train risk: $R_\tau(\phi', \boldsymbol{f}') \leq R_T(\phi', \boldsymbol{f}')/\sigma$, and*

(iii) *the ratio between the gaps are sufficiently small:*

$$1 - \delta \leq \frac{R_T(\phi', \boldsymbol{f}')/\sigma - R_\tau(\phi', \boldsymbol{f}')}{\bar{d}_{\mathcal{F}}(\phi'; \phi^*, \boldsymbol{f}^*)/\sigma - d_{\tau, \mathcal{F}}(\phi'; \phi^*))} \leq 1 + \delta.$$

Assuming that the learned representation $\hat{\phi}$ and task-specific mappings $\hat{\boldsymbol{f}}$ are $(\delta, \sigma)$-diverse for the target task, we can obtain an estimate $\hat{\sigma} > 0$ through the ratio of the expected KL-divergences:

$$\hat{\sigma} = \frac{\sum_{t=1}^{T} \mathbb{E}_{s_t} \left[ D_{KL}(\pi_t^*(s_t) \| \mathrm{softmax}(\hat{f}_t \circ \hat{\phi})(s_t)) \right]}{T \min_{t \in [T]} \mathbb{E}_{s_\tau} \left[ D_{KL}(\pi_\tau^*(s_\tau) \| \mathrm{softmax}(\hat{f}_t \circ \hat{\phi})(s_\tau)) \right]}.$$

Intuitively, this measures whether the best imitation policy from the source tasks can perform well in the target task. If so, then the target task is "close" to the corresponding source task. On the other hand, if the best imitation policy performs poorly in the target task, then the target task is "far" from the closest source task. We note that our estimate is asymmetrical, which is a critical property for transferring (Sugiyama et al., 2007; Mansour et al., 2009; Hanneke & Kpotufe, 2019).

## C    DETAILED PROOFS

In this section we provide the proofs for the theorems in appendix A. We first list out our assumptions (appendix C.1), then a few known lemmas and definitions used (appendix C.2), and finally the proofs of our theorems (appendices C.3–C.6).

### C.1    ASSUMPTIONS

In the analysis, we offload the softmax function to the loss, then we can define a new loss function that can be used for analysis. That is, let $x \in \mathbb{R}^D$, then we define the log-softmax loss to be

$$\ell(x, a) = -\log\left(\mathrm{softmax}_a(x)\right), \tag{13}$$

where $\mathrm{softmax}_a(x)$ corresponds to the $a$'th component of $\mathrm{softmax}(x)$.

We further make the following assumptions for the analysis.

**Assumption 1** *(Bounded Representation.) The representation $\phi \in \Phi$ is bounded in $\ell_2$-norm: $\Phi \subseteq \{\phi : \mathcal{S} \to \mathbb{R}^D | \|\phi(s)\|_2 \leq C_\Phi, \forall s \in \mathcal{S}\}$.*

**Assumption 2** *(Linear Task-specific Mapping and Bounded Parameters.) The task-specific mapping is linear and is bounded: $\mathcal{F} = \{f : \mathbb{R}^D \to \mathbb{R}^{|\mathcal{A}|} | f = Wx, W \in \mathbb{R}^{|\mathcal{A}| \times D}, \|W\|_F \leq C_{\mathcal{F}}, x \in \mathbb{R}^D\}$, where $\|\cdot\|_F$ is the Frobenius norm*

**Assumption 3** *(Deterministic Expert Policies.) The expert policy $\pi_\tau^*$ for each task $\tau$ is deterministic and can be written as $\pi_\tau^*(s) = \mathbf{1}_{a = \arg\max_{a' \in \mathcal{A}}(f_\tau^* \circ \phi^*)(s)}$ for all $s \in \mathcal{S}$.*

**Assumption 4** *(Shared Representation.) There is a representation $\phi^*$ such that for every task $\tau$, there exists a task-specific mapping $f_\tau^*$ such that the discounted state-action stationary distribution is $\mu_{\pi_\tau^*}(s, a)$.*

**Assumption 5** *(Realizability.) The true shared representation $\phi^*$ is contained in $\Phi$. Additionally, for some fixed $\zeta < 1/2$, we have that for all tasks $\tau$, there exists a task-specific mapping $f_\tau \in \mathcal{F}$ such that for all $s \in \mathcal{S}$,*

$$\pi^{f_\tau, \phi^*}(a^*|s) \geq 1 - \zeta,$$

*where $a^* = \arg\max_{a \in \mathcal{A}} \pi_\tau^*(s)$.*

Note that for any infinite-horizon MDPs, there always exists a deterministic optimal policy, thus assumption 3 is often reasonable to obtain. With the recent successes in foundation models in various applications (Bommasani et al., 2021; Wei et al., 2021; Ouyang et al., 2022), both assumptions 4 and 5 may be reasonable. We note that our results also apply when replacing assumptions 3 and 5 with the standard realizability assumption on stochastic expert policies. Finally, assumptions 1 and 2 are standard regularity conditions in statistical learning theory—we further note that consequently the composed mapping $f \circ \phi$ is bounded: $\sup_{(s,a) \in \mathcal{S} \times \mathcal{A}} |(f \circ \phi)(s)| \leq C_{\mathcal{S}}$, for any $f \in \mathcal{F}$ and $\phi \in \Phi$.

## C.2 Useful Definitions and Results

**Proposition 1** *With assumptions 1 and 2, the log-softmax loss, defined in equation 13, is bounded and Lipschitz (continuous) in its first argument.*

**Proof:** For the first claim, we first fix $(s, a) \in \mathcal{S} \times \mathcal{A}$. Let $\phi_s = \phi(s) \in \mathbb{R}^D$ and $x_s = W\phi_s$. Then, we have that

$$-\log\left(\mathrm{softmax}_a(x_s)\right) = -\left(W\phi_s\right)(a) + \log \sum_{a' \in \mathcal{A}} \exp\left[\left(W\phi_s\right)(a')\right].$$

Let $a^* = \arg\max_{a \in \mathcal{A}} W\phi_s$ and overload the notation $(W\phi_s)(a) = \phi_s^\top W^\top a$, where $a$ is a one-hot vector with non-zero at the $a$'th entry. Then, the second term can be upper bounded:

$$\log \sum_{a' \in \mathcal{A}} \exp\left[\phi_s^\top W^\top a'\right] \leq \log \sum_{a' \in \mathcal{A}} \exp\left[\phi_s^\top W^\top a^*\right]$$

$$= \log|\mathcal{A}| + \phi_s^\top W^\top a^*.$$

Thus,

$$-\log\left(\mathrm{softmax}_a(x_s)\right) \leq -\phi_s^\top W^\top a + \log|\mathcal{A}| + \phi_s^\top W^\top a^*$$

$$\leq \log|\mathcal{A}| + 2\phi_s^\top W^\top a^*.$$

Taking the supremum norm over $\mathcal{S} \times \mathcal{A}$, we have that

$$\sup_{(s,a) \in \mathcal{S} \times \mathcal{A}} \left| -\log\left(\mathrm{softmax}_a(x_s)\right) \right| \leq \sup_{(s,a) \in \mathcal{S} \times \mathcal{A}} \left| \log|\mathcal{A}| + 2\phi_s^\top W^\top a_s^* \right|$$

$$\leq \log|\mathcal{A}| + 2 \sup_{(s,a) \in \mathcal{S} \times \mathcal{A}} \left| \phi_s^\top W^\top a^* \right|$$

$$\leq \log|\mathcal{A}| + 2C_\Phi C_\mathcal{F},$$

where the last inequality follows upper bounding the second term through Hölder's inequality, setting both norms to be the 2-norm. This verifies the boundedness of the log-softmax loss.

For the second claim, we can bound the gradient of the log-softmax loss with respect to the first argument and apply mean-value theorem. Let us first consider the partial derivatives of $\ell(x, a)$. Let $x_i$ be the $i$'th component of $x$. For any $x \in \mathbb{R}^D$, we have that

$$\frac{\partial}{\partial x_i} \ell(x, a) \Big|_{i=a} = -\frac{\sum_{a' \neq i} \exp x^\top a'}{\sum_{a' \in \mathcal{A}} \exp x^\top a'} \qquad \frac{\partial}{\partial x_i} \ell(x, a) \Big|_{i \neq a} = \frac{\exp x^\top i}{\sum_{a' \in \mathcal{A}} \exp x^\top a'},$$

where we overload the notation and represent $a' \in \mathcal{A}$ as a one-hot vector. Then, consider the $\ell_2$-norm of the gradient, we have that:

$$\|\nabla_x \ell(x, a)\|_2^2 = \left(-\frac{\sum_{a' \neq a} \exp x^\top a'}{\sum_{a' \in \mathcal{A}} \exp x^\top a'}\right)^2 + \frac{\sum_{a' \neq a} \left(\exp x^\top a'\right)^2}{\left(\sum_{a' \in \mathcal{A}} \exp x^\top a'\right)^2}$$

$$\leq 2\left(\frac{\sum_{a' \neq a} \exp x^\top a'}{\sum_{a' \in \mathcal{A}} \exp x^\top a'}\right)^2$$

$$= 2,$$

where the first inequality follows from Jensen's inequality. Consequently, by mean-value theorem, we have that $|\ell(x, a) - \ell(y, a)| \leq \sqrt{2}\|x - y\|_2$, for any $x, y \in \mathbb{R}^D$, verifying the Lipschitzness of the log-softmax loss in its first argument. $\square$

**Proposition 2** *Let $\ell$ be the log-softmax loss defined in equation 13 and $f \in \mathcal{F}$. Under assumptions 1 and 2, the function $h_a(\phi(s)) = \ell(f(\phi(s)), a)$ is Lipschitz in $\phi(s)$, for any $s \in \mathcal{S}$.*

**Proof:** We first note that $h_a(\phi_s)$ can be written as $\log \sum_{b \in \mathcal{A}} \exp\left(b^\top W \phi_s\right) - a^\top W \phi_s$, where $\phi_s = \phi(s) \in \mathbb{R}^D$, and we overload the notation and write $a, b$ as the one-hot vectors.

For the first term, the $\ell_2$-norm of its gradient with respect to $\phi_s$ can be upper bounded by $C_{\mathcal{F}}$:

$$
\left\| \nabla_{\phi_s} \log \sum_{b \in \mathcal{A}} \exp\left(b^\top W \phi_s\right) \right\|_2 = \left\| \frac{\sum_b \exp\left(b^\top W \phi_s\right) W^\top b}{\sum_b \exp\left(b^\top W \phi_s\right)} \right\|_2
$$
$$
= \|W^\top \mathrm{softmax}(W \phi_s)\|_2
$$
$$
\leq \|W\|_F \|\mathrm{softmax}(W \phi_s)\|_2
$$
$$
\leq C_{\mathcal{F}}.
$$

For the second term, the $\ell_2$-norm of its gradient with respect to $\phi_s$ can be upper bounded by $C_{\mathcal{F}}$:

$$
\|\nabla_{\phi_s} a^\top W \phi_s\|_2 = \|W^\top a\|_2 \leq \|W\|_F \|a\|_2 \leq C_{\mathcal{F}}.
$$

Since the first term is convex and the second term is linear, $h_a(\phi_s)$ is $2C_{\mathcal{F}}$-Lipschitz. $\qquad\square$

**Definition 8** *(Bounded Difference Property.) The function $f : \mathbb{R}^N \to \mathbb{R}$ satisfies the bounded difference inequality with positive constants $(L_1, \dots, L_N)$ if, for each $n \in [N]$,*

$$
\sup_{x_1, \dots, x_N, x_n' \in \mathbb{R}} |f(x_1, \dots, x_n, \dots, x_N) - f(x_1, \dots, x_n', \dots, x_N)| \leq L_n. \tag{14}
$$

**Lemma 1** *(McDiarmid's Inequality/Bounded Difference Inequality.) Fix a data distribution $\mathcal{D}_{\mathcal{X}}$. Suppose $f$ satisfies the bounded difference property defined in equation 14, with positive constants $L_1, \dots, L_N$ and that $X = (X_1, \dots, X_N)$ is drawn independently from $\mathcal{D}_{\mathcal{X}}$. Then*

$$
\mathbb{P}\left[ |f(X) - \mathbb{E}f(X)| \geq t \right] \leq 2 \exp\left( \frac{-2t^2}{\sum_{n=1}^{N} L_n^2} \right), \forall t \geq 0.
$$

**Proof:** We refer the readers to corollary 2.21 of Wainwright (2019) for the proof. $\qquad\square$

**Theorem 5** *(Rademacher Complexity Bound.) Fix a data distribution $\mathcal{D}_{\mathcal{X}}$ and parameter $\delta \in (0, 1)$. Suppose $\mathcal{F} \subseteq \{f : \mathcal{X} \to [0, B]\}$ and $X = (X_1, \dots, X_N)$ is drawn i.i.d. from $\mathcal{D}_{\mathcal{X}}$. Then with probability at least $1 - \delta$ over the draw of $X$, for any function $f \in \mathcal{F}$,*

$$
\left| \mathbb{E}_{x \sim \mathcal{D}_{\mathcal{X}}} [f(x)] - \frac{1}{n} \sum_{n=1}^{N} f(X_n) \right| \leq 2 \mathfrak{R}_N(\mathcal{F}) + B \sqrt{\frac{\log(2/\delta)}{2n}}. \tag{15}
$$

**Proof:** This result follows closely to the proof of Theorem 10 in Koltchinskii & Panchenko (2002), where the only modifications come from applying $B$-boundedness of $f$ when applying McDiarmid's inequality (i.e. $L_n \leq \frac{B}{N}, \forall n \in [N]$.) $\qquad\square$

**Theorem 6** *(Vector-contraction Inequality (Maurer, 2016).) Let $x_1, \dots, x_N \in \mathcal{X}$, $\mathcal{F} \subseteq \{f : \mathcal{X} \to \mathbb{R}^D\}$ be a class of functions, and $h_n : \mathbb{R}^D \to \mathbb{R}$ to be $L$-Lipschitz, for all $n \in [N]$. Then*

$$
\mathbb{E}\left[ \sup_{f \in \mathcal{F}} \sum_{n=1}^{N} \varepsilon_n h_n(f(x_n)) \right] \leq \sqrt{2} L \mathbb{E}\left[ \sup_{f \in \mathcal{F}} \sum_{d=1}^{D} \sum_{n=1}^{N} \varepsilon_{d,n} f_d(x_n) \right],
$$

*where $\varepsilon_n, \varepsilon_{d,n}$ are i.i.d. sequences of Rademacher variables, and $f_d(x_n)$ is the d'th component of $f(x_n)$.*

**Proof:** We refer the readers to theorem 3 and corollary 4 of Maurer (2016) for the proof. $\qquad\square$

## C.3 Proof of Theorem 2

Theorem 2 states the following:

*Let $\hat{\phi}$ be the ERM of $\hat{R}_{train}$ defined in equation 1. Let $\phi^*$ be the true representation and $\boldsymbol{f}^* = (f_1^*, \ldots, f_T^*) \in \mathcal{F}^{\otimes T}$ be the T true task-specific mappings. If the assumptions 1 to 5 hold, then with probability $1 - \delta$,*

$$\bar{d}_{\mathcal{F}}(\hat{\phi}; \phi^*, \boldsymbol{f}^*) \leq 8\sqrt{2} C_{\mathcal{F}} \mathfrak{R}_{NT}(\Phi) + 2B\sqrt{\frac{\log(2/\delta)}{2NT}}.$$

**Proof:** The proof follows closely from the analysis of Tripuraneni et al. (2020). First, recall that $\hat{\phi}, \hat{\boldsymbol{f}}$ are respectively the ERMs of equation 1 and equation 2. For any $T$ task-specific mappings $\boldsymbol{f}' = (f_1', \ldots, f_T')$ and a representation $\phi'$. Let $\boldsymbol{f}^* = (f_1^*, \ldots, f_T^*)$ be the true task-specific mappings and $\phi^*$ be the true shared representation. We define the centered training risk and its empirical counterpart respectively as:

$$L(\boldsymbol{f}', \phi', \boldsymbol{f}^*, \phi^*) = \frac{1}{T} \sum_{t=1}^{T} \mathbb{E}_{(s_t, a_t)} \left[ \ell(\pi^{f_t', \phi'}(s_t), a_t) - \ell(\pi^{f_t^*, \phi^*}(s_t), a_t) \right],$$

$$\hat{L}(\boldsymbol{f}', \phi', \boldsymbol{f}^*, \phi^*) = \frac{1}{T} \sum_{t=1}^{T} \sum_{n=1}^{N} \left( \ell(\pi^{f_t', \phi'}(s_{t,n}), a_{t,n}) - \mathbb{E}_{(s_t, a_t)} \left[ \ell(\pi^{f_t^*, \phi^*}(s_t), a_t) \right] \right).$$

Define $\tilde{\boldsymbol{f}} = \frac{1}{T} \sum_{t=1}^{T} \arg\inf_{f_t \in \mathcal{F}} \mathbb{E}_{s_t, a_t} \left[ \ell(\pi^{f_t, \hat{\phi}}(s_t), a_t) - \ell(\pi^{f_t^*, \phi^*}(s_t), a_t) \right]$ to be the minimizer of the centered training risk by fixing $\hat{\phi}$. Then, we have that $L(\tilde{\boldsymbol{f}}, \hat{\phi}, \boldsymbol{f}^*, \phi^*) = \bar{d}_{\mathcal{F}}(\hat{\phi}; \phi^*, \boldsymbol{f}^*)$. Now, we aim to upper bound $\bar{d}_{\mathcal{F}}(\hat{\phi}; \phi^*, \boldsymbol{f}^*)$ through the difference in the centered training risk:

$$\bar{d}_{\mathcal{F}}(\hat{\phi}; \phi^*, \boldsymbol{f}^*) = L(\tilde{\boldsymbol{f}}, \hat{\phi}, \boldsymbol{f}^*, \phi^*) - L(\boldsymbol{f}^*, \phi^*, \boldsymbol{f}^*, \phi^*)$$
$$= \underbrace{L(\tilde{\boldsymbol{f}}, \hat{\phi}, \boldsymbol{f}^*, \phi^*) - L(\hat{\boldsymbol{f}}, \hat{\phi}, \boldsymbol{f}^*, \phi^*)}_{\leq 0} + L(\hat{\boldsymbol{f}}, \hat{\phi}, \boldsymbol{f}^*, \phi^*) - L(\boldsymbol{f}^*, \phi^*, \boldsymbol{f}^*, \phi^*),$$

where the first difference is non-positive by definition of $\tilde{\boldsymbol{f}}$. Thus, it remains to bound the second difference, which can be done via standard risk decomposition. First, recall that:

$$\hat{R}_{train}(\boldsymbol{f}, \phi) = \frac{1}{NT} \sum_{t=1}^{T} \sum_{n=1}^{N} \ell(\pi^{f_t, \phi}(s_{t,n}), a_{t,n}),$$

$$R_{train}(\boldsymbol{f}, \phi) = \mathbb{E} \left[ \hat{R}_{train}(\boldsymbol{f}, \phi) \right].$$

Then, we have that

$$L(\hat{\boldsymbol{f}}, \hat{\phi}, \boldsymbol{f}^*, \phi^*) - L(\boldsymbol{f}^*, \phi^*, \boldsymbol{f}^*, \phi^*) = L(\hat{\boldsymbol{f}}, \hat{\phi}, \boldsymbol{f}^*, \phi^*) - \hat{L}(\hat{\boldsymbol{f}}, \hat{\phi}, \boldsymbol{f}^*, \phi^*)$$
$$+ \hat{L}(\hat{\boldsymbol{f}}, \hat{\phi}, \boldsymbol{f}^*, \phi^*) - \hat{L}(\boldsymbol{f}^*, \phi^*, \boldsymbol{f}^*, \phi^*)$$
$$+ \hat{L}(\boldsymbol{f}^*, \phi^*, \boldsymbol{f}^*, \phi^*) - L(\boldsymbol{f}^*, \phi^*, \boldsymbol{f}^*, \phi^*)$$
$$\overset{(i)}{\leq} L(\hat{\boldsymbol{f}}, \hat{\phi}, \boldsymbol{f}^*, \phi^*) - \hat{L}(\hat{\boldsymbol{f}}, \hat{\phi}, \boldsymbol{f}^*, \phi^*)$$
$$+ \hat{L}(\boldsymbol{f}^*, \phi^*, \boldsymbol{f}^*, \phi^*) - L(\boldsymbol{f}^*, \phi^*, \boldsymbol{f}^*, \phi^*)$$
$$\leq 2 \sup_{\boldsymbol{f} \in \mathcal{F}^{\otimes T}, \phi \in \Phi} |R_{train}(\boldsymbol{f}, \phi) - R_{train}(\boldsymbol{f}^*, \phi^*)|$$
$$\overset{(ii)}{\leq} 2 \left( 2\mathfrak{R}_{NT}(\ell \circ \mathcal{F} \circ \Phi) + B\sqrt{\frac{\log(2/\delta)}{2NT}} \right) \text{ w.p. } 1 - \delta,$$

where in (i) the second difference is non-positive due to assumptions 3 and 5; (ii) is a result of theorem 5, where we used proposition 1.

We now bound the Rademacher complexity term $\Re_{NT}(\ell \circ \mathcal{F} \circ \Phi)$ Note that $\ell \circ \mathcal{F}$ is $2C_{\mathcal{F}}$-Lipschitz by proposition 2, thus by theorem 6, we have that $\Re_{NT}(\ell \circ \mathcal{F} \circ \Phi) \leq 2\sqrt{2}C_{\mathcal{F}}\Re_{NT}(\Phi)$. Substituting this upper bound into the above, we get that with probability at least $1 - \delta$,

$$\bar{d}_{\mathcal{F}}(\hat{\phi}; \phi^*, \boldsymbol{f}^*) \leq 8\sqrt{2}C_{\mathcal{F}}\Re_{NT}(\Phi) + 2B\sqrt{\frac{\log(2/\delta)}{2NT}}.$$

$\square$

### C.4 PROOF OF THEOREM 3

Theorem 3 states the following:

*let $\hat{f}_\tau$ be the ERM of $\hat{R}_{test}$ defined in equation 2 with some fixed $\hat{\phi} \in \Phi$. If the assumptions 1 to 5 hold, then with probability $1 - \delta$,*

$$R_{transfer}(\hat{f}_\tau, \hat{\phi}) \leq 8C_{\mathcal{F}}C_{\Phi}\sqrt{\frac{|\mathcal{A}|}{M}} + 2B\sqrt{\frac{\log(2/\delta)}{2M}} + d_{\tau,\mathcal{F}}(\hat{\phi}; \phi^*).$$

**Proof:** The proof follows closely from the analysis of Tripuraneni et al. (2020). First recall that:

$$R_{transfer}(\hat{f}_\tau, \hat{\phi}) = R_{test}(\hat{f}_\tau, \hat{\phi}) - R_{test}(f_\tau^*, \phi^*),$$

$$R_{test}(f, \phi) = \mathbb{E}\left[\hat{R}_{test}(f, \phi)\right],$$

$$\hat{R}_{test}(f, \phi) = \frac{1}{M}\sum_{m=1}^{M}\ell((f \circ \phi)(s_m), a_m),$$

$$\hat{f} = \arg\min_{f \in \mathcal{F}}\hat{R}_{test}(f, \hat{\phi}).$$

Consider the minimizer of the test risk given the estimated representation $\hat{\phi}$: $\hat{f}^* = \arg\min_{f \in \mathcal{F}} R_{test}(f, \hat{\phi})$. Then, we have that

$$R_{test}(\hat{f}_\tau, \hat{\phi}) - R_{test}(f_\tau^*, \phi^*)$$
$$= \underbrace{R_{test}(\hat{f}_\tau, \hat{\phi}) - R_{test}(\hat{f}_\tau^*, \hat{\phi})}_{(a)} + \underbrace{R_{test}(\hat{f}_\tau^*, \hat{\phi}) - R_{test}(f_\tau^*, \phi^*)}_{(b)}.$$

We bound the term (a) via standard risk decomposition:

$$R_{test}(\hat{f}_\tau, \hat{\phi}) - R_{test}(\hat{f}_\tau^*, \hat{\phi})$$
$$= R_{test}(\hat{f}_\tau, \hat{\phi}) - \hat{R}_{test}(\hat{f}_\tau, \hat{\phi}) + \hat{R}_{test}(\hat{f}_\tau, \hat{\phi}) - \hat{R}_{test}(\hat{f}_\tau^*, \hat{\phi}) + \hat{R}_{test}(\hat{f}_\tau^*, \hat{\phi}) - R_{test}(\hat{f}_\tau^*, \hat{\phi})$$
$$\overset{(i)}{\leq} R_{test}(\hat{f}_\tau, \hat{\phi}) - \hat{R}_{test}(\hat{f}_\tau, \hat{\phi}) + \hat{R}_{test}(\hat{f}_\tau^*, \hat{\phi}) - R_{test}(\hat{f}_\tau^*, \hat{\phi})$$
$$\leq 2\sup_{f \in \mathcal{F}}|R_{test}(f, \hat{\phi}) - \hat{R}_{test}(f, \hat{\phi})|$$
$$\overset{(ii)}{\leq} 2\left(2\Re_M(\ell \circ \mathcal{F}) + B\sqrt{\frac{\log(2/\delta)}{2M}}\right) \text{ w.p. } 1 - \delta,$$

where (i) follows from the fact that $\hat{f}$ is the empirical test risk minimizer by definition—clearly $\hat{R}_{test}(\hat{f}_\tau, \hat{\phi}) - \hat{R}_{test}(\hat{f}_\tau^*, \hat{\phi}) \leq 0$; (ii) is a result of theorem 5.

We now bound the Rademacher complexity term $\Re_M(\ell \circ \mathcal{F})$. Note that $\ell$ is $\sqrt{2}$-Lipschitz in its first argument for every $a \in \mathcal{A}$ from proposition 1. By theorem 6, we have that $\Re_M(\ell \circ \mathcal{F}) \leq 2\Re_M(\mathcal{F})$. Thus, substituting this upper bound into the above, we get that with probability at least $1 - \delta$,

$$R_{test}(\hat{f}_\tau, \hat{\phi}) - R_{test}(\hat{f}_\tau^*, \hat{\phi}) \leq 8\Re_M(\mathcal{F}) + 2B\sqrt{\frac{\log(2/\delta)}{2M}}.$$

Now, by assumption 2, $\mathcal{F}$ is the class of linear functions with bounded Frobenius norm. Consider the empirical Rademacher complexity $\hat{\mathfrak{R}}_S(\mathcal{F})$:

$$\hat{\mathfrak{R}}_S(\mathcal{F}) = \frac{1}{M}\mathbb{E}\left[\sup_{f\in\mathcal{F}}\sum_{a=1}^{|\mathcal{A}|}\sum_{m=1}^{M}\varepsilon_{am}f_a(\phi_m)\right]$$

$$\overset{(i)}{\leq} \frac{1}{M}C_{\mathcal{F}}\sqrt{|\mathcal{A}|\sum_{m=1}^{M}\|\phi_m\|_2^2}$$

$$\overset{(ii)}{\leq} \frac{1}{M}C_{\mathcal{F}}\sqrt{\frac{|\mathcal{A}|C_\Phi^2}{M}}$$

$$\leq C_{\mathcal{F}}C_\Phi\sqrt{\frac{|\mathcal{A}|}{M}},$$

where (i) follows from section 4.2 of Maurer (2016) and applying the Frobenius norm to the inequality; (ii) follows from the fact that $\phi_m = \phi(s_m), s_m \sim \nu_\pi$, thus $\|\phi_m\|_2 \leq C_\Phi$ by assumption 1. Consequently, taking the expectation over $X$ we get $\mathfrak{R}_M(\mathcal{F}) \leq C_{\mathcal{F}}C_\Phi\sqrt{|\mathcal{A}|/M}$.

We can bound the term (b) as follows:

$$R_{test}(\hat{f}_\tau^*, \hat{\phi}) - R_{test}(f_\tau^*, \phi^*)$$

$$= \inf_{f\in\mathcal{F}} R_{test}(f, \hat{\phi}) - \inf_{f'\in\mathcal{F}} R_{test}(f', \phi^*)$$

$$\leq \sup_{f'\in\mathcal{F}}\inf_{f\in\mathcal{F}} R_{test}(\hat{f}_\tau^*, \hat{\phi}) - R_{test}(f', \phi^*)$$

$$= d_{\mathcal{F}}(\hat{\phi}; \phi^*).$$

Finally, by comining both (a) and (b) terms, we conclude that with probability at least $1 - \delta$,

$$R_{transfer}(\hat{f}_\tau, \hat{\phi}) \leq 8C_{\mathcal{F}}C_\Phi\sqrt{\frac{|\mathcal{A}|}{M}} + 2B\sqrt{\frac{\log(2/\delta)}{2M}} + d_{\tau,\mathcal{F}}(\hat{\phi}; \phi^*).$$

$\square$

## C.5 Proof of Corollary 1

Corollary 1 states the following:

*Let $\hat{\phi}$ be the ERM of $\hat{R}_{train}$ defined in equation 1 and let $\hat{f}_\tau$ be the ERM of $\hat{R}_{test}$ defined in equation 2 by fixing $\hat{\phi}$. Suppose the source tasks are $\sigma$-diverse. If the assumptions 1 to 5 hold, then with probability $1 - 2\delta$, $R_{transfer}(\hat{f}_\tau, \hat{\phi})$ is upper bounded by:*

$$\mathcal{O}\left(C_{\mathcal{F}}C_\Phi\sqrt{\frac{|\mathcal{A}|}{M}} + B\sqrt{\frac{\log(2/\delta)}{M}} + \frac{1}{\sigma}\left(C_{\mathcal{F}}\mathfrak{R}_{NT}(\Phi) + B\sqrt{\frac{\log(2/\delta)}{NT}}\right)\right).$$

**Proof:** Set the probability of bad events of theorems 2 and 3 to be $\delta$ each. We obtain the desired results by taking complment of the union bound over the bad events and merging the terms. $\square$

## C.6 Proof of Theorem 1

Theorem 1 states the following:

*Let $\hat{\phi}$ be the ERM of $\hat{R}_{train}$ defined in equation 1 and let $\hat{f}_\tau$ be the ERM of $\hat{R}_{test}$ defined in equation 2 by fixing $\hat{\phi}$. Suppose the source tasks are $\sigma$-diverse. Under assumptions 1 to 5, we have that with probability $1 - 2\delta$,*

$$\|v^{\pi_\tau^*} - v^{\text{softmax}(\hat{f}_\tau\circ\hat{\phi})}\|_\infty \leq \frac{2\sqrt{2}}{(1-\gamma)^2}\sqrt{\varepsilon_{gen} + 2\zeta}, \tag{16}$$

*where $\varepsilon_{gen}$ is the RHS of equation 10.*

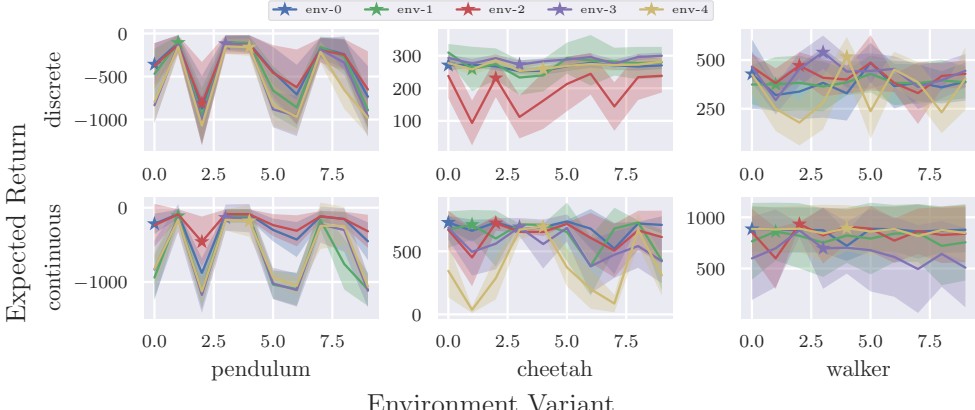

Figure 6: Policy transfer without finetuning results in varying performance across environment variants Stars show the environment variant the policy was trained on.

**Proof:** Our goal is to apply theorem 4. Let $\pi = \pi^*$ and $\pi' = \mathrm{softmax}(\hat{f}_\tau \circ \hat{\phi})$, we need to bound their expected Kullback–Leibler (KL) divergence. To accomplish this, note that theorem 3 essentially means that we can bound the test risk:

$$R_{test}(\hat{f}_\tau, \hat{\phi}) \leq \varepsilon_{gen} + \varepsilon_{best}, \tag{17}$$

where $\varepsilon_{gen}$ is the RHS of equation 10 and $\varepsilon_{best} = R_{test}(f_\tau^*, \phi^*)$. Indeed, $R_{test}(\hat{f}_\tau, \hat{\phi})$ is the expected KL divergence between $\pi_\tau^*(s)$ and $\mathrm{softmax}(\hat{f}_\tau \circ \hat{\phi})(s)$:

$$R_{test}(\hat{f}_\tau, \hat{\phi}) = \mathbb{E}_{(s,a) \sim \mu_{\pi_\tau^*}} \left[ -\log \mathrm{softmax}_a(\hat{f}_\tau \circ \hat{\phi})(s) \right]$$

$$= \mathbb{E}_{s \sim \nu_{\pi^*}} \left[ D_{KL}(\pi_\tau^*(s) \| \mathrm{softmax}(\hat{f}_\tau \circ \hat{\phi})(s)) \right].$$

Finally, note that assumptions 3 and 5 imply that we can assume $f^* \in \mathcal{F}$, thus we have that:

$$\varepsilon_{best} = R_{test}(f^*, \phi^*)$$
$$\leq \min_{f \in \mathcal{F}} R_{test}(f, \phi^*)$$
$$\leq \mathbb{E}_{(s,a) \sim \mu_{\pi_\tau^*}} \left[ -\log(1 - \zeta) \right]$$
$$\leq 2\zeta,$$

where the third lines come from assumption 5 and last inequality comes from $-\log(1 - x) \leq 2x$ for $x \leq 1/2$. Finally, we upper bound $\varepsilon_{gen}$ using corollary 1 and get the desired result. $\square$

## D  EXPERT POLICY ROBUSTNESS

In this section we analyze how policies degrade when they are transferred to a new task without any finetuning. Figure 6 shows that for the pendulum environment, the expert policies trained on their corresponding environment variant perform the best with performance often degrading when the policies are transferred. Surprisingly, cheetah and walker environments have policies that often keep a consistent performance after transfer to most environment variants. The walker policies exhibit walking behaviours and the cheetah policies exhibit running and hopping behaviours, which keep a similar performance as long as the body and joint parameters do not differ too much. Therefore, these policies exhibit robustness to some environment changes.

To demonstrate this, we increase the range of randomized joint damping and stiffness by at least $50\%$ in the cheetah tasks. Figure 7 shows a comparison of policy performance when the environment has small and large parameter variation. The transferred policies degrade in performance when there is a large variation in the environmental parameters. Visually, the cheetah policies often exert too much

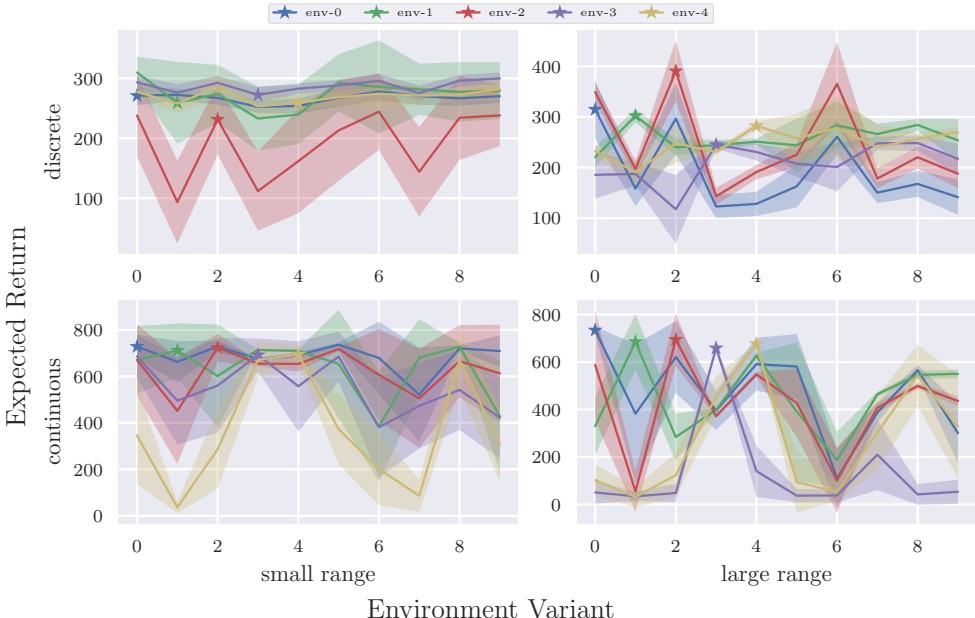

Figure 7: Policy transfer without finetuning under different environmental parameter range in cheetah task.

joint torque and flipping over. We also notice that there are held-out environmental parameters for which these expert policies remain consistent in performance. This suggests that there is a region of environmental parameters for which the expert policy is robust to.

## E   ALGORITHM DETAILS

In this section we provide details on the implementation of the multitask behaviour cloning (MTBC) algorithm used to obtain the results highlighted in section 4. Recall that in the training phase we aim to obtain a shared representation $\hat{\phi}$ by minimizing $\hat{R}_{train}$, as described in equation 1. As proposed by (Arora et al., 2020), this can be done by solving the following bi-level optimization objective:

$$\hat{\phi} = \arg\min_{\phi} \frac{1}{T} \sum_{t=1}^{T} \min_{\pi \in \Pi^{\phi}} \frac{1}{N} \sum_{n=1}^{N} \ell(\pi(s_{t,n}), a_{t,n}), \tag{18}$$

where $(s_{t,n}, a_{t,n})$ is the $n$'th state-action pair in the $t$'th expert dataset and $\ell : \Delta^{\mathcal{A}} \times \mathcal{A} \to [0, \infty)$ is the loss function. MTBC optimizes equation 18 through a gradient-based approach on a joint objective equation 19 (see algorithm 1). During the transfer phase, MTBC fixes the representation $\hat{\phi}$ and minimizes $\hat{R}_{test}(f, \hat{\phi})$, as described in equation 2—this can be done via BC where we set the loss function $\ell$ to be the cross-entropy loss for discrete action space and mean-squared loss for continuous action space. In practice, since $f$ is a linear function parameterized by $W$, MTBC propagates the gradient to update $W$ (see algorithm 2).

## F   IMPLEMENTATION DETAILS

In this section we provide details on the implementation of the environments used in the experiments in section 4.

### F.1   ENVIRONMENTS

We perform empirical analyses on five tasks: frozen lake, pendulum, cartpole swing up, cheetah run, and walker walk. The five environments are shown in figure 8. The first two environments are based

---

**Algorithm 1** Multitask Behavioural Cloning (MTBC): Training Phase

---

1: **input**: Number of epochs $K$, $T$ expert datasets $\{\mathcal{D}_t\}_{t=1}^T$, where $|\mathcal{D}_t| = N$, and learning rates $\eta_\phi$ and $\eta_W$.

2: Initialize parameters for each of the $T$ task-specific mappings: $\{W_t^{(0)}\}_{t=1}^T$.

3: Initialize shared representation parameters $\phi^{(0)}$.

4: **for** $k = 1, \dots, K$ **do**

5:     Compute loss:

$$\mathcal{L}\left(\phi^{(k)}, \{W_t^{(k)}\}_{t=1}^T\right) = \frac{1}{NT} \sum_{t=1}^T \sum_{n=1}^N \ell\left(\pi^{W_t^{(k-1)}, \phi^{(k-1)}}(s_{t,n}), a_{t,n}\right). \tag{19}$$

6:     Update parameters:

$$\phi^{(k)} = \phi^{(k-1)} - \eta_\phi \nabla_\phi \mathcal{L}\left(\phi^{(k)}, \{W_t^{(k)}\}_{t=1}^T\right),$$

$$W_t^{(k)} = W_t^{(k-1)} - \eta_W \nabla_{W_t} \mathcal{L}\left(\phi^{(k)}, \{W_t^{(k)}\}_{t=1}^T\right), \forall t \in [T].$$

7: **return** $\phi$.

---

**Algorithm 2** Multitask Behavioural Cloning (MTBC): Transfer Phase

---

1: **input**: Representation $\phi$, number of epochs $K$, expert dataset $\mathcal{D}$, where $|\mathcal{D}| = M$, and learning rate $\eta_W$.

2: Initialize parameters of the task-specific mapping: $W_\tau^{(0)}$.

3: **for** $k = 1, \dots, K$ **do**

4:     Compute loss: $\mathcal{L}\left(\phi, W_\tau^{(k)}\right) = \frac{1}{M} \sum_{m=1}^M \ell\left(\pi^{W_\tau^{(k-1)}, \phi}(s_m), a_m\right)$

5:     Update parameters: $W_\tau^{(k)} = W_\tau^{(k-1)} - \eta_W \nabla_W \mathcal{L}\left(\phi, W_\tau^{(k)}\right)$.

6: **return** $W^{(K)}$.

---

on Gymnasium (Towers et al., 2023) and the remaining environments are based on Deepmind control suite (Tunyasuvunakool et al., 2020). We validate our theoretical contributions on discrete action space variants of the environments and test how well these contributions hold in the continuous action space variants. While in general discretizing action space may exclude the true optimal policy (Dadashi et al., 2022; Seyde et al., 2021; 2022), our goal is to demonstrate that MTBC can obtain the expert policy with less target data when compared with BC. Thus, we argue that not having the true optimal policy still validates our goal so long as the expert policy is non-trivial. To generate multiple source tasks with shared state and action spaces, we modify phyiscal properties of the environment. All environmental parameters are sampled from uniform distributions of bounded ranges.

**Frozen Lake**    The frozen lake task requires an agent the navigate through a $8 \times 8$ grid with non-deterministic transitions. The action space is the set $\{\text{LEFT}, \text{DOWN}, \text{RIGHT}, \text{UP}, \text{STAY}\}$. To vary the tasks, we sample the initial state, the goal state, and the transition function following the procedure:

    *(i)* We vary the initial and goal positions uniformly (disallowing equal positions).

    *(ii)* We sample uniformly the probability of not slipping $p_1 \sim U[0, 1]$, then sample uniformly the probability of slipping to one adjacent cell $p_2 \sim U[0, 1 - p_1]$, and assign the remaining mass $1 - p_1 - p_2$. This probability is assigned to all state-action pairs.

**Cartpole**    The cartpole task requires an agent to swing and keep the link upright by applying forces onto the cart. The action space is the interval $[-1, 1]$. To vary the tasks, we sample the joint stiffness following the distribution $U[0, 5]$.

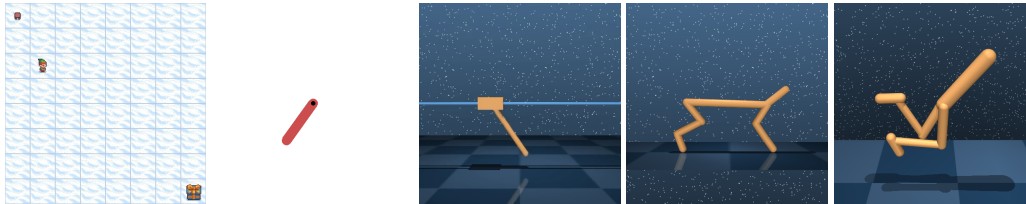

Figure 8: Environments used for empirical analyses. From left to right: frozen lake, pendulum, cartpole, cheetah, and walker.

**Pendulum**  The pendulum task requires an agent to swing and keep the link upright. The default action space is the torque of the revolute joint between $[-2, 2]$. The discretized action space is the set $\{0, \pm2^{-3}, \ldots, , \pm2^{1}\}$ (i.e. 11 actions.) This allows the agent to perform large-magnitude action for swinging the link up and to perform low-magnitude action for keeping the link upright. To vary the tasks, we sample the maximum torque applied to the joint following the distribution $U[0.01, 4]$.

**Locomotion**  Both the cheetah-run and walker-walk tasks require an agent to move above a specified velocity. The latter task further scales the velocity based on the height of the agent. The default action space is the torque applied to each of the joint, all bounded between $[-1, 1]$. The discretized action space is the set $\{-1, 1\}$ per joint (i.e. Bang-Bang control (Seyde et al., 2021),) thus we have $2^{\dim(\mathcal{A})}$ actions after discretizing the action space—we emphasize that we use softmax policies as opposed to per-dimension Bernoulli policies. To vary the source tasks, we sample the links' size and the joint parameters as follows:

*(i)* Cheetah:
- Front and back thigh joints:
    - Damping: $U[4, 8]$
    - Stiffness: $U[230, 250]$
- Front and back feet:
    - Link size: $U[0.04, 0.05] \times U[0.065, 0.075]$

*(ii)* Walker:
- Left and right feet:
    - Link size: $U[0.04, 0.06] \times U[0.09, 0.11]$
- Left and right legs:
    - Link size: $U[0.03, 0.05] \times U[0.23, 0.27]$
- Left and right thighs:
    - Link size: $U[0.04, 0.06] \times U[0.2, 0.25]$
- Left and right knee joint:
    - Range: $U[-140, -160]$

## F.2 GENERATING THE EXPERTS

We use Proximal Policy Optimization (PPO) (Schulman et al., 2017) to train the expert policies for all environments. In general, all PPO policies are trained using Adam optimizer (Kingma & Ba, 2015). We further use observation normalization and advantage normalization, as commonly done in practice (Engstrom et al., 2020; Hsu et al., 2020). For the discrete tasks, we use the reverse-KL objective (Hsu et al., 2020) rather than the commonly used clipped objective:

$$J(\theta) = \mathbb{E}_{s \sim \nu_{\pi^{old}}} \left[ \mathbb{E}_{a \sim \pi^{old}(s)} \left[ \frac{\pi^{\theta}}{\pi^{old}} \hat{A}^{\pi^{old}}(s, a) \right] - \beta D_{KL} \left( \pi^{\theta}(s) \| \pi^{old}(s) \right) \right],$$

where $\pi^{old}$ is the reference policy, $\pi^{\theta}$ is the learner policy, and $\hat{A}^{\pi^{old}}(s, a) = \hat{Q}^{\pi^{old}}(s, a) - \hat{V}^{\pi^{old}}(s)$ is the (estimated) advantage function—we use generalized advantage estimation (GAE) (Schulman et al., 2016) to estimate $\hat{A}^{\pi_{old}}$. Our preliminary result in figure 9 aligns with the findings by Hsu

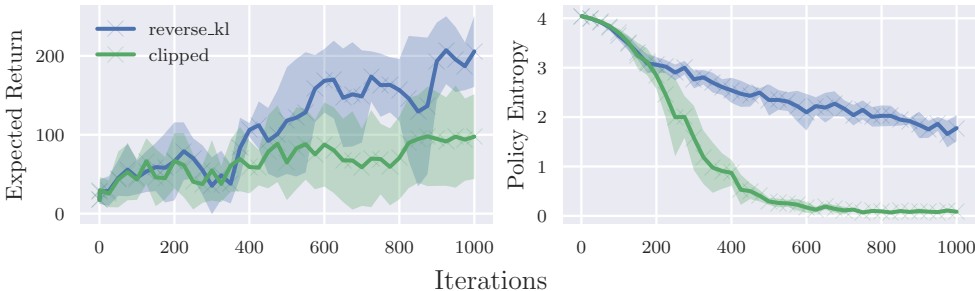

Figure 9: Preliminary analysis on training PPO policies using clipped objective and reverse-KL objective, performed across 5 seeds in the discrete cheetah environment. **Left**: Expected returns during evaluation. **Right**: Policy entropy during training. The solid line corresponds to the mean and the shaded region is 1 standard error from the mean.

Table 3: Hyperparameters for PPO across tasks. $X \xrightarrow{Z} Y$ means linear schedule from $X$ to $Y$ over $Z$ updates.

| | Frozen Lake | Pendulum | Cartpole | Cheetah | Walker |
|---|---|---|---|---|---|
| **Shared** | | | | | |
| Number of epochs | | | 500 | | |
| Interactions per epoch | | | 2048 | | |
| Buffer size | | | 2048 | | |
| $\gamma$ | | | 0.99 | | |
| GAE $\lambda$ | | | 0.95 | | |
| Learning rate | | | $3 \times 10^{-4}$ | | |
| Number of minibatches | | | 200 | | |
| **Discrete** | | | | | |
| Minibatch size | 256 | 128 | N/A | 256 | 128 |
| $\beta$ | $2 \times 10^{-3}$ | $2 \times 10^{-3}$ | N/A | $2 \times 10^{-3}$ | $2 \times 10^{-3}$ |
| Max gradient norm | 0.5 | None | N/A | 0.5 | None |
| Entropy coefficient | $0.002 \xrightarrow{100} 0$ | None | N/A | $0.002 \xrightarrow{100} 0$ | $0.002 \xrightarrow{100} 0$ |
| **Continuous** | | | | | |
| Minibatch size | N/A | 256 | 64 | 64 | 64 |
| Clipping | N/A | 0.2 | 0.1 | 0.1 | 0.1 |
| Max gradient norm | N/A | 0.5 | None | 0.5 | None |
| Entropy coefficient | N/A | None | $0.002 \xrightarrow{100} 0$ | $0.002 \xrightarrow{100} 0$ | $0.002 \xrightarrow{100} 0$ |

et al. (2020), where the softmax policies with tanh activation tend to converge to a suboptimal policy in environments with the clipped objective.

We now describe how we generate the experts for all environments. We first obtain the expert policy $\pi^{default}$ for each environment using the default environmental parameters. Then, to obtain the expert policy $\pi^{new}$ for each environmental variant, we initialize $\pi^{new}$ using $\pi^{default}$ and train with PPO. The specific hyperparameters are provided in table 3. This pretraining strategy speeds up training on new enviroment variant, as opposed to training a new policy from scratch.

### F.3 EXPERT DEMONSTARTIONS

To gather the expert transitions, we execute the expert policies and deterministically select the action with highest logits for discrete action space and select the mean action for continuous action space. In practice, there is a time-out $H$ for all environments even though in theory they are infinite-horizon MDPs. Existing works use various subsampling schemes to generate expert transitions (Ho & Ermon, 2016; Kostrikov et al., 2019; Arora et al., 2020). We performed a simple ablation study on the

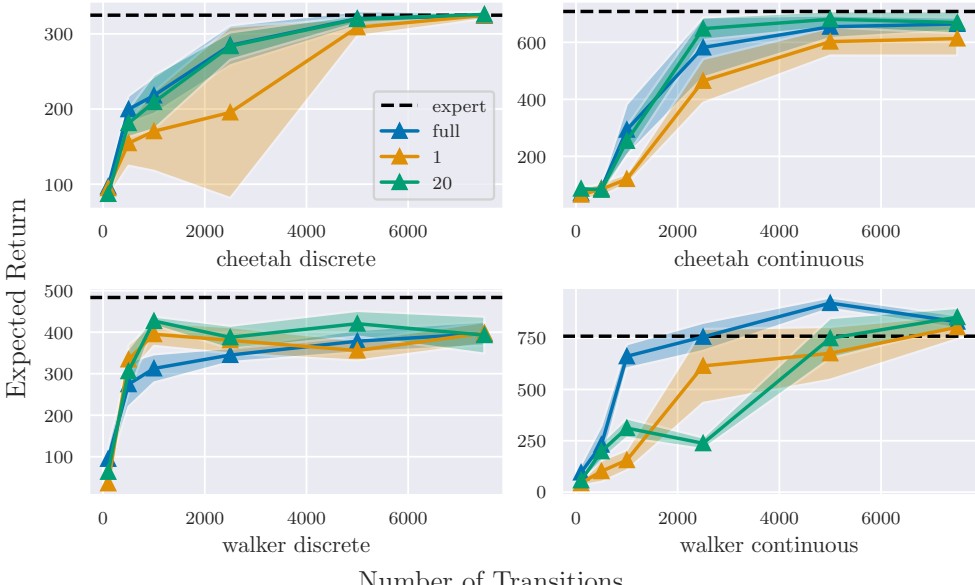

Figure 10: BC policy performance with varying number of transitions. Each curve corresponds to a subsampling length. The solid line corresponds to the mean and the shaded region is 1 standard error from the mean.

pendulum task and the cheetah-run task. Figure 10 shows that while sampling only one transition per episode matches closer to the theory, subsampling can make a significant difference in BC's sample efficiency in practice but with no clear pattern. Conseqeuntly, we gather a total of $\lceil N/H \rceil$ length $H$ time-out episodes and trim the extra $N \mod H$ samples, which is more practical and time efficient, and similar to how practitioners leverage data in real-life applications. We also provide an ablation on the number of expert transitions required to obtain near-expert policy via BC (see figure 11). Table 4 specifies the amount of expert transitions $|\mathcal{D}|$ used in the main experiments.

Table 4: The number of expert transitions $|\mathcal{D}|$ for each environment.

|  | Frozen Lake | Pendulum | Cartpole | Cheetah | Walker |
|---|---|---|---|---|---|
| Discrete | 500 | 1000 | N/A | 1000 | 500 |
| Continuous | N/A | 1000 | 500 | 1000 | 2000 |

## F.4 HYPERPARAMETERS FOR IMITATION LEARNING ALGORITHMS

To fulfill realizability, we ensure that the BC policies have the same architecture as expert policies. The boundedness of each layer can be set such that the norms are at least the expert policy's norms. The policies in cartpole, cheetah, and walker tasks are 2-layered multilayer perceptrons, each with 64 hidden units followed by tanh activation. The policies are trained using the Adam optimizer (Kingma & Ba, 2015). We share the hyperparameters across continuous and discrete action spaces in BC and MTBC and outline them in table 5.

## G EXTRA EXPERIMENTAL RESULTS

**Ablation on Behavioural Cloning Sample Efficiency** We provide an ablation on how behavioural cloning performs based on the number of target task data (see figures 12 and 13). We run each experiment on 10 variants of the environment with 5 random seeds. The returns are normalized based on the corresponding expert policies and random policies.

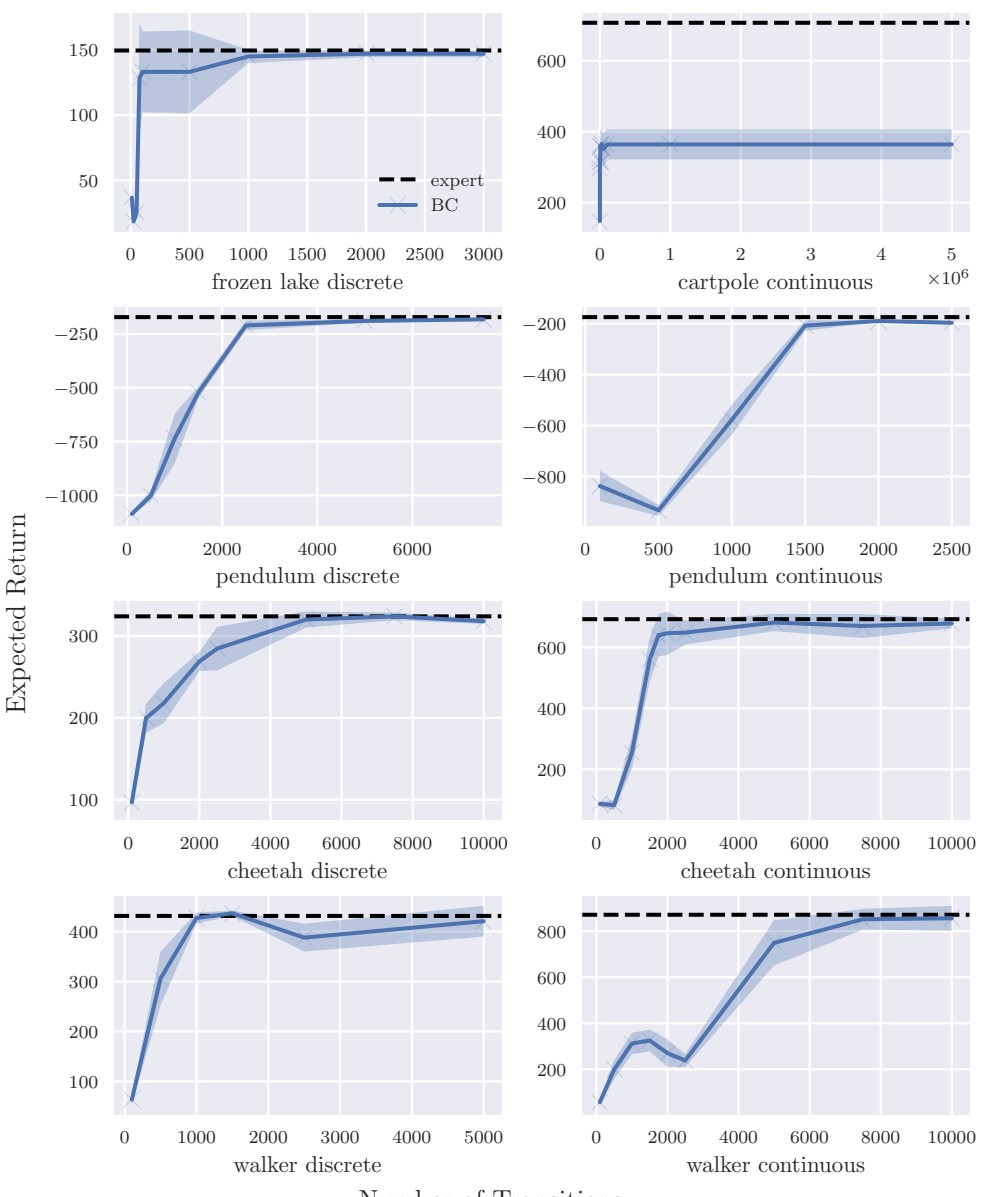

Figure 11: BC policy performance with respect to the number of transitions.

**Inclusion of Target Data during Training Phase** We notice that the imitation performance of MTBC in figures 2 and 3 is generally worse than BC with small number of source tasks. We hypotheize that in the low-source-task regime that the learned representation deviates significantly from the shared representation. To verify this, we conduct an experiment on Cartpole to include the target task during the training phase. Figure 14 shows that including the target task does indeed help with the imitation performance of MTBC, generally matching the imitation performance of BC and improved upon MTBC without the target task for the training phase. Furthermore, as the number of source tasks increases, the imitation performance of MTBC excluding the target task can match MTBC including the target task.

**Impact of Source Tasks and Source Data** Recall that the MTBC agents are trained with $8|\mathcal{D}|$ source data per task on figures 4 and 5. We consequently conduct an ablation on understanding

Table 5: Hyperparameters for BC and MTBC across tasks.

| | BC | MTBC | |
|---|---|---|---|
| **Shared** | | | |
| Updates per epoch | | 100 | |
| Learning rate | | $3 \times 10^{-4}$ | |
| Batch size | | 128 | |
| Phase | N/A | Pretrain | Transfer |
| Number of epochs | 1000 | 5000 | 1000 |
| L2 regularization | None | $2 \times 10^{-4}$ | None |

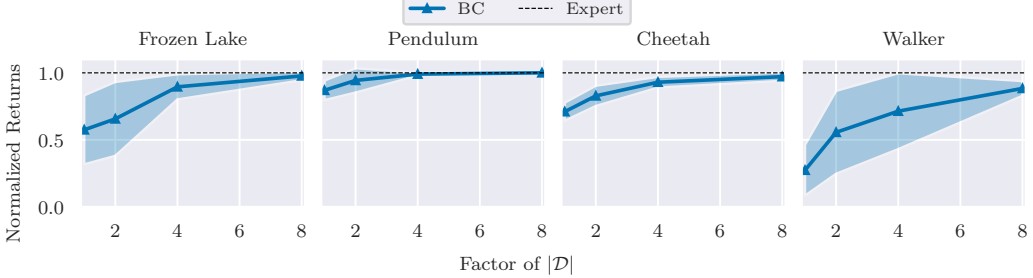

Figure 12: The performance of BC generally improves with increasing target data in discrete tasks. The solid line corresponds to the mean and the shaded region is 1 standard error from the mean.

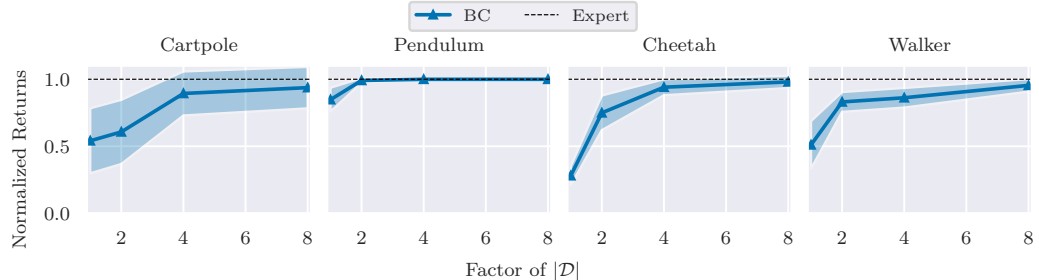

Figure 13: The performance of BC generally improves with increasing target data in continuous tasks. The solid line corresponds to the mean and the shaded region is 1 standard error from the mean.

what happens when the agents are trained with $|\mathcal{D}|$ source data per task instead. Figures 15 and 16 demonstrate that when $T$ is small, MTBC with larger $N$ significantly outperforms smaller $N$.

We also provide per-environment plots for further analysis on how MTBC performs with varying number of source tasks $T$ and source data $N$ (see figures 17–24.)

**Impact of Target Data**  We provide per-environment plots for further analysis on how MTBC performs with varying number of source tasks $T$ and target tasks $M$ (see figures 25–32.)

**Task Diversity**  We include two extra task-diversity metrics on the discrete tasks in table 6.

*(i)* Rather than using approximate KL-divergence, we instead use approximate Bhattacharyya distance (**Bhattacharyya**):

$$\hat{\sigma} = \frac{M \sum_{t=1}^{T} \sum_{n=1}^{N} \left[ D_B(\pi_t^*(s_{nt}) \| \text{softmax}(\hat{f}_t \circ \hat{\phi})(s_{nt})) \right]}{NT \min_{t \in [T]} \sum_{m=1}^{M} \left[ D_B(\pi_\tau^*(s_{m,\tau}) \| \text{softmax}(\hat{f}_t \circ \hat{\phi})(s_{m,\tau})) \right]},$$

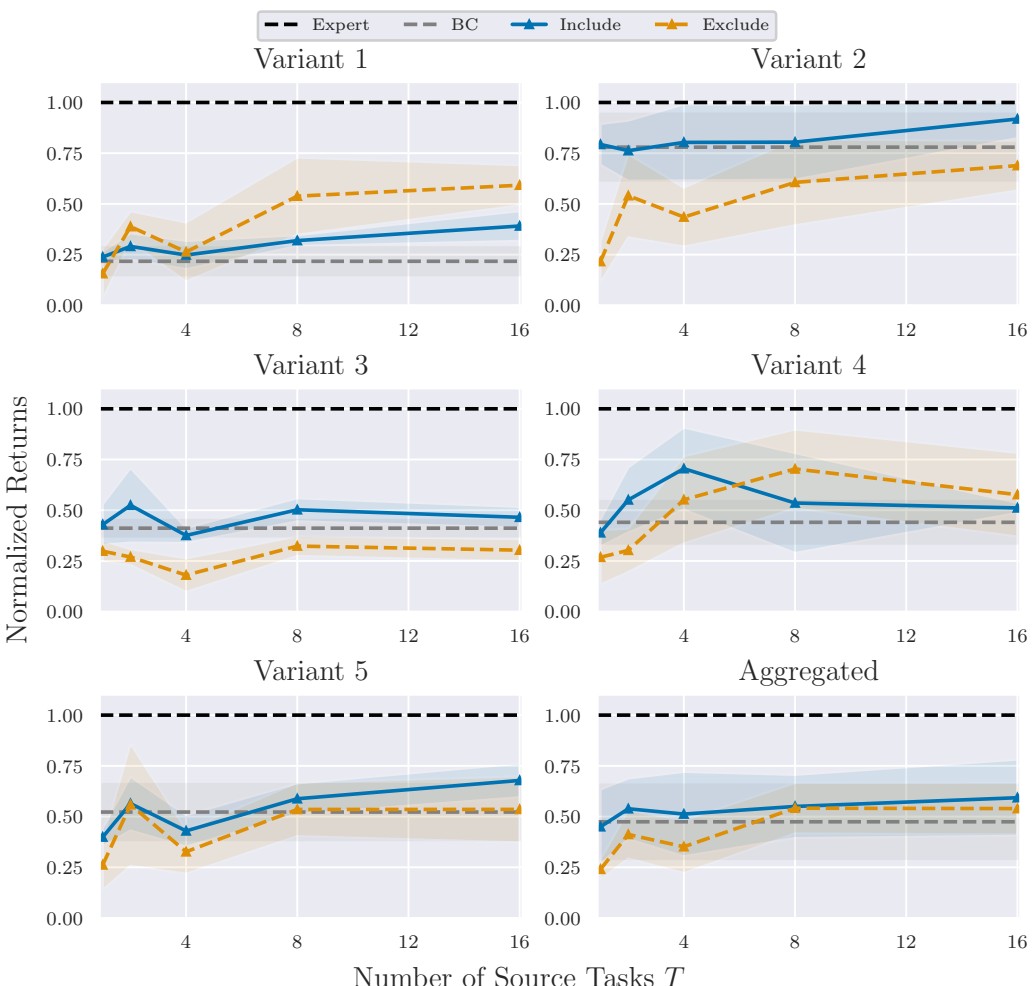

Figure 14: The performance of MTBC on Cartpole when including the target task as one of the source tasks during training. Generally, including the target task improves the imitation performance of MTBC, matching the imitation performance of BC and improved upon MTBC without the target task for the training phase. As $T$ increases, MTBC without the target task can match MTBC with the target task.

where $D_B(P\|Q) = -\ln\left(\sum_{x\in\mathcal{X}}\sqrt{P(x)Q(x)}\right)$ is the Bhattacharyya distance. While this metric can be approximated using samples similar to Approx. KL, we note that the Bhattacharyya distance itself is symmetric.

*(ii)* We also consider the $\ell_2$-norm of between the expert policy representation $\phi^E$ and the MTBC policy representation $\hat{\phi}$ (**L2 Repr. Norm**):

$$\hat{\sigma} = \frac{1}{\|\phi^E - \hat{\phi}\|_2}.$$

Using the representation difference (Khodak et al., 2019) has been analyzed previously in the meta-learning literature. However, this metric requires access to the true representation of the expert policy.

Table 6 indicates that Bhattacharyya is generally weaker than Approx. KL, furthermore in the frozen lake environment Bhattacharyya negatively correlates to the imitation performance. On the other hand, L2 Repr. Norm positively correlates with the imitation performance, mostly stronger than Approx. KL. However, we again emphasize that L2 Repr. Norm requires access to the representation of the expert policy which may be infeasible to obtain in practice.

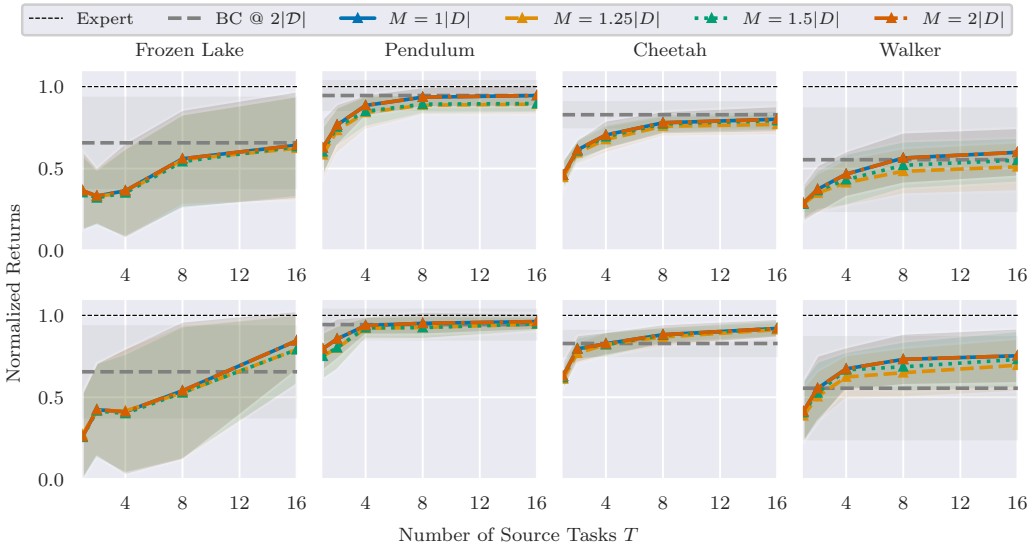

Figure 15: The performance of MTBC on with varying number of source data $N$ on discrete tasks. **Top:** MTBC is trained with $N = |\mathcal{D}|$. **Bottom:** MTBC is trained with $N = |8\mathcal{D}|$. Generally, larger $N$ corresponds to better imitation performance. Without smaller number of source tasks, the improvement is more significant.

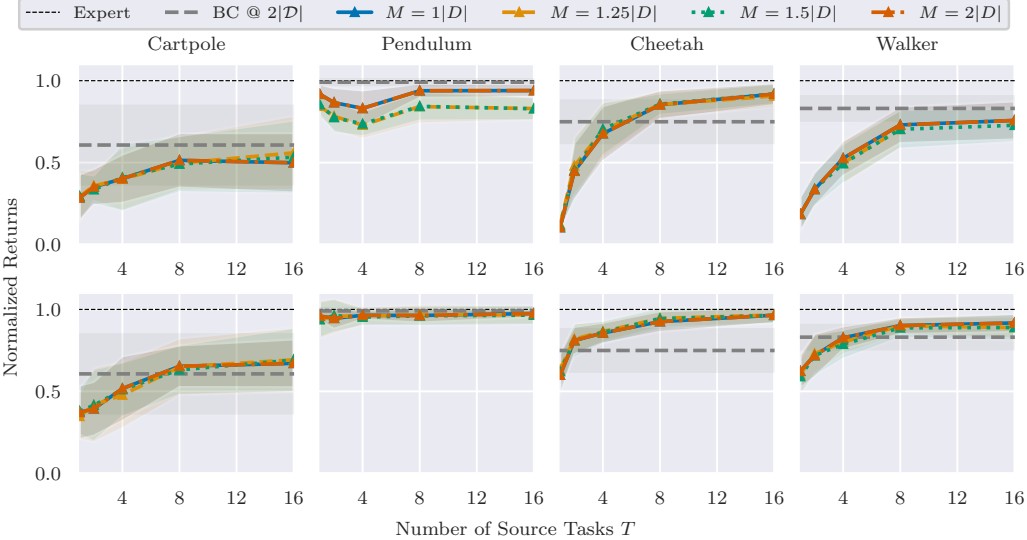

Figure 16: The performance of MTBC on with varying number of source data on continuous tasks. **Top:** MTBC is trained with $N = |\mathcal{D}|$. **Bottom:** MTBC is trained with $N = |8\mathcal{D}|$. Generally, larger $N$ corresponds to better imitation performance. As $T$ decreases, the improvement is more significant.

We provide detailed plots on the correlations between various diversities and normalized returns (see figures 33–40).

Table 6: The correlations between various task diversity metrics and the normalized returns in discrete tasks. Bolded text means highest mean.

|  | Frozen Lake | Pendulum | Cheetah | Walker |
|---|---|---|---|---|
| **Pearson** | | | | |
| L2 | $-0.008 \pm 0.023$ | $0.059 \pm 0.069$ | $-0.008 \pm 0.080$ | $-0.016 \pm 0.096$ |
| Data Perf. | $0.035 \pm 0.223$ | $-0.027 \pm 0.023$ | $-0.057 \pm 0.041$ | $0.279 \pm 0.128$ |
| Bhattacharyya | $-0.358 \pm 0.098$ | $0.071 \pm 0.013$ | $0.150 \pm 0.029$ | $-0.041 \pm 0.011$ |
| L2 Repr. Norm | $0.169 \pm 0.040$ | $\mathbf{0.190 \pm 0.034}$ | $0.381 \pm 0.060$ | $0.048 \pm 0.015$ |
| Approx. KL (Ours) | $\mathbf{0.206 \pm 0.086}$ | $-0.117 \pm 0.023$ | $\mathbf{0.394 \pm 0.065}$ | $\mathbf{0.397 \pm 0.071}$ |
| **Spearman** | | | | |
| L2 | $-0.019 \pm 0.010$ | $-0.064 \pm 0.032$ | $0.091 \pm 0.024$ | $-0.008 \pm 0.155$ |
| Data Perf. | $0.035 \pm 0.028$ | $0.116 \pm 0.026$ | $0.053 \pm 0.071$ | $-0.159 \pm 0.083$ |
| Bhattacharyya | $-0.138 \pm 0.046$ | $0.079 \pm 0.026$ | $0.249 \pm 0.037$ | $0.045 \pm 0.014$ |
| L2 Repr. Norm | $0.002 \pm 0.010$ | $\mathbf{0.288 \pm 0.048}$ | $\mathbf{0.404 \pm 0.064}$ | $0.125 \pm 0.027$ |
| Approx. KL (Ours) | $\mathbf{0.163 \pm 0.036}$ | $0.168 \pm 0.031$ | $0.369 \pm 0.059$ | $\mathbf{0.228 \pm 0.038}$ |
| **Kendall** | | | | |
| L2 | $0.111 \pm 0.022$ | $0.052 \pm 0.105$ | $-0.082 \pm 0.059$ | $0.010 \pm 0.044$ |
| Data Perf. | $-0.033 \pm 0.006$ | $0.137 \pm 0.068$ | $0.046 \pm 0.023$ | $-0.355 \pm 0.053$ |
| Bhattacharyya | $-0.203 \pm 0.030$ | $0.136 \pm 0.043$ | $0.422 \pm 0.064$ | $0.022 \pm 0.026$ |
| L2 Repr. Norm | $\mathbf{0.206 \pm 0.033}$ | $\mathbf{0.185 \pm 0.031}$ | $\mathbf{0.490 \pm 0.070}$ | $0.168 \pm 0.028$ |
| Approx. KL (Ours) | $0.038 \pm 0.037$ | $0.050 \pm 0.020$ | $0.474 \pm 0.070$ | $\mathbf{0.222 \pm 0.040}$ |

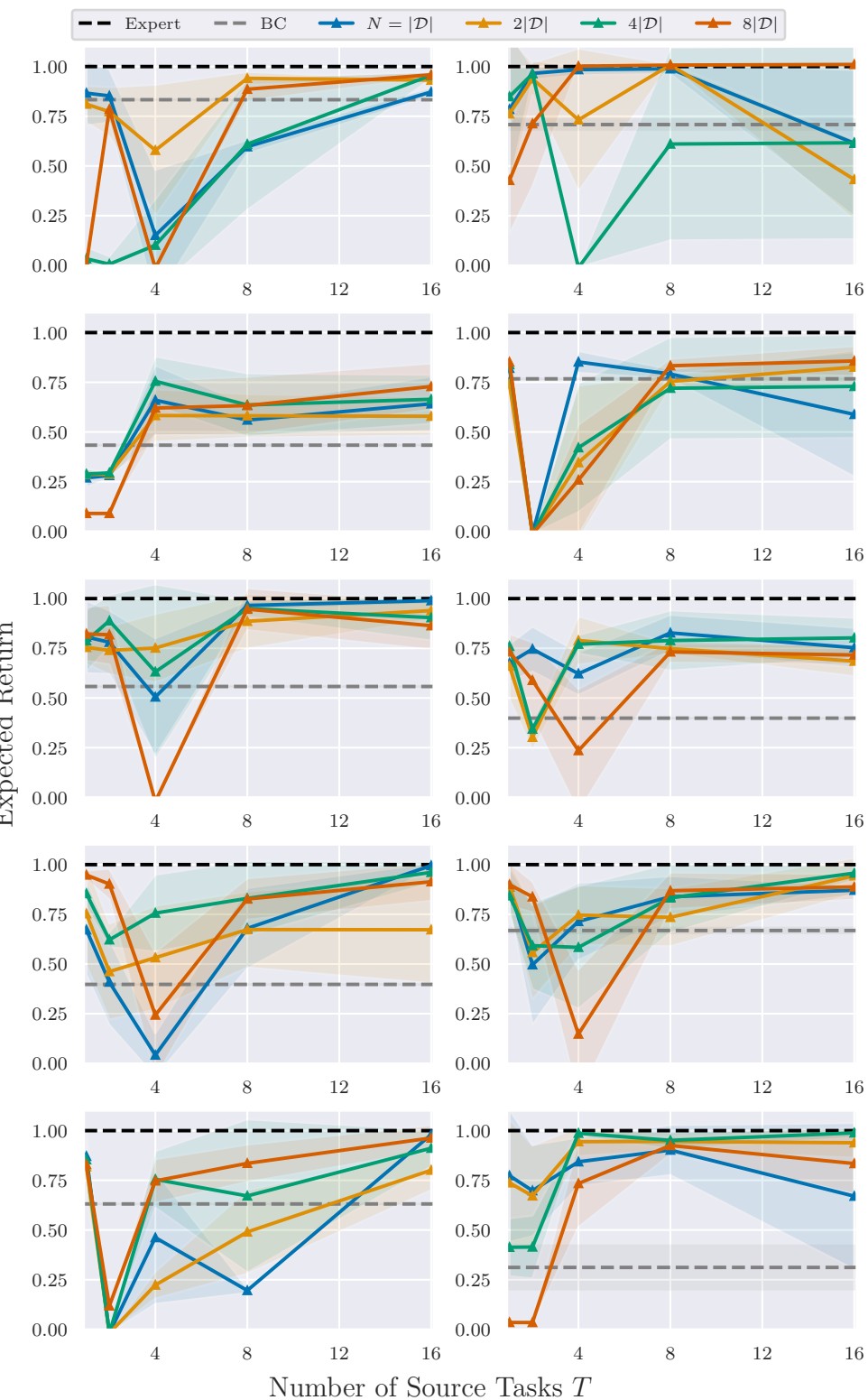

Figure 17: MTBC performance as we vary $N$ and $T$ in the frozen lake task. Each solid line colour corresponds to a particular $N$. The solid line corresponds to the mean and the shaded region is 1 standard error from the mean.

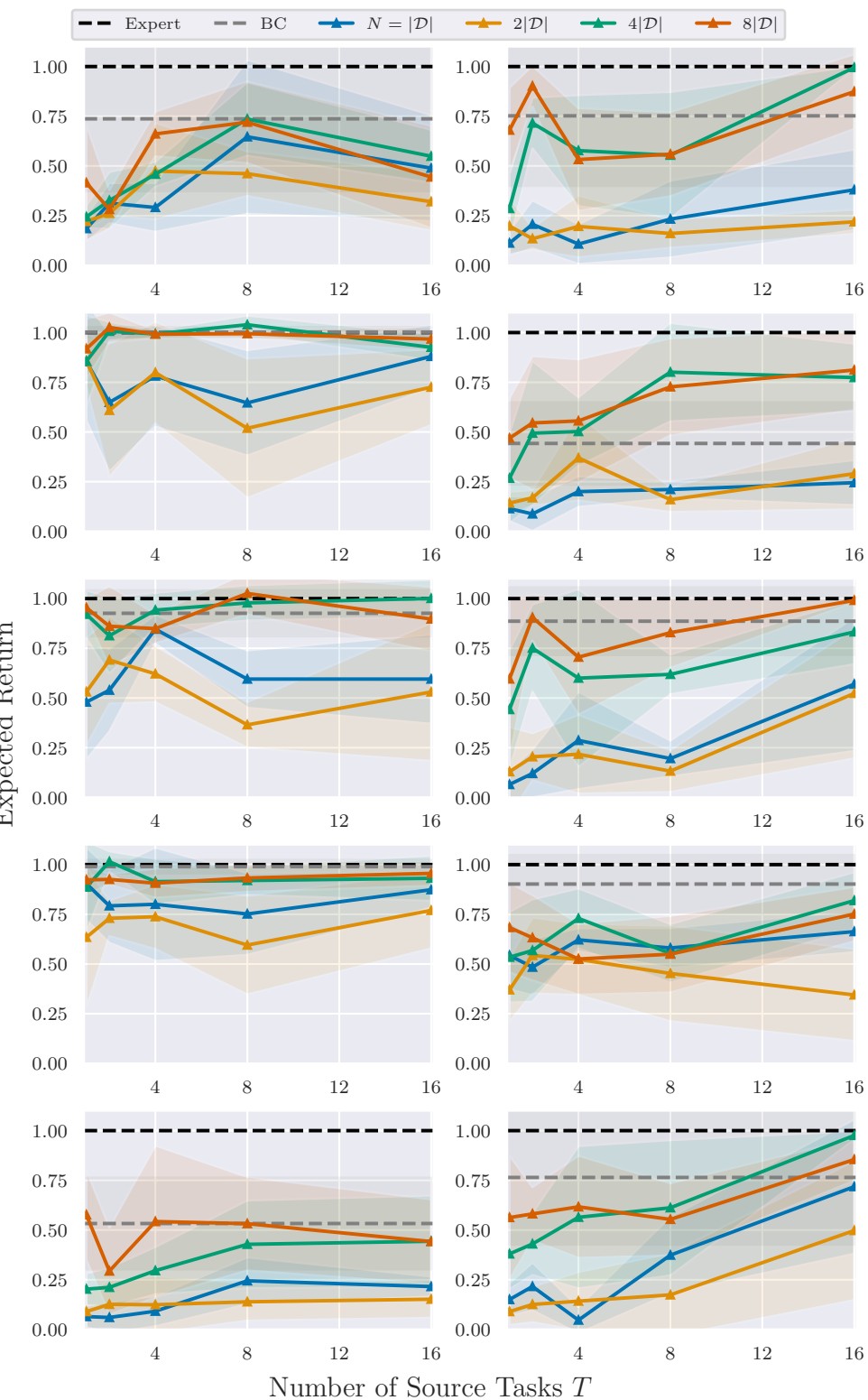

Figure 18: MTBC performance as we vary $N$ and $T$ in the cartpole task. Each solid line colour corresponds to a particular $N$. The solid line corresponds to the mean and the shaded region is 1 standard error from the mean.

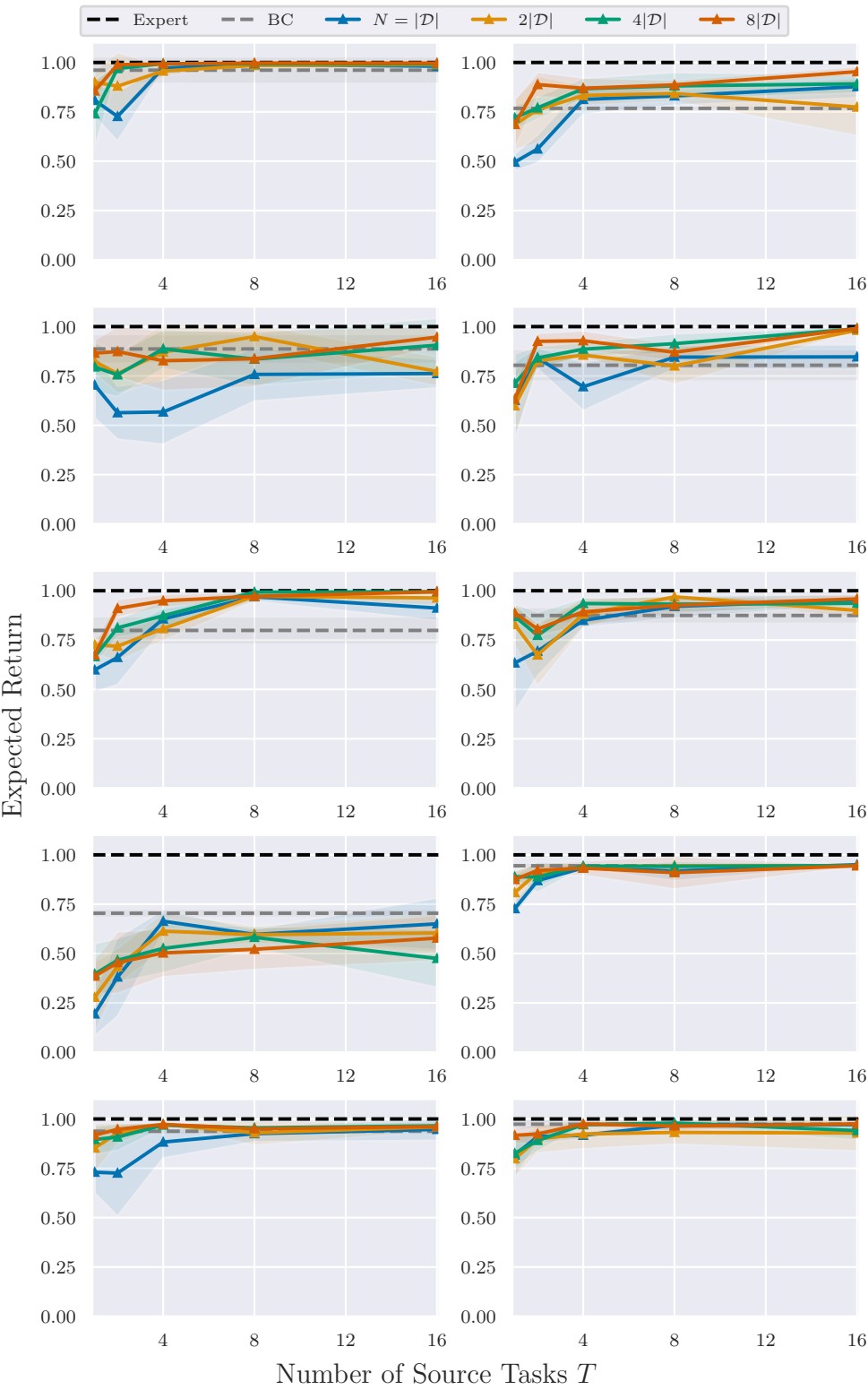

Figure 19: MTBC performance as we vary $N$ and $T$ in the discrete pendulum task. Each solid line colour corresponds to a particular $N$. The solid line corresponds to the mean and the shaded region is 1 standard error from the mean.

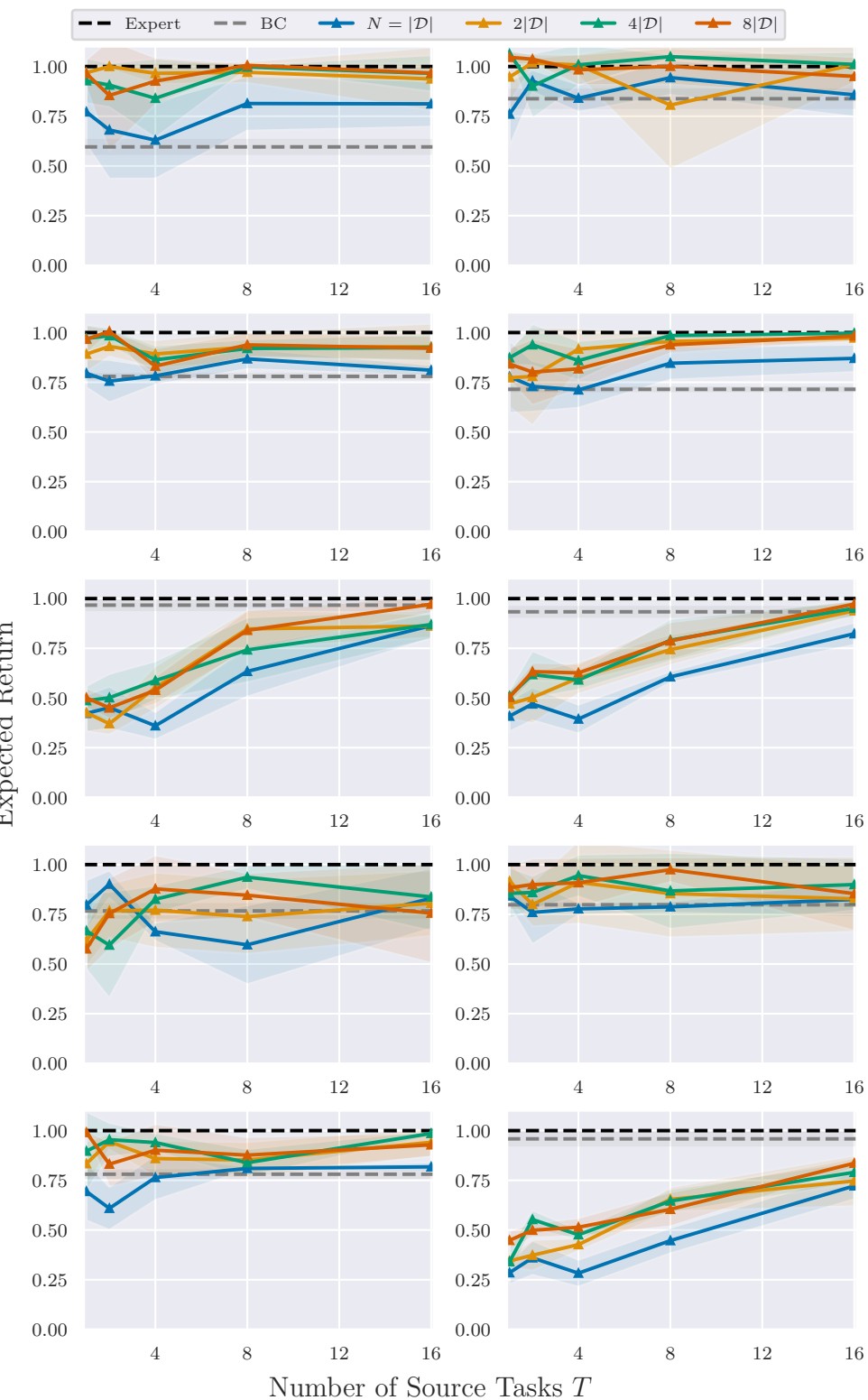

Figure 20: MTBC performance as we vary $N$ and $T$ in the continuous pendulum task. Each solid line colour corresponds to a particular $N$. The solid line corresponds to the mean and the shaded region is 1 standard error from the mean.

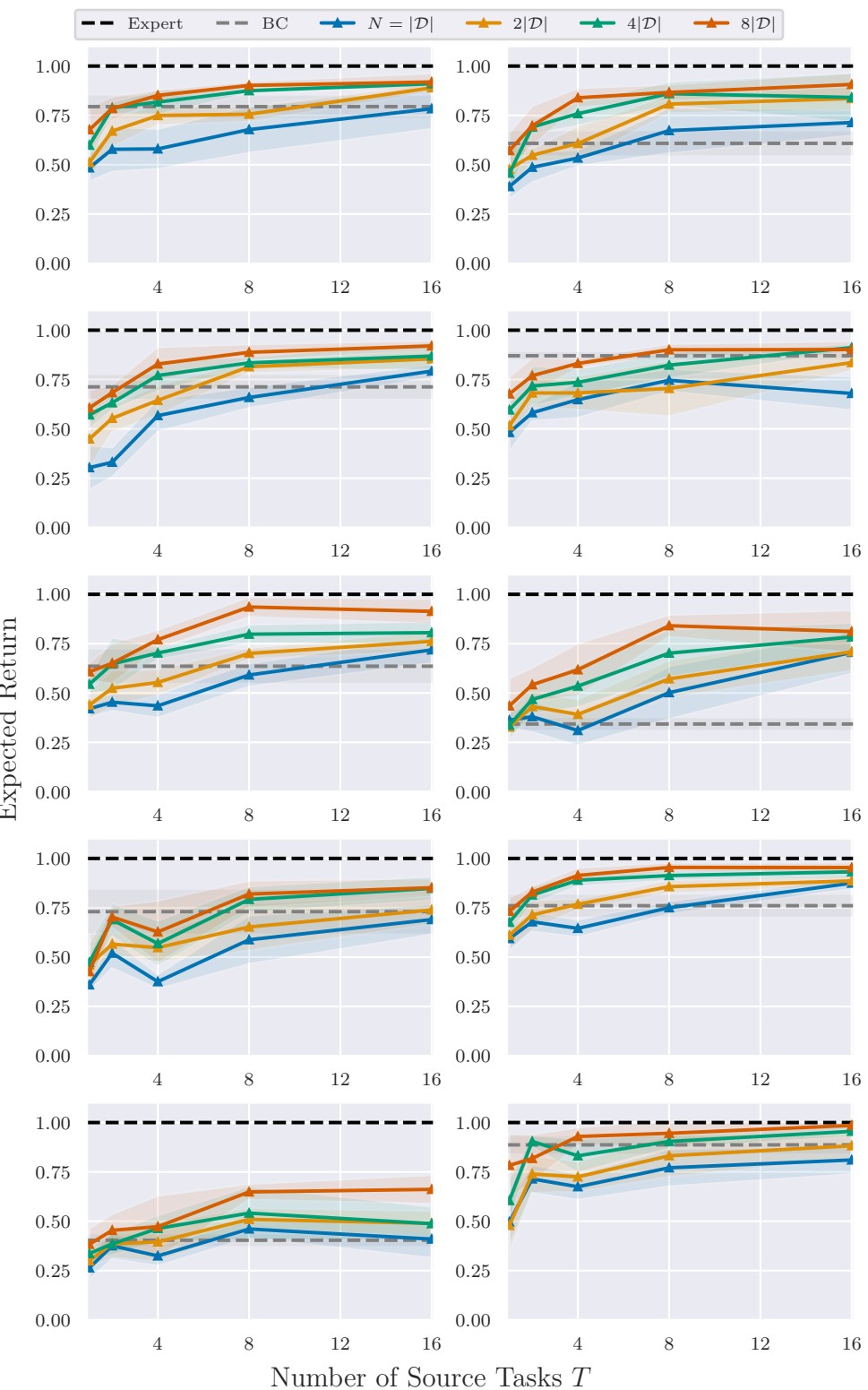

Figure 21: MTBC performance as we vary $N$ and $T$ in the discrete cheetah task. Each solid line colour corresponds to a particular $N$. The solid line corresponds to the mean and the shaded region is 1 standard error from the mean.

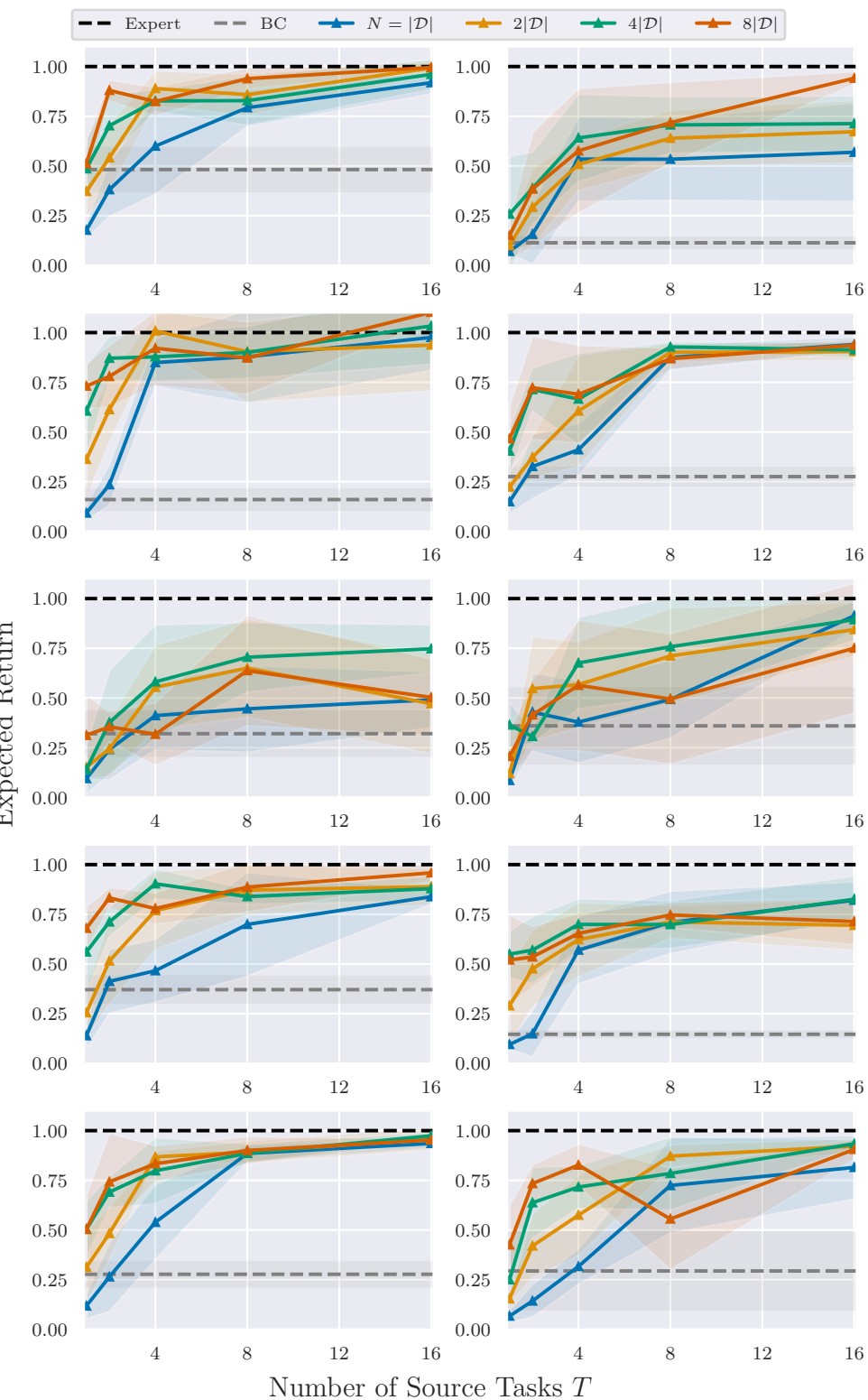

Figure 22: MTBC performance as we vary $N$ and $T$ in the continuous cheetah task. Each solid line colour corresponds to a particular $N$. The solid line corresponds to the mean and the shaded region is 1 standard error from the mean.

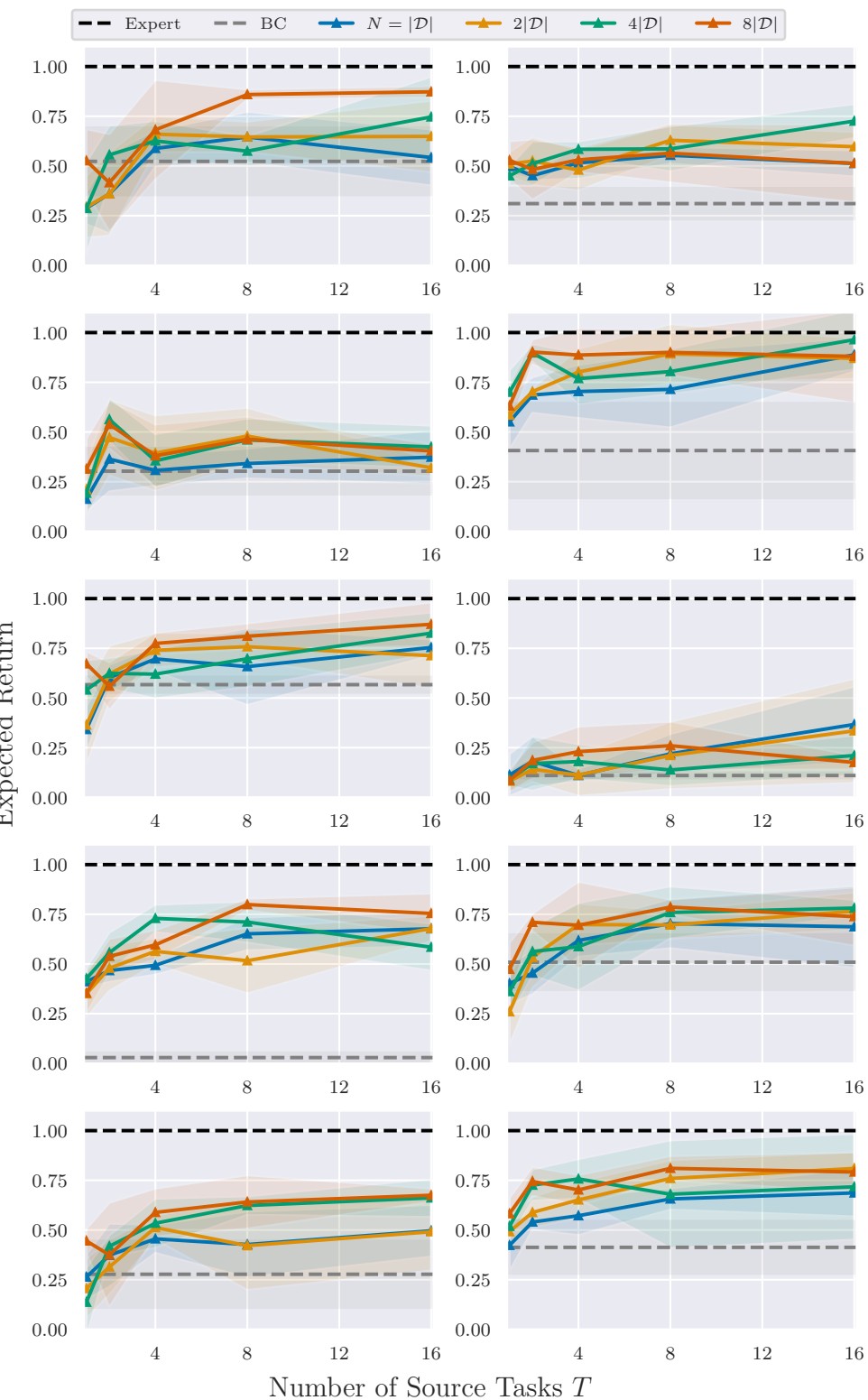

Figure 23: MTBC performance as we vary $N$ and $T$ in the discrete walker task. Each solid line colour corresponds to a particular $N$. The solid line corresponds to the mean and the shaded region is 1 standard error from the mean.

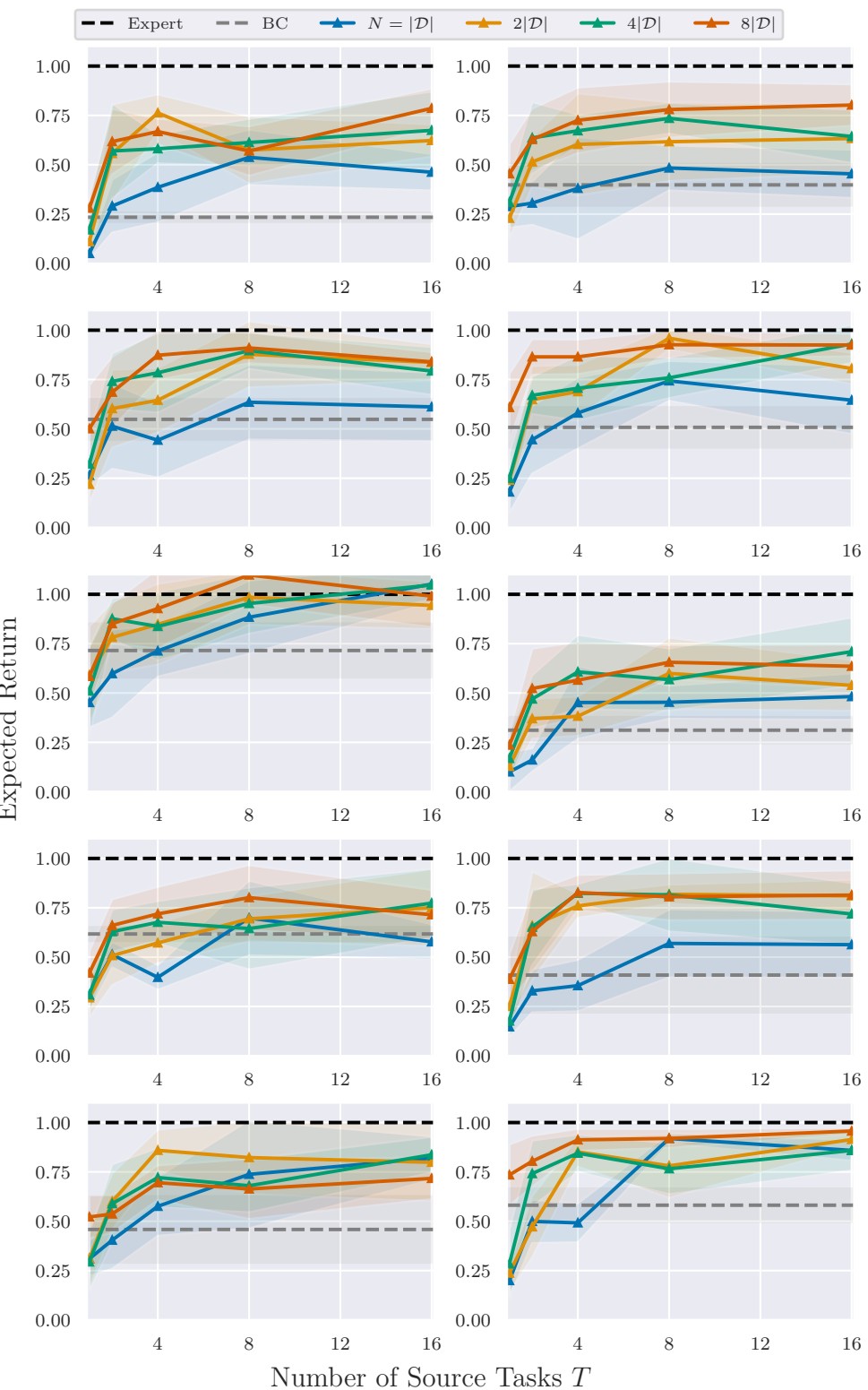

Figure 24: MTBC performance as we vary $N$ and $T$ in the continuous walker task. Each solid line colour corresponds to a particular $N$. The solid line corresponds to the mean and the shaded region is 1 standard error from the mean.

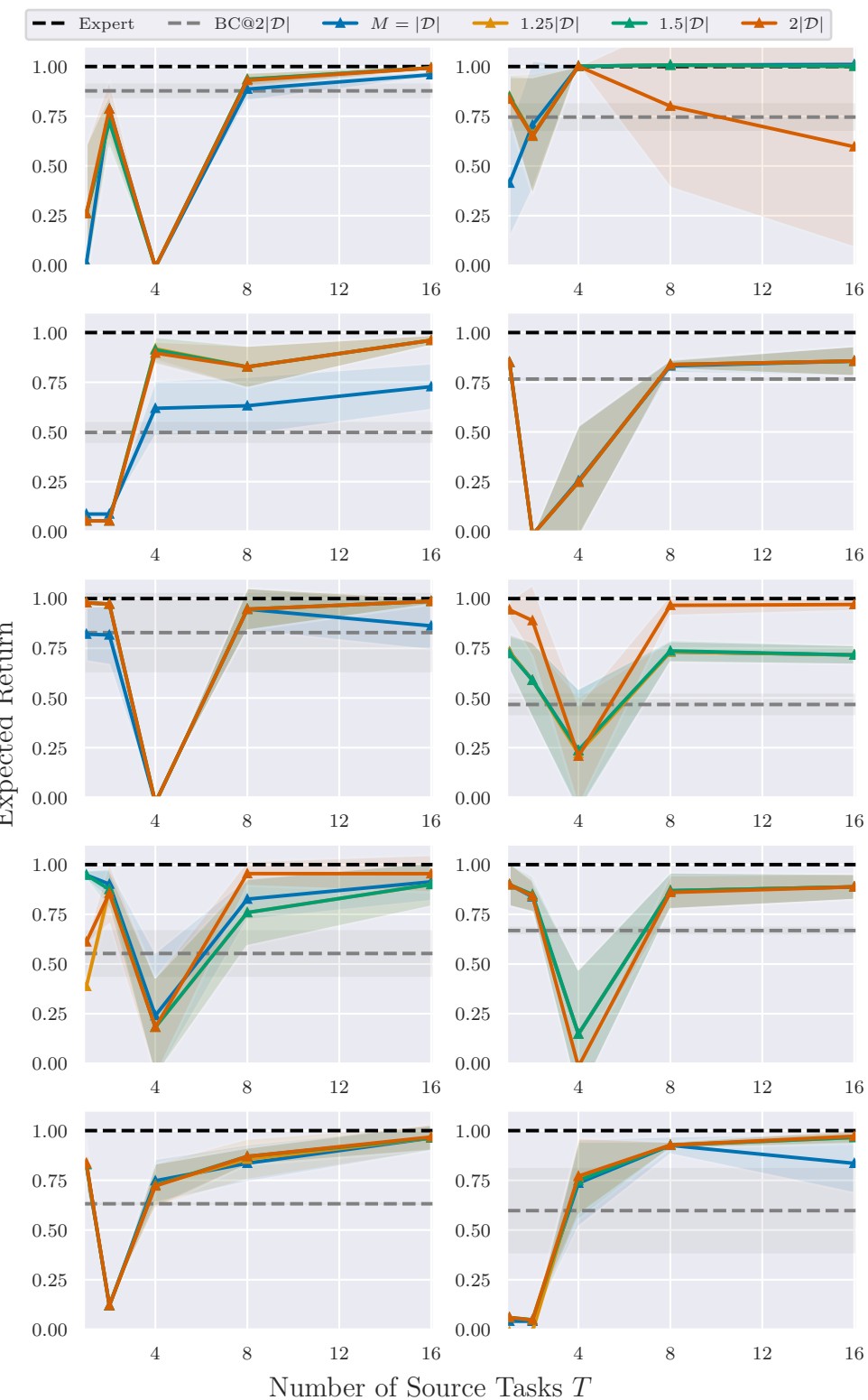

Figure 25: MTBC performance as we vary $M$ and $T$ in the frozen lake task. Each solid line colour corresponds to a particular $M$. The solid line corresponds to the mean and the shaded region is 1 standard error from the mean.

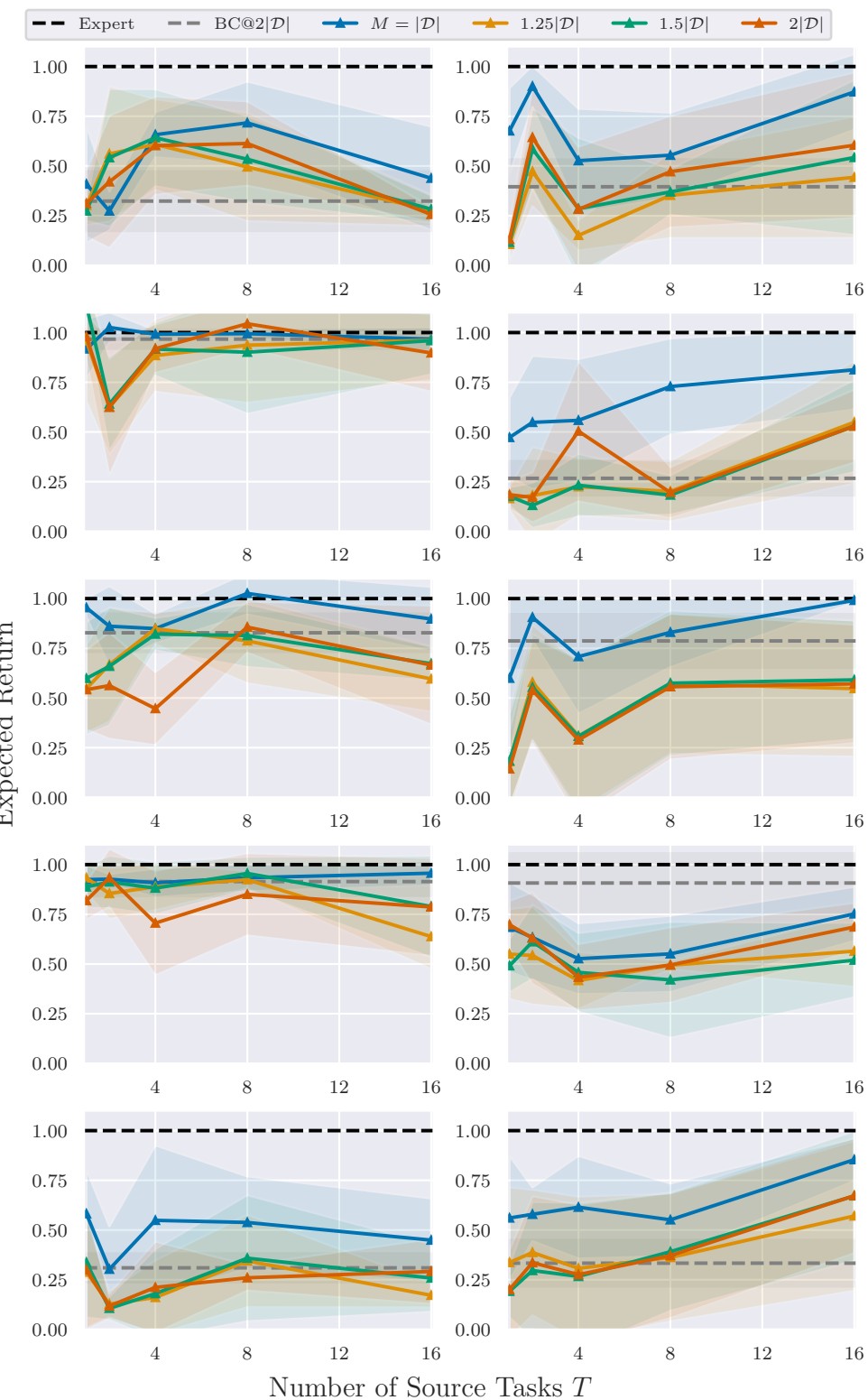

Figure 26: MTBC performance as we vary $M$ and $T$ in the cartpole task. Each solid line colour corresponds to a particular $M$. The solid line corresponds to the mean and the shaded region is 1 standard error from the mean.

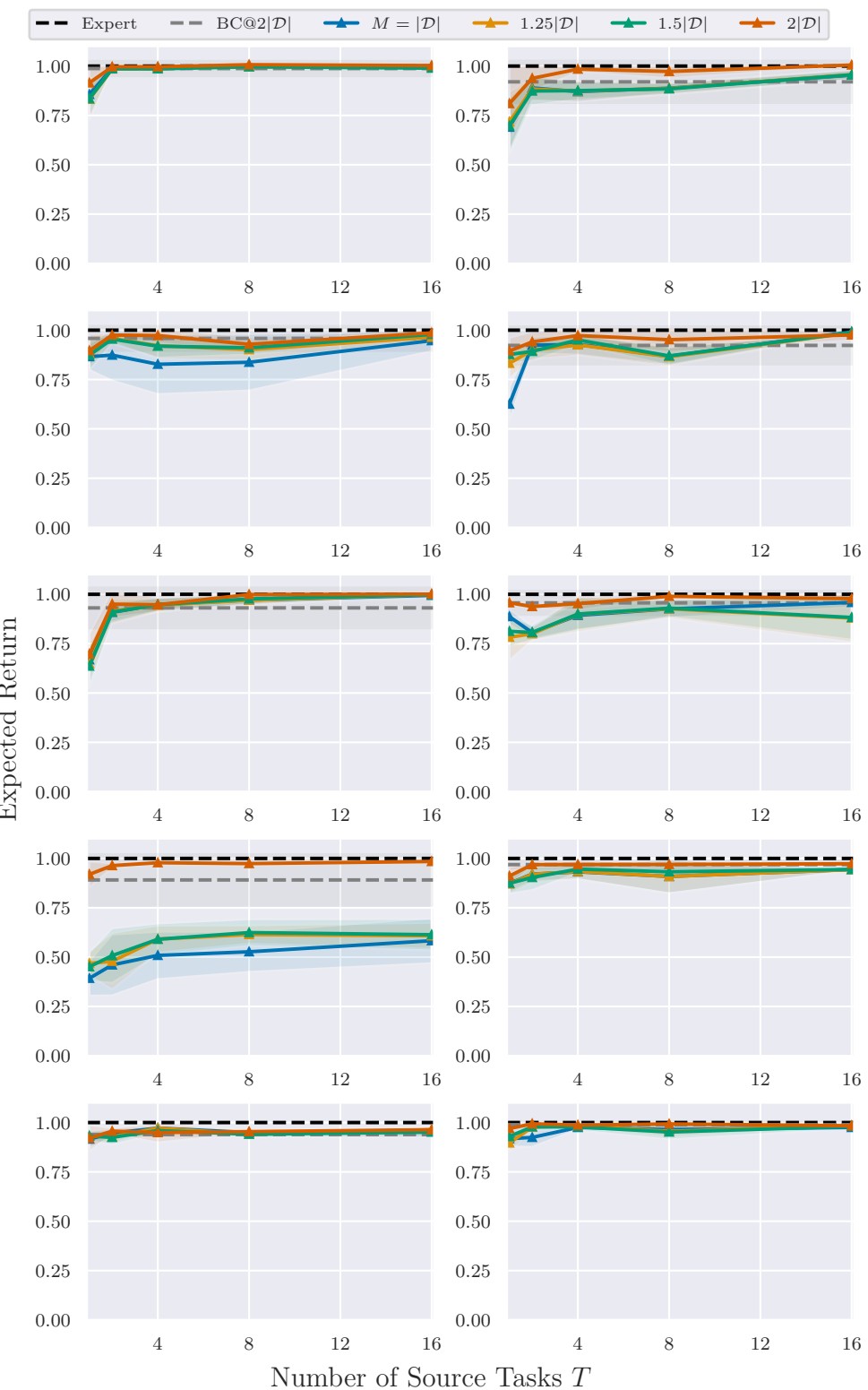

Figure 27: MTBC performance as we vary $M$ and $T$ in the discrete pendulum task. Each solid line colour corresponds to a particular $M$. The solid line corresponds to the mean and the shaded region is 1 standard error from the mean.

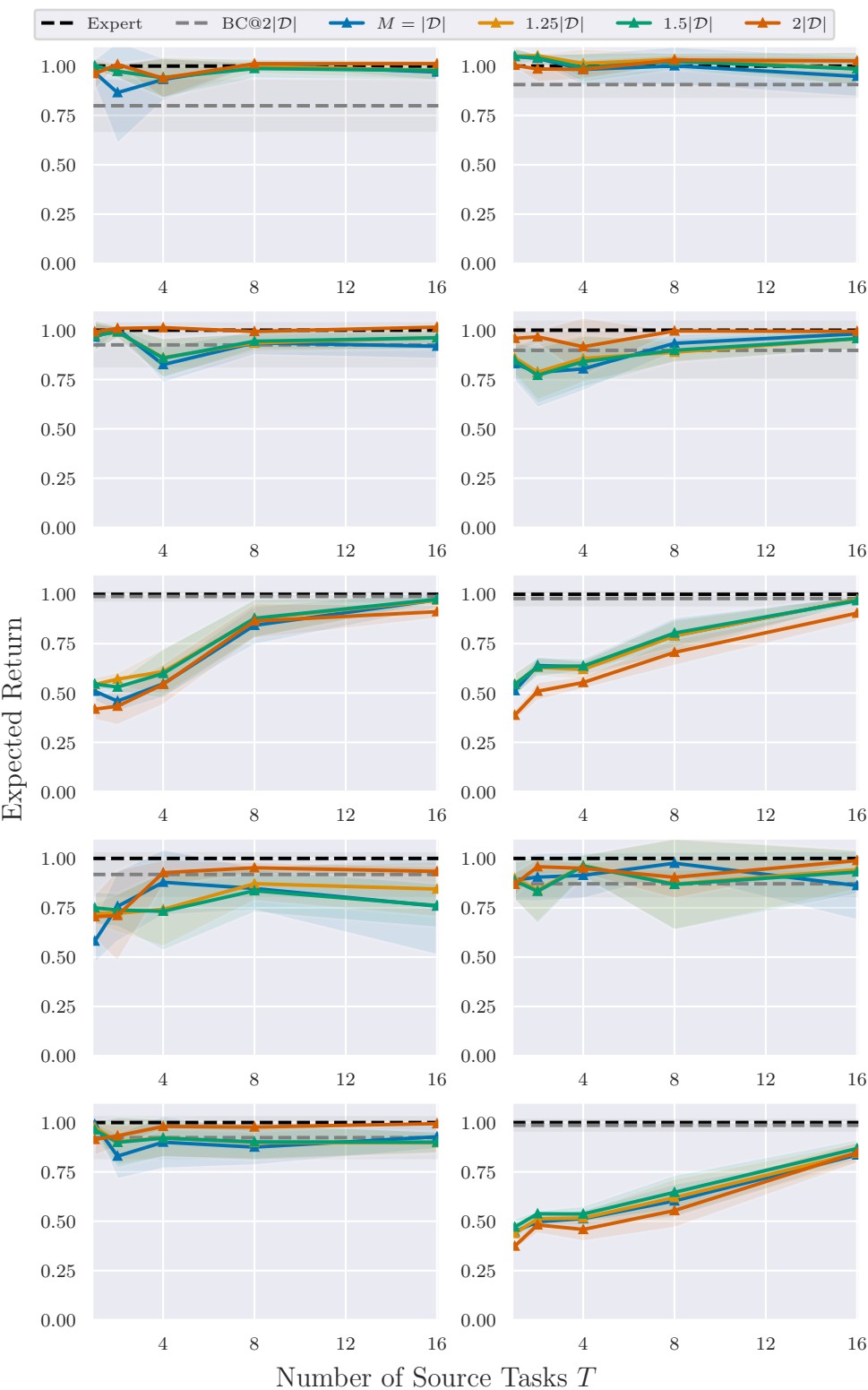

Figure 28: MTBC performance as we vary $M$ and $T$ in the continuous pendulum task. Each solid line colour corresponds to a particular $M$. The solid line corresponds to the mean and the shaded region is 1 standard error from the mean.

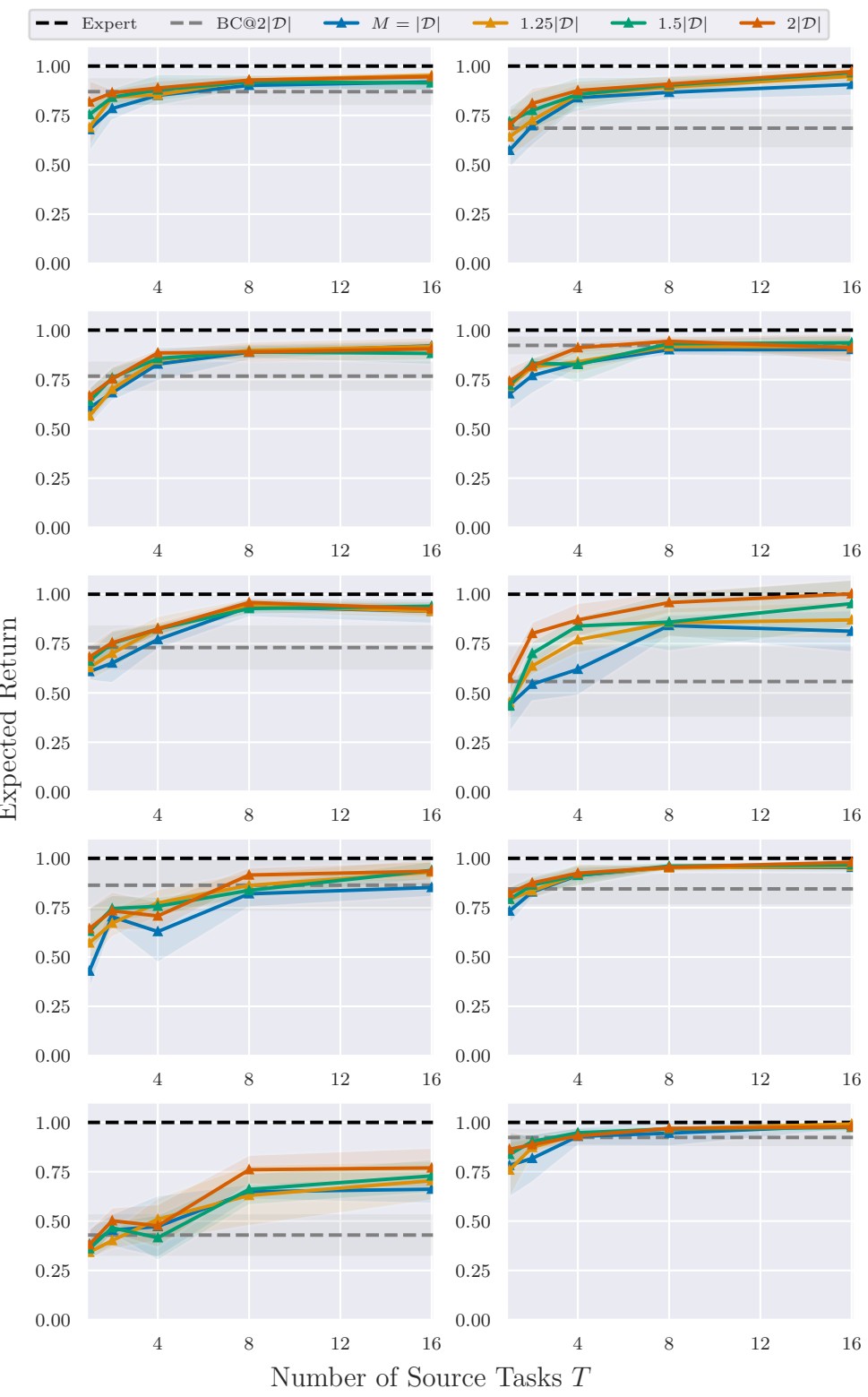

Figure 29: MTBC performance as we vary $M$ and $T$ in the discrete cheetah task. Each solid line colour corresponds to a particular $M$. The solid line corresponds to the mean and the shaded region is 1 standard error from the mean.

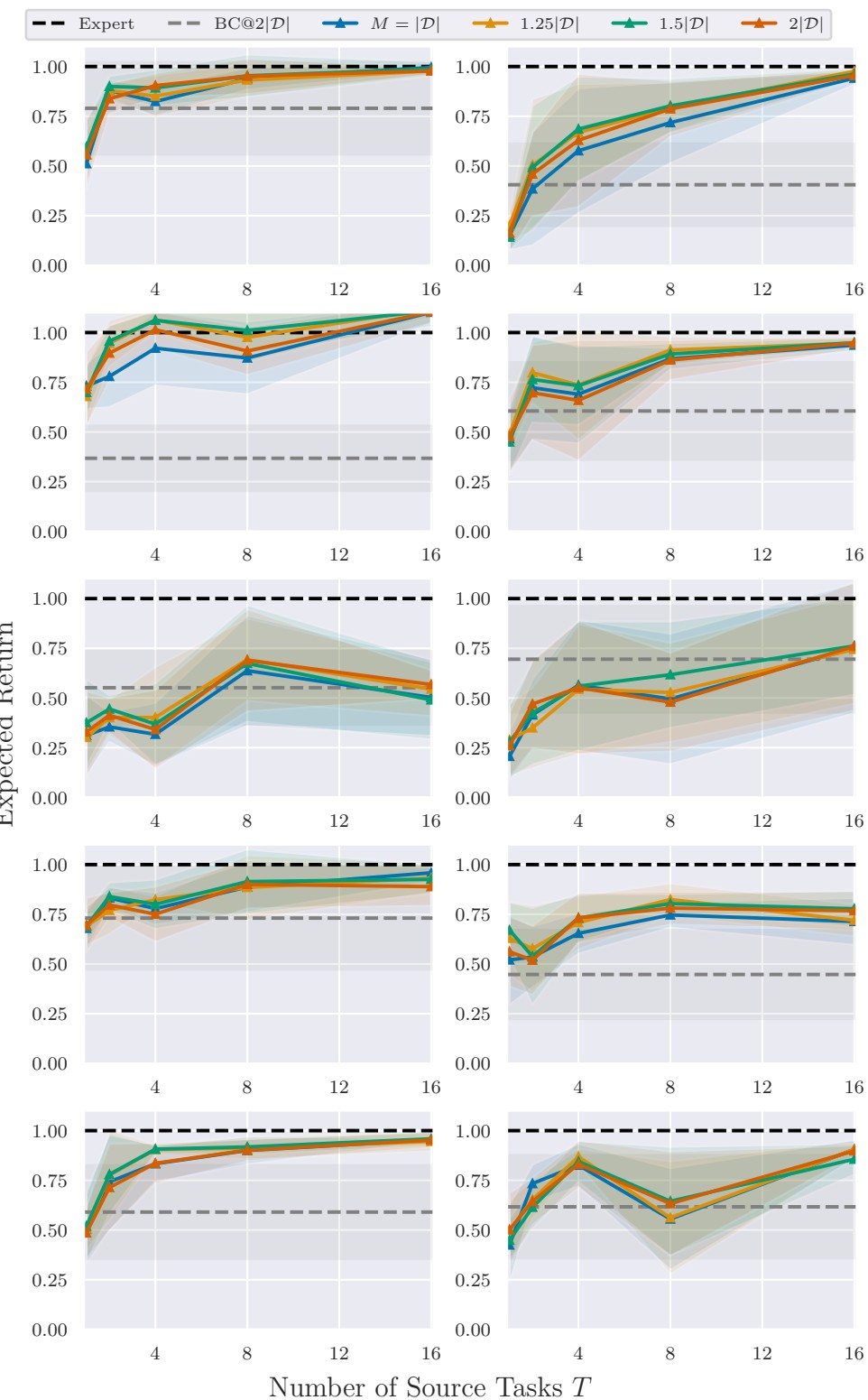

Figure 30: MTBC performance as we vary $M$ and $T$ in the continuous cheetah task. Each solid line colour corresponds to a particular $M$. The solid line corresponds to the mean and the shaded region is 1 standard error from the mean.

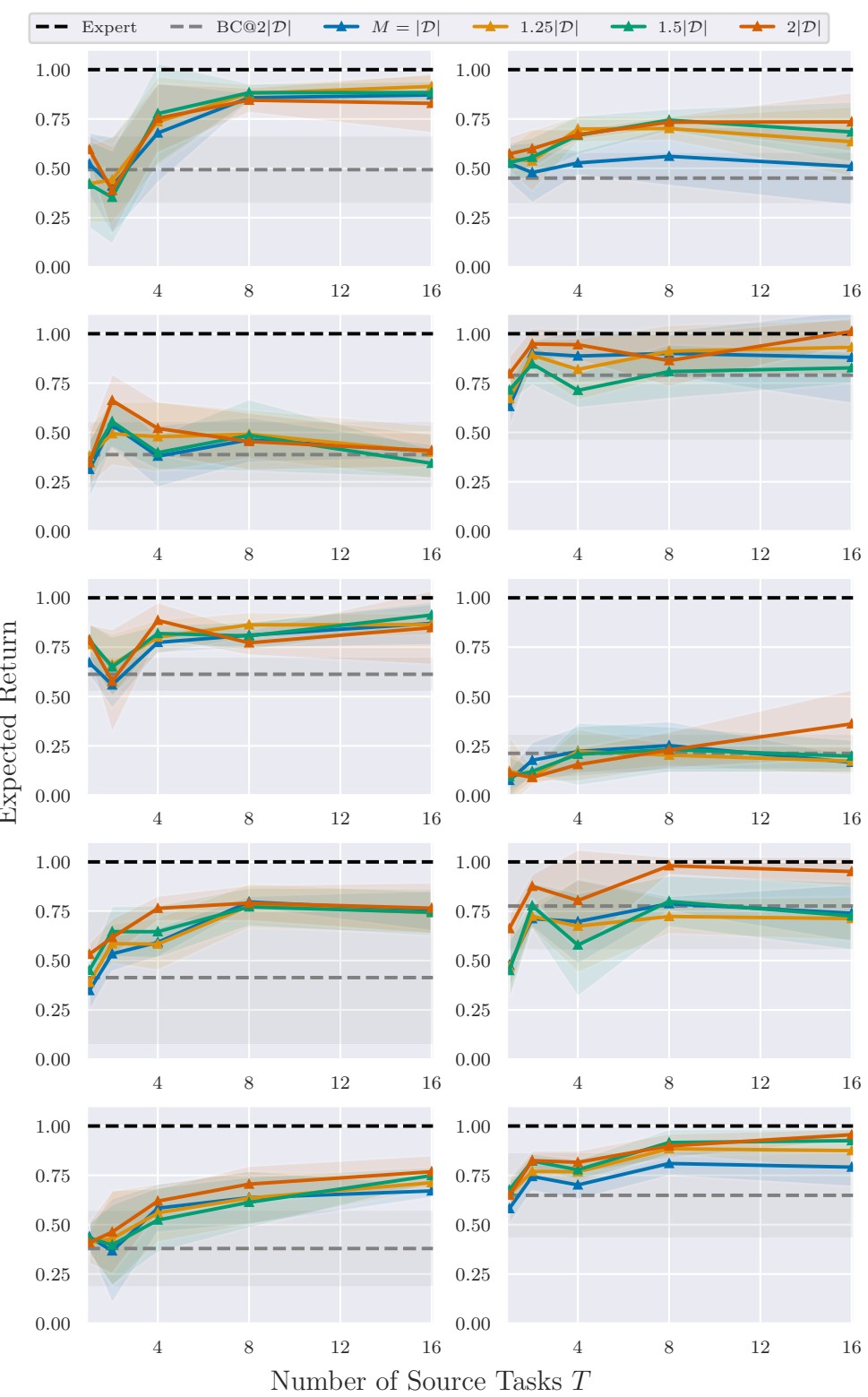

Figure 31: MTBC performance as we vary $M$ and $T$ in the discrete walker task. Each solid line colour corresponds to a particular $M$. The solid line corresponds to the mean and the shaded region is 1 standard error from the mean.

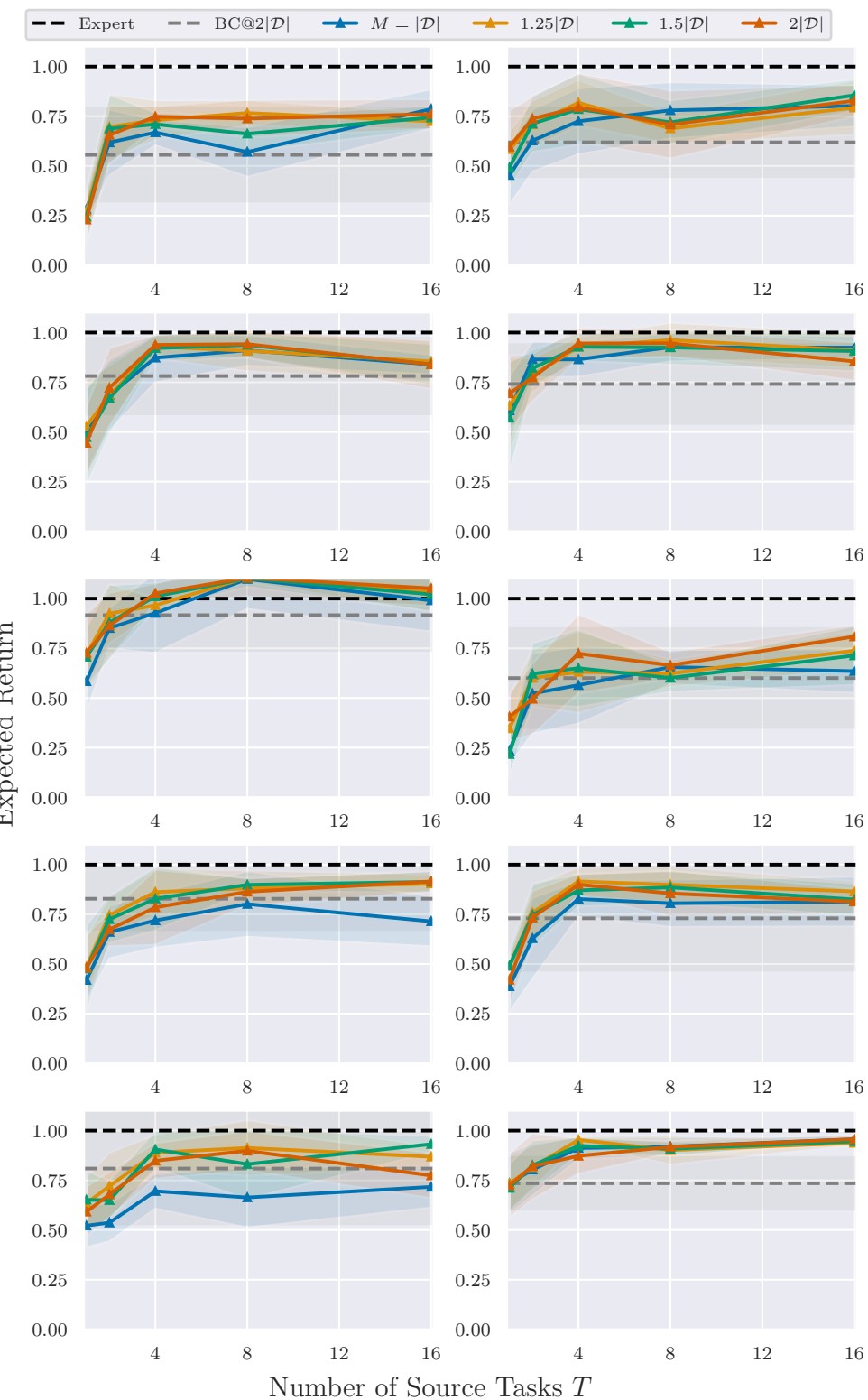

Figure 32: MTBC performance as we vary $M$ and $T$ in the continuous walker task. Each solid line colour corresponds to a particular $M$. The solid line corresponds to the mean and the shaded region is 1 standard error from the mean.

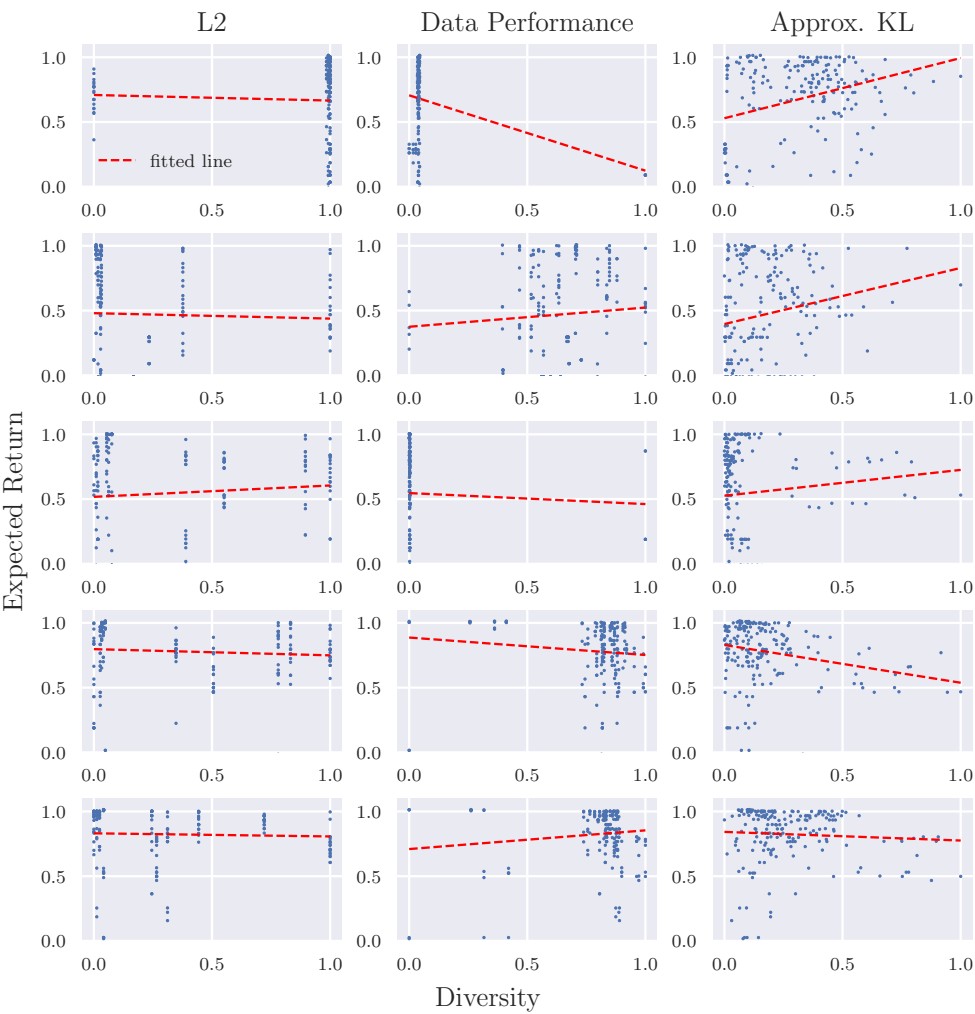

Figure 33: MTBC performance with respect to various diversity metrics in the frozen lake task. Each row corresponds to number of source tasks in $\{1, 2, 4, 8, 16\}$. x-axis is normalized to be in $[0, 1]$.

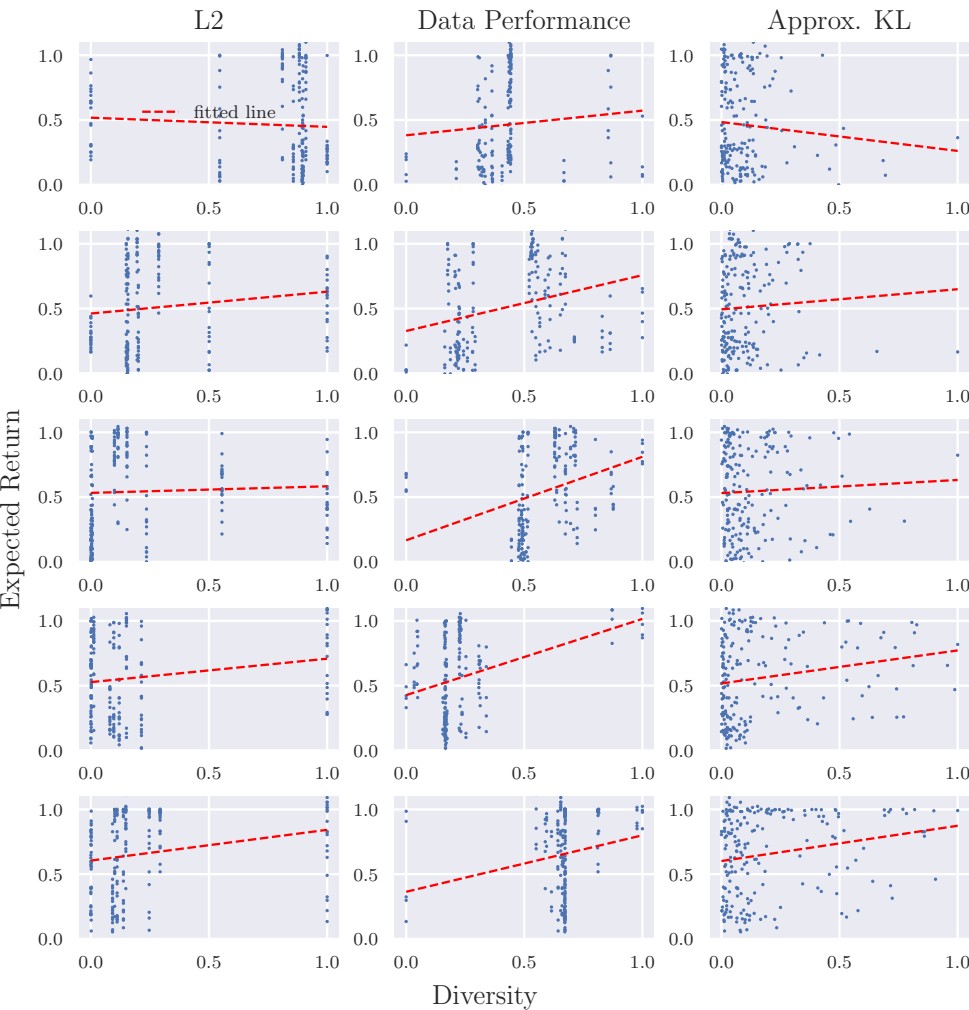

Figure 34: MTBC performance with respect to various diversity metrics in the cartpole task. Each row corresponds to number of source tasks in $\{1, 2, 4, 8, 16\}$. x-axis is normalized to be in $[0, 1]$.

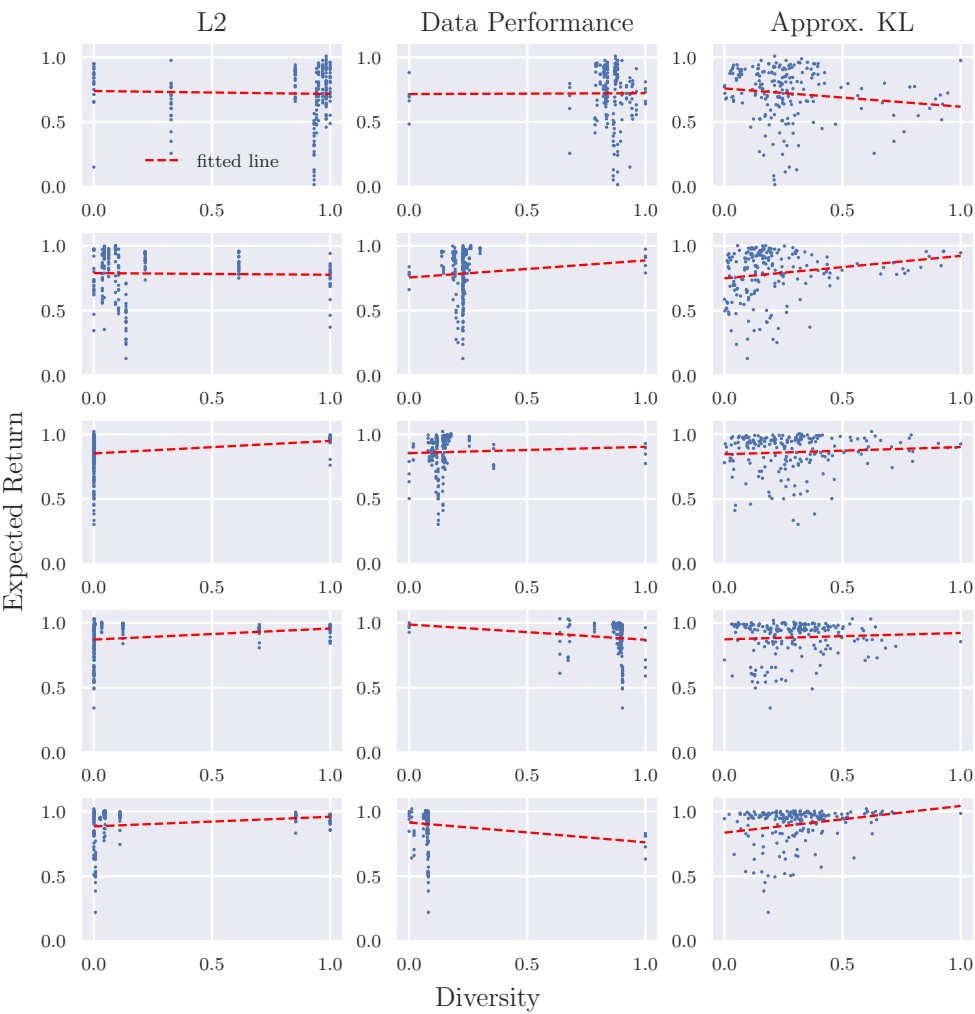

Figure 35: MTBC performance with respect to various diversity metrics in the discrete pendulum task. Each row corresponds to number of source tasks in $\{1, 2, 4, 8, 16\}$. x-axis is normalized to be in $[0, 1]$.

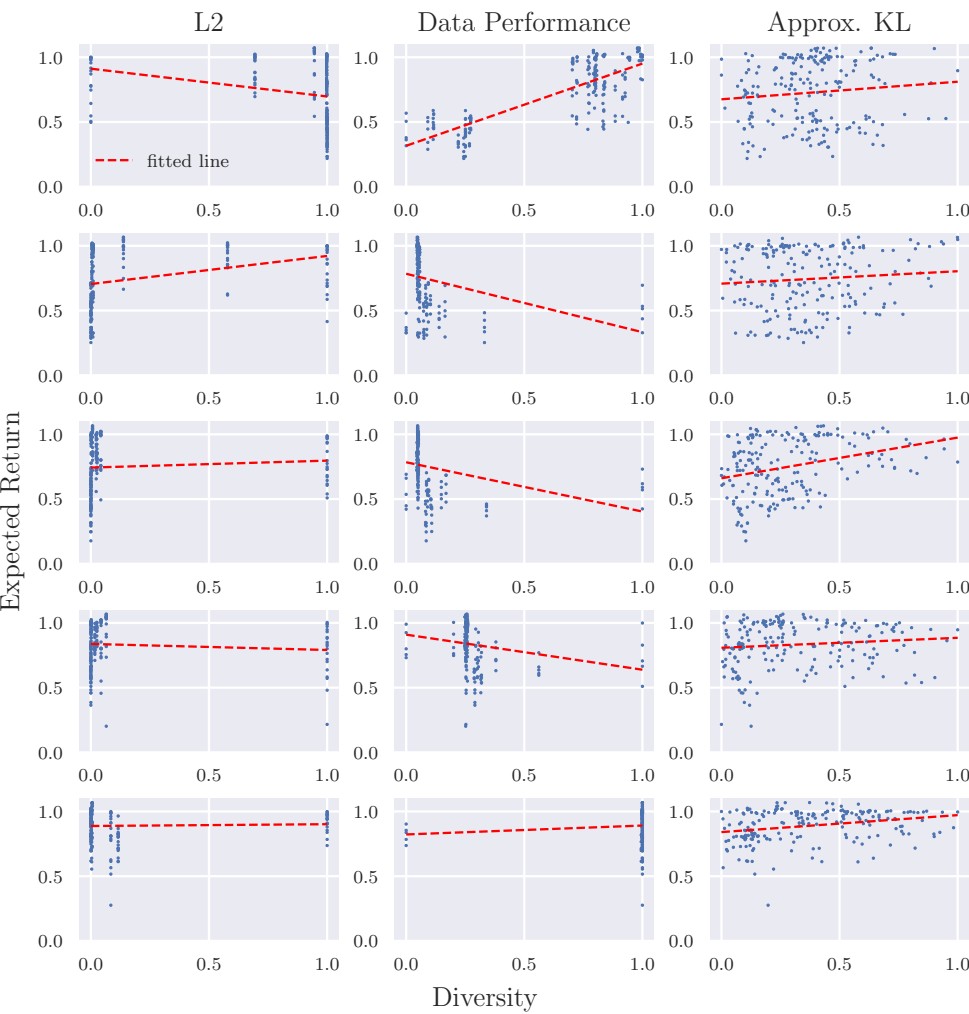

Figure 36: MTBC performance with respect to various diversity metrics in the continuous pendulum task. Each row corresponds to number of source tasks in $\{1, 2, 4, 8, 16\}$. x-axis is normalized to be in $[0, 1]$.

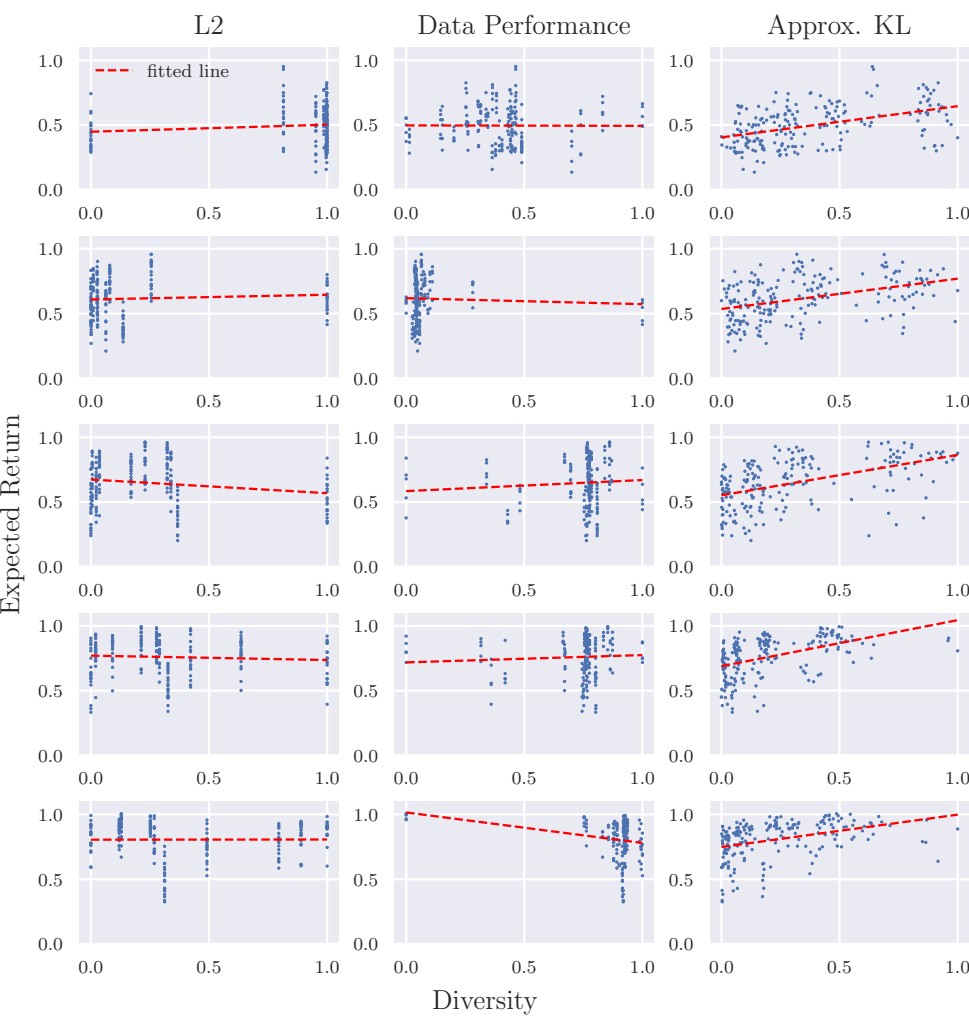

Figure 37: MTBC performance with respect to various diversity metrics in the discrete cheetah task. Each row corresponds to number of source tasks in $\{1, 2, 4, 8, 16\}$. x-axis is normalized to be in $[0, 1]$.

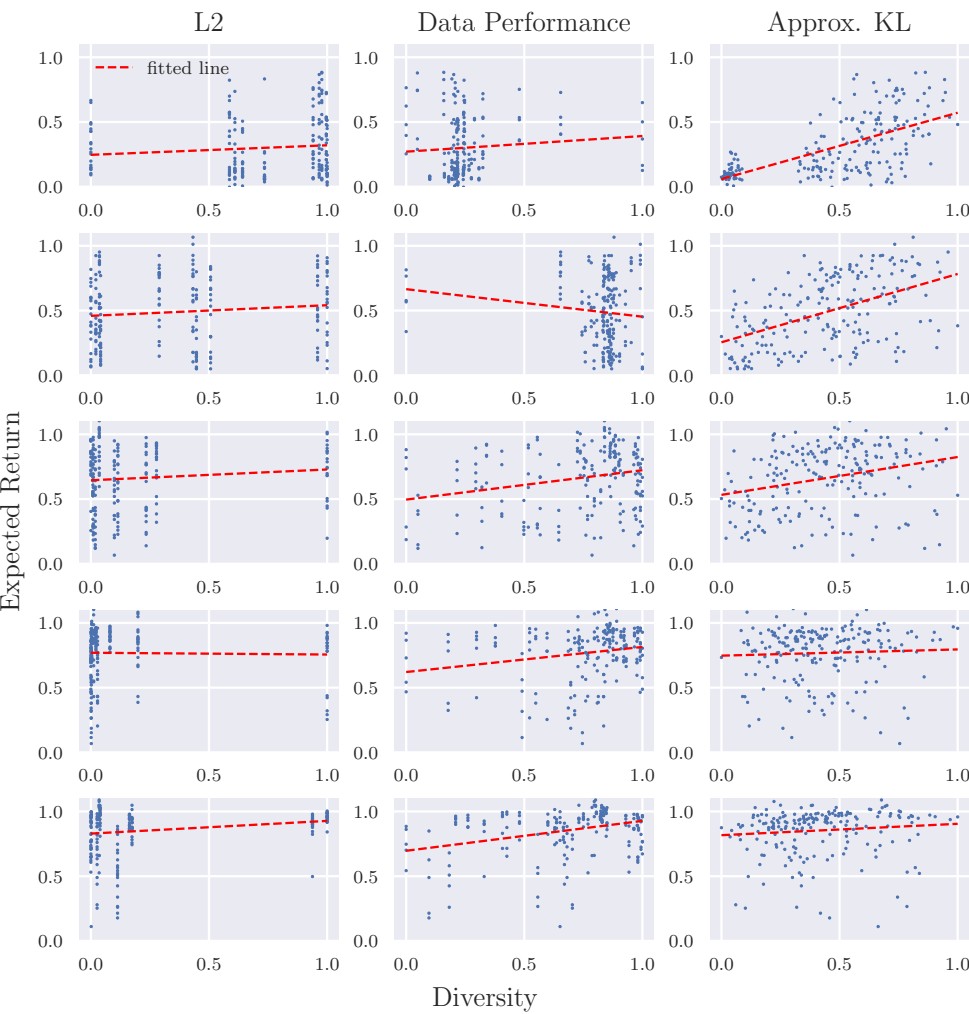

Figure 38: MTBC performance with respect to various diversity metrics in the continuous cheetah task. Each row corresponds to number of source tasks in $\{1, 2, 4, 8, 16\}$. x-axis is normalized to be in $[0, 1]$.

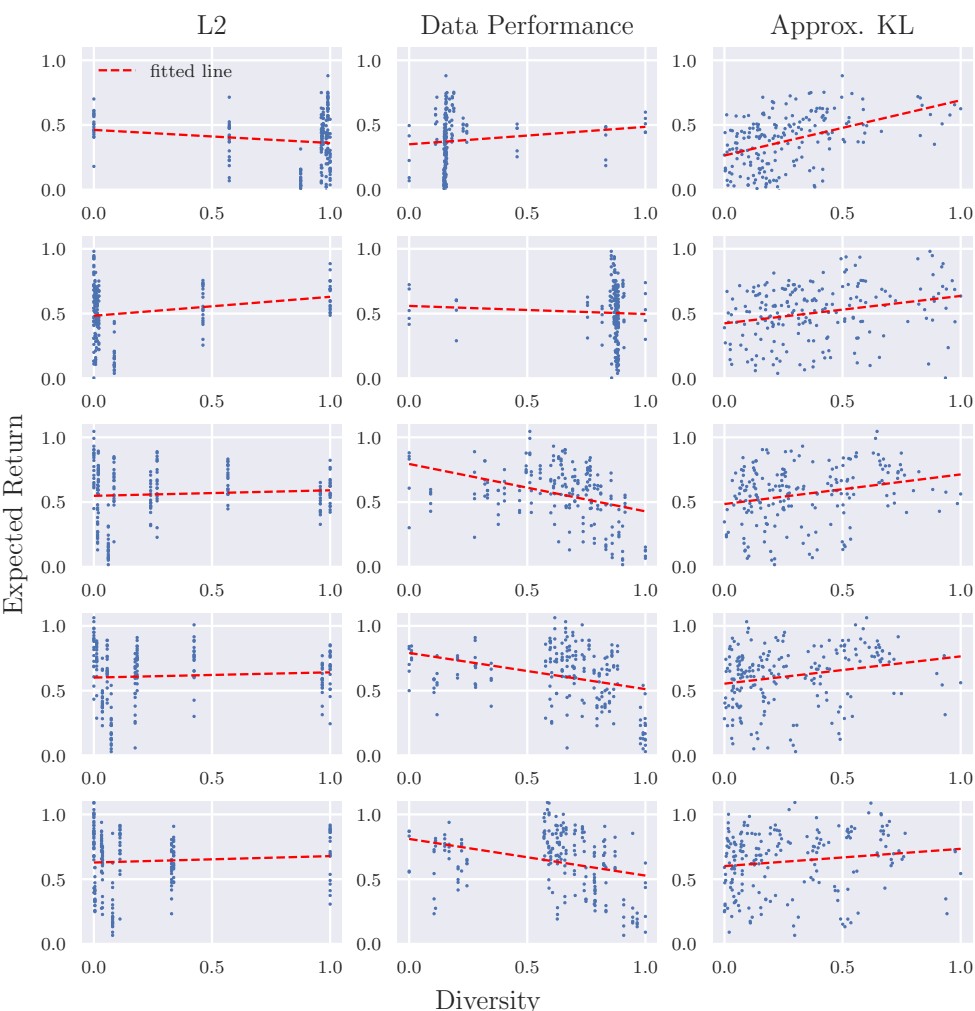

Figure 39: MTBC performance with respect to various diversity metrics in the discrete walker task. Each row corresponds to number of source tasks in $\{1, 2, 4, 8, 16\}$. x-axis is normalized to be in $[0, 1]$.

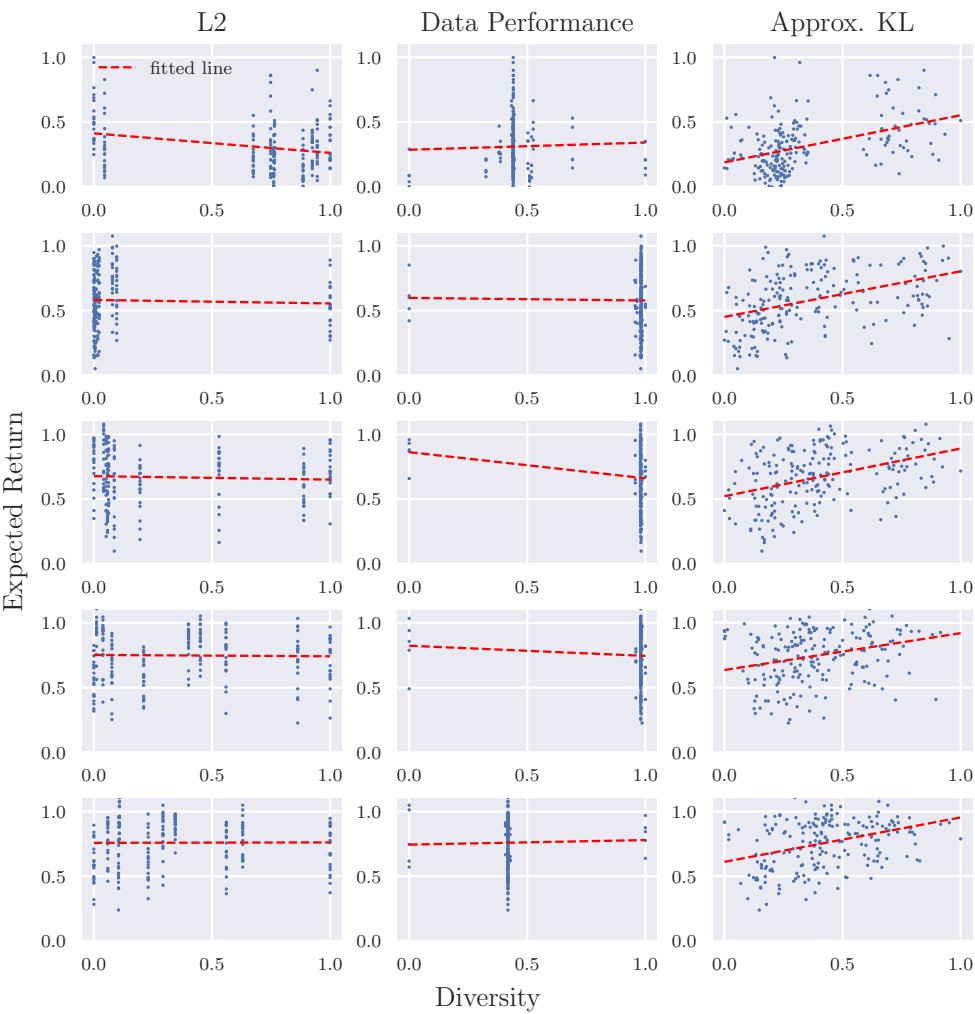

Figure 40: MTBC performance with respect to various diversity metrics in the continuous walker task. Each row corresponds to number of source tasks in $\{1, 2, 4, 8, 16\}$. x-axis is normalized to be in $[0, 1]$.

