# OpenReview forum: "The Role of Representation Transfer in Multitask Imitation Learning"
_ICLR.cc/2024/Conference — Submitted to ICLR 2024_

### Official Review · Reviewer_Fs7F · 2023-10-27

**Soundness:** 3 good
**Presentation:** 2 fair
**Contribution:** 3 good
**Rating:** 8
**Confidence:** 3

**Summary:**

This paper provides a statistical guarantee for multitask imitation learning for improved sample efficiency. Their contribution builds on others work such as Arora et al by using the Rademacher complexity. Using this they have a tighter bound which will provide benefits of transferring for imitation learning. With theoretical insights, they provide empirical results while comparing with multitask behavioral cloning. The environments they utilize are Cartpole, Frozen Lake, Pendulum, Cheetah, and Walker.

**Strengths:**

With the experiments, you have measured some scenarios in both discrete and continuous domains to show that it works. The additional experiments in the supplementary material show the amount of rigor especially with showing the task diversity metric.

The motivation of the theory makes sense and you provide a good amount of related works to show the relevance of the significance.

**Weaknesses:**

Writing
You state in the abstract “readily extended to account for commonly used neural network architectures such as multilayer perceptron and convolutional network with realistic assumptions” It would strengthen this claim if you had experiments with convolutional networks to show that it can be done. If not please reconsider modifying your claim.

In the theoretical contributions paragraph in page one, what do you mean by the second sentence, is that from the Arora et al. paper, if so please say that it refers to that because it sounds off?

Experiment
For the BC baseline, it seems like an easy one to compare and in the SM you have BC with 2|D| would it not be appropriate to also show that similar comparison with the main text experiments to show what if BC had more |D| to what your method has?

**Questions:**

Please refer to the weaknesses section.

---

> ### Author Response · Authors · 2023-11-16
>
> We thank reviewer Fs7F for their feedback. Thank you for emphasizing the signficance and rigor of our work.
>
> **W1:** Thank you for pointing this out. Our statement here refers to the theoretical result that we derived. However, as suggested by you and reviewer Eicc, we aim to provide extra analysis on a new task with image-based observations shortly.
>
> **W2:** Thank you for pointing out the clarity issue in our statement. Our sentence aims to describe that Arora et al. provided a statistical bound that holds in expectation---while this is an interesting bound, in practice we are given a particular target task. Practitioners may want to know the sample complexity of the particular transfer which is described by a high-probability bound.
>
> **W3:** Thank you for the constructive feedback. We will include an extra ablation on larger number of demonstrations shortly during the discussion period.
>
> Finally, is there any specific improvements that we can make to increase the score? Thank you reviewer Fs7F for your feedback again!

---

> ### Author Response · Authors · 2023-11-16
>
> Thank you reviewer Fs7F for suggesting to run an extra ablation on BC. We have updated our manuscript to include this result on page 29, figures 11 and 12. Our experiment includes running BC on 10 target tasks per environment, each with 5 random seeds. We normalize the returns as each target task may have different performance range.
>
> As expected, as we increase the amount of data to 4x and 8x, BC performs very closely to the expert with very low variance. While one may question the usage of multitask imitation learning, we note that the premise of pre-training using other source data is to reduce the amount of target data required. Recall that figures 3 and 4 on page 7 already demonstrates that generally we have achieved this.
>
> We thank you again for the suggestion and we are happy to continue our discussions on this work.

---

> > ### Comment · Reviewer_Fs7F · 2023-11-20
> > **Re**
> >
> > Thank you for providing the rebuttal. With the other experiments provided, I will increase my score.

---

> > > ### Author Response · Authors · 2023-11-20
> > >
> > > We thank reviewer Fs7F for their constructive feedback and reconsideration of our score. We sincerely appreciate your time and effort!

---

### Official Review · Reviewer_F18A · 2023-11-09

**Soundness:** 3 good
**Presentation:** 3 good
**Contribution:** 3 good
**Rating:** 6
**Confidence:** 3

**Summary:**

This paper discusses the advantages of using transferred representations in multitask imitation learning. The authors propose that such transfer can improve the efficiency of learning the target task by using representations learned from sufficiently diverse and related tasks, which can lead to a reduced need for data when training on a new task. They provide theoretical guarantee to support the idea that representation transfer is beneficial, which can be extended to neural network architectures such as multilayer perceptron and convolutional networks. The paper also provides empirical analysis that validate the theoretical findings. Experiments are done in simulated environments to show that leveraging data from diverse source tasks can indeed improve learning efficiency on new tasks.

**Strengths:**

- The paper is well organized and nicely written. The contributions are outlined and well emphasized, and the settings/backgrounds are well introduced. Definitions and theorem are formally stated and discussed with remarks.
- Theoretical results are reasonable as far as I read into. Extensive discussions are provided in the appendix.
- The topic on multi-task representation learning is interesting and important.

**Weaknesses:**

- It would be better to state clearly in the main paper about what assumptions are made in the paper and discuss about the limitations.
- It is difficult to read Figure 1-4. The lines are hard to distinguish from one another.

**Questions:**

See in weaknesses.

---

> ### Author Response · Authors · 2023-11-16
>
> We thank reviewer F18A for their feedback. Thank you for emphasizing the importance of our work.
>
> **W1:** Assumptions and limitations of the current theoretical results: We thank you for emphasizing the interpreting consequences of the assumptions made. Due to space limitation we would like to use the space to highlight our theoretical and experimental findings, and deferred the exact assumptions and discussions on the limitations in the appendix. However, we have modified our main paper to include the high-level assumptions made and the intuitions. See paragraph before theorem 1 on page 4.
>
> **W2:** Thank you for pointing out the clarity issue. We have updated the plot in the manuscript.
>
> Finally, is there any specific improvements that we can make to increase the score? Thank you reviewer F18A for your feedback again!

---

> > ### Author Response · Authors · 2023-11-21
> >
> > Dear reviewer F18A, since the rebuttal time is almost over, we are curious if our response addressed your questions and concerns. Thank you once again for your feedback!

---

### Official Review · Reviewer_Eicc · 2023-11-09

**Soundness:** 3 good
**Presentation:** 3 good
**Contribution:** 3 good
**Rating:** 6
**Confidence:** 4

**Summary:**

This paper addresses the statistic guarantees of transfer learning with regards to its improvements in sample efficiency, specifically with regards to imitation learning paradigms. The main result of this paper is a bound on the policy error that is indirectly related to the task diversity of the source tasks $T$, the number of demonstrations of the source task $N$, and the number of demonstrations of the target class $M$, and directly related to the Rademacher complexity. Task diversity is intuitively defined as how closely can some learned policy $\pi^*$ perform on a new task given source tasks.

The proposed method is split into two stages: first learn a representation embedding $\hat{\phi}$ from source tasks, then learn a policy $\pi$ conditioned on the task-specific mapping $\hat{f}$ and $\hat{\phi}$. The training objective of the first phase is to minimize the log loss of $\pi$ given a task-specific mapping for source task $t$ and the parameter $\phi$. The training objective of the second phase is to minimize that same loss using $\hat{\phi}$ from above, this time varying the task-specific mapping $f_\tau$. The authors perform their analysis in the tabular setting, but mention that it may be possible to extend to continuous state-action spaces in theory, and provide empirical results to support this.

The empirical questions the authors aim to answer is whether multi-task behavioral cloning training can do better than single task behavior cloning training, and ablate over $N$, $T$, and $M$ to see which affects performance the most. They find that increasing $N$ and $T$ are most impactful in improving performance and reducing the demand on target data demonstrations.

**Strengths:**

The main contributions form this paper are two fold: 1) a tighter bound on sample efficiency of multi-task imitation learning paradigms, and 2) empirical results focus on the effectiveness of representation transfer and a new metric to measure task diversity.

- The paper clearly presents the hyperparameters it is interested in that is relevant to their main bound, and does a thorough ablation over each parameter.
- The proposed KL metric is described in a digestible manner, and good to see thorough results testing its effectiveness. As mentioned, it is an important direction to prompt more empirical work analyzing task diversity.
- Empirical findings are interpretable, and good combination of graphs and tables.

Finally, the paper is generally free of grammatical errors and typos, and written in a clear manner. Overall, it is likely to be of interest to a smaller community in the multi-task learning space. However, if the authors could provide more results on experiments outside of Mujoco, such as the more challenging tasks mentioned below, it has the potential to raise interest in the larger multi-task imitation learning community.

**Weaknesses:**

### High Level Technicals:
- While the story told in Figures 1-4 are clear, it would have been nice to see some evaluations on at least one multitask environments such such as FrankaKitchen [[1](https://robotics.farama.org/envs/franka_kitchen/franka_kitchen/)] or Metaworld [[1](https://meta-world.github.io/)]. While the current results on Mujoco support the claim, the environments are relatively simple. The results of this paper would be of interest to a much larger community if the same compelling results are shown on just one of the above environments.
- It would have been nice to see some more interpretation on the results of Spearman and Kendall correlations on the bottom of page 8.
- Due to the boldness of the lines and overlap, it is a bit challenging to tell the differences between the blue, yellow, green, and red lines. I might suggest using different shapes in or decreasing the boldness of the lines.

### Low Level Technicals
- On (ii) towards the bottom half of page 7, "let" should be capitalized in "let \hat{r}_t be the average rewar of the expert..."

**Questions:**

1. Is there any demand on the optimality of the source task demonstrations? For example, would a large batch of suboptimal demonstrations actually deteriorate or stagnate improvement given the proposed method?
2. I'm curious whether increasing amount of source task data would also make the policy more robust to covariate shift, since it theoretically should have a larger state-space coverage. Or when transition dynamics are stochastic.
3. Have the authors attempted to generalize their framework to imitation learning algorithms beyond behavioral cloning, such as IRL methods?

---

> ### Author Response · Authors · 2023-11-16
>
> We thank reviewer Eicc for their feedback. Thank you for emphasizing the potential significance of our work.
>
> **W1:** Thank you for pointing this out. We agree that a task in extra environments will make the paper more impactful. As suggested by you and reviewer Fs7F, we aim to provide extra analysis on a new task with image-based observations shortly.
>
> **W2:** Both Spearman and Kendall correlations aim to find monotonic correlations. In the main manuscript we briefly discussed that they are both positively correlated, albeit rather weak in some cases. Notice that frozen lake, cartpole, and pendulum tasks are the environments with weaker correlations. Our suspicion is that each variant of these environments are very sensitive to the policy's actions. In some cases, the policy will need to make consecutively accurate actions to perform well. For example, each variant of frozen lake differs in its dynamics and initial and goal positions. Consequently the learner policy in the target task may vary significantly in actions, when compared to the learned policies in the source tasks. This relates to the last sentence where in discrete action space, there is a possibility that we can simply permute the action space and have low task diversity even if the policy can learn perfectly well.
>
> **W3:** Thank you for pointing out the clarity issue. We have updated the plot in the manuscript.
>
> **W4:** We have fixed the typo in the manuscript. Thank you for pointing that out.
>
> **Q1:** Thank you for the interesting question. Based on our theory, so long as the expert policy shares representation with the true representation, the pretraining will not stagnate or deteriorate the improvement.
> Consequently, we can likely use MMD or other divergences if the sample complexity improves and can be easily tied to the transfer risk, as suggested by reviewer StV9.
> We find that there has been work showing that some objectives work and some do not [1, 2, 3].
>
> **Q2:** Yes, we believe that is the case based on the theory. Unless we have the degenerate case where both the representation and the task mappings are tabular, intuitively the learned representation should be capturing shared dynamics information relevant across all tasks. This is because the pretraining loss is induced by the dynamics and the policies of the source tasks. Consequently, this information should improve policy robustness to the covariate shift problem.
>
> **Q3:** We have not yet considered inverse RL (IRL) methods, however we believe this work opens a new avenue to analyze various combinations of supervised IL and IRL approaches.
>
> Finally, is there any specific improvements that we can make to increase the score? Thank you reviewer Eicc for your feedback again!
>
> References:
> [1] Yang, M., & Nachum, O. (2021, July). Representation matters: Offline pretraining for sequential decision making. In International Conference on Machine Learning (pp. 11784-11794). PMLR.
> [2] Nachum, O., & Yang, M. (2021). Provable representation learning for imitation with contrastive fourier features. Advances in Neural Information Processing Systems, 34, 30100-30112.
> [3] Kumar, A., Singh, A., Ebert, F., Nakamoto, M., Yang, Y., Finn, C., & Levine, S. (2022). Pre-training for robots: Offline rl enables learning new tasks from a handful of trials. arXiv preprint arXiv:2210.05178.

---

> > ### Author Response · Authors · 2023-11-21
> >
> > Dear reviewer Eicc, since the rebuttal time is almost over, we are curious if our response addressed your questions and concerns, particular the extra interpretation of the correlations. Thank you once again for your feedback!

---

### Official Review · Reviewer_kWs2 · 2023-11-09

**Soundness:** 4 excellent
**Presentation:** 3 good
**Contribution:** 2 fair
**Rating:** 5
**Confidence:** 4

**Summary:**

The authors consider, theoretically and empirically, the sample-complexity benefits pre-training a shared representation on multi-task data might provide for behavioral cloning. In theory, they prove a high-probability bound on the performance difference between BC learned on top of the shared representation and the expert data. In practice, they show that on a variety of discrete and continuous control problems, pre-training on multi-task data allows for effective policy learning with limited target-task data.

**Strengths:**

(+) The theoretical analysis is easy to follow and uses standard tools.

(+) I appreciated how the experiments section was broken down into the statement and testing of various hypotheses.

**Weaknesses:**

(-) Overall, I found the theoretical statements to be fairly simple extensions of known results by Tripuraneni et al. and Ross & Bagnell. In essence, by focusing only on behavioral cloning, the authors are able to almost entirely ignore the sequential nature of the imitation problem and apply the standard analysis for multi-task supervised learning. Then, once they have a bound on the KL divergence between the learner and the expert, they can apply the well-known upper bounds for behavioral cloning (https://www.cs.cmu.edu/~sross1/publications/Ross-AIStats11-NoRegret.pdf) to get an overall policy performance guarantee. So, I really didn't get much out of the theorems they proved.

(-) There's a few pieces of odd terminology throughout the paper. First, instead of "policy error", people usually use "performance difference" or "imitation gap" (https://arxiv.org/abs/2103.03236). Also, I think you're missing a $H^2$ or $\frac{1}{(1-\gamma)^2}$ in the first equation in the paper? Second, when people talk about "number of demonstrations" (i.e. $|\mathcal{D}|$ in the paper), they usually mean the number of whole trajectories rather than the number of state-action tuples (as you seem to use it in Table 1). Do you mind re-naming this? I got super confused for a while by why one would need millions of samples for BC to work on a Mujoco task. Can you also clarify whether $N$ and $M$ are measured in terms of trajectories or state-action pairs?

(-) Once you divide the numbers in table 1 by the horizon of the problem (1000 for Mujoco), you realize that they're attempting to learn based on effectively 1-2 demonstrations. This is a somewhat absurdly small amount of data (usually people do ~25 demos for Mujoco tasks). So, while the experimental results make sense to me, I think it is somewhat important to note that they are under fairly contrived settings.

(-) While the idea of a metric to capture the effectiveness of multi-task IL data for learning a transferrable representation is interesting, I found the ideas in Appendix B to be a bit sloppy and the empirical reported correlations in Tables 2/3 to be fairly low.

(-) Most analysis of imitation learning doesn't have to make assumptions about the optimality of the expert policy. In Footnote 1, you note that you do this. Is this actually important for any of your analysis?

(-) I might add some more citations for multi-task imitation learning outside of behavioral cloning. While they are clearly different than your work, it would be good to add in some references and discuss the differences: https://arxiv.org/pdf/1805.12573.pdf, https://arxiv.org/pdf/1909.09314.pdf, https://arxiv.org/pdf/1805.08882.pdf, https://arxiv.org/pdf/2309.00711.pdf. You might also want to cite some work on representation learning for sequential decision making (e.g. https://arxiv.org/abs/2207.08229).

**Questions:**

(1) I think it would be helpful if you could add in a comparable statement to Theorem 1 for single-task BC. You could then give ranges of T and N under which you'd have a meaningful difference in upper bounds between multi-task and single-task BC. The sharpest analysis I know for IL under a deterministic expert assumption is in https://arxiv.org/pdf/2205.15397.pdf -- you might be able to just copy some of their theorems.

(2) In Figure 1, why do some of the performances for the multi-task method start lower than the corresponding BC performances? Do you think this would be fixed if you included target task $\tau$ in the representation learning step?

(3) Do you have any hypothesis for why, in Figure 2, things look quite bad for CartPole? Is it perhaps because of the kinds of environment modifications you were considering?

(4) In Figures 3 and 4, you're giving the learner quite a bit of source data so perhaps the performance gap is already quite close. If you have the compute resources, could you ablate these results across smaller values of $N$?

(5) Generally, could you be more specific about the ranges over which you varied environment parameters to generate the multi-task data (e.g. what link sizes for Walker)?

---

> ### Author Response · Authors · 2023-11-16
>
> We thank reviewer Fs7F for their insightful feedback. We believe we have addressed your concerns and we are very thrilled to continue the thoughtful discussion with you.
>
> **W1:** We thank reviewer Fs7F for noting the simplicity of our theoretical proofs. We agree that our theoretical result is a synthesis of known results and is straightforward. Our novelty is not a new proving technique. Instead, our novelty lies in bridging and connecting learning theories in multitask learning and imitation learning. As noted by many other reviewers (StV9, Eicc, F18A, Fs7F, Ss6u), this result does not appear to be commonly known and can have significant impact in the community. Furthermore, we note that our result is an improvement of existing results by leveraging the properties of the log loss. With any policy parameterization that is Lipschitz, we can improve the sample complexity by $\log(NT)$ when compared to using Gaussian complexity. We finally argue that a simpler proof is often preferred since it can be easily understandable, which appears to be the case as stated by many reviewers (StV9, kWs2, Eicc, F18A).
>
> **W2:** We thank reviewer Fs7F for raising an undefined term in the paper. Based on your feedback, we have modified our terminology to imitation gap.
>
> Regarding the horizon term, indeed we have neglected the horizon term. Due to Xu et al, 2020. this is an unavoidable term and we thus treat it as a constant. We further note that we also neglected other constants (e.g. $\lvert \mathcal{A} \rvert$) for presentation purposes.
>
> Finally, thank you for raising this concern of the term "demonstration". We emphasize that we treat each demonstration as a state-action pair (i.e. a transition) and we have explicitly defined this in section 2.2. We wish to keep the term demonstration as transitions can be referred to any policies, and terms such as "expert transitions" are cumbersome from the reading perspective. However, we are open to suggestions on what can be a better term.
>
> **W3:** Thank you for pointing out that the number of trajectories is small for MuJoCo tasks in many papers [1,2]. We believe that this "contrived setting" is preferrable in practice---practitioners aim to minimize the number of demonstrations as much as possible since it can be costly (e.g. human-operating time, computational cost, etc.)
>
> **W4:** We acknowledge reviewer kWs2's comment on the task-diversity metric. We note that this is an attempt to defining a task-diversity metric that may positively correlate with the imitation performance. Indeed, we do not provide any theoretical guarantees on the metric and the conditions in appendix B are very restrictive. Intuitively we hope to have the irreducible error to be sufficiently small such that the ratio can be estimated by our proposed metric. We consequently experimentally showed that this metric can be strongly correlated to the imitation performance. Furthermore, the metric still captures the asymmetry of task transfer which is a desirable property.
>
> **W5:** Thank you for raising this. Our result does not require optimality of the expert policy as the learner policy only aims to imitate the expert. We have removed footnote 1 which hopefully resolves the confusion.
>
> **W6:** Thank you for providing extra references for multi-task IRL methods. We have included them in the first paragraph of the related work. We hope this better covers the literature and demonstrates the differences in our work.

---

> > ### Author Response · Authors · 2023-11-16
> >
> > **Q1:** Thank you for the suggestion. We believe your provided reference paper, Xu et al., 2020, and Ross and Bagnell, 2011 are all great references for indicating the BC sample complexity. In fact, our proof of Theorem 3 also works with minimal changes.
> > Consider the ERM $\min_{f \in \mathcal{F}, \phi \in \Phi} R_{test}(f, \phi)$ and follow similar proof techniques, then we have that w.p. $1 - \delta$,
> > $$
> > R_{test}(f, \phi) - R_{test}(f^*, \phi^*) \leq 4 \mathfrak{R}_M (\ell \circ \mathcal{F} \circ \Phi) + 2B \sqrt{\frac{\log (2/\delta)}{2M}}.
> > $$
> >
> > Perhaps through this bound we can easily see that the first term will dominate RHS and will require larger $M$ when we have an expressive model such as a neural network.
> >
> > **Q2:** Thank you for the question and the suggestion. We hypothesize that it has to do with the shared representation is significantly different from the true shared representation, as the theory would suggest. We plan to verify this hypothesis with your suggestion on including the target samples in the pre-training step.
> >
> > **Q3:** We find cartpole as a very sensitive environment generally. If you consider figure 10 in the appendix on page 27, you can see that BC performance does not increase after very small number of demonstrations on the default environment. We suspect that some of the source tasks include these difficult variants, thereby impeding the performance of MTBC.
> >
> > **Q4:** Thank you for the suggestion, we are currently gathering the results for this experiment and will provide them along with other extra experimental results.
> >
> > **Q5:** Yes, we are happy to share this:
> >
> > Frozen lake:
> > 1. We vary the initial and goal positions uniformly (disallowing equal positions)
> > 1. We sample uniformly the probability of not slipping $p_1 \sim U[0, 1]$, then sample uniformly the probability of slipping to one adjacent cell $p_2 \sim U[0, 1 - p_1]$, and assign the remaining mass $1 - p_1 - p_2$. This probability is assigned to all state-action pairs.
> >
> > Cartpole:
> > 1. Joint stiffness: $U[0, 5]$
> >
> > Pendulum:
> > 1. Pendulum max torque: $U[0.01, 4]$
> >
> > Cheetah:
> > 1. Front and back thigh joints:
> >     1. Damping: $U[4, 8]$
> >     1. Stiffness: $U[230, 250]$
> > 1. Front and back feet:
> >     1. Link size: $U[0.04, 0.05]$ by $U[0.065, 0.075]$
> >
> > Walker:
> > 1. Left and right feet:
> >     1. Link size: $U[0.04, 0.06$ by $U[0.09, 0.11]$
> > 1. Left and right legs:
> >     1. Link size: $U[0.03, 0.05$ by $U[0.23, 0.27]$
> > 1. Left and right thighs:
> >     1. Link size: $U[0.04, 0.06$ by $U[0.2, 0.25]$
> > 1. Left and right knee joint:
> >     1. Range: $U[-140, -160]$
> >
> > We hope you can reconsider our score after our response. Thank you reviewer kWs2 for your feedback again!
> >
> > References:
> > [1] Ho, J., & Ermon, S. (2016). Generative adversarial imitation learning. Advances in neural information processing systems, 29.
> > [2] Kostrikov, I., Agrawal, K. K., Dwibedi, D., Levine, S., & Tompson, J. (2018, September). Discriminator-Actor-Critic: Addressing Sample Inefficiency and Reward Bias in Adversarial Imitation Learning. In International Conference on Learning Representations.

---

> > > ### Author Response · Authors · 2023-11-18
> > >
> > > We once again thank reviewer Fs7F for their feedback. We have conducted an experiment on including the target task during the training phase. The result can be found on page 29, figure 13. Our preliminary result shows that including the target task does indeed help with the imitation performance of MTBC, generally matching the imitation performance of BC and improved upon MTBC without the target task for the training phase. We can also see that as the number of source tasks increases, the imitation performance of MTBC **without the target task** can match MTBC **with the target task**, which aligns with our theoretical result.

---

> > > > ### Author Response · Authors · 2023-11-18
> > > >
> > > > Thank you reviewer Fs7F for proposing the various ablation studies. We have finally conducted the experiment where the number of source data is decreased to $N = \lvert \mathcal{D} \rvert$ from $N = 8 \lvert \mathcal{D} \rvert$. The result is on page 30, figures 14 and 15. In short, we can see that when $T$, the number of source tasks is small, MTBC with larger $N$ significantly outperforms smaller $N$. We once again thank you for your detailed review and insightful discussion and would love you to reconsider our score.

---

> > > > > ### Comment · Reviewer_kWs2 · 2023-11-19
> > > > > **Re:**
> > > > >
> > > > > First off, let me thank the authors for their thorough and prompt rebuttal. Responding to the points raised:
> > > > >
> > > > > **W2:** You might want to cite some paper for the term "imitation gap" -- https://arxiv.org/abs/2103.03236 is where I first saw the term if it helps. While I agree that the $O(H^2)$ factor is unavoidable without further assumptions (e.g. coverage / recoverability, https://arxiv.org/pdf/2102.02872.pdf), I don't think this is a good reason to not include it in the presented bound, given most other papers in the offline IL space do.
> > > > >
> > > > > Re: "demonstrations:" given I and other reviewers seemed to be confused by this point, I would suggest using something like "expert samples" / "expert tuples" or divide all the numbers by the horizon. As a general principle, if multiple folks in your target audience are confused by your use of a term (regardless of whether it was re-defined in the text), this is not a good thing.
> > > > >
> > > > > Q1: Yup, makes total sense. Do you mind adding this in as a corollary / remark? I think it'd make the precise sample-complexity benefits of multi-task BC more apparent.
> > > > >
> > > > > Q2: Nice work! Do you mind adding in a sentence on this point in the figure caption / a footnote?
> > > > >
> > > > > Q4: Ditto the above.
> > > > >
> > > > > Q5: Thanks! Do you mind sticking this somewhere in the appendix?

---

> ### Author Response · Authors · 2023-11-19
>
> We thank reviewer kWs2 for the feedback. We have updated our manuscript to reflect the follow-up suggestions.
>
> **W2:**
> - We have included the suggested reference on page 2, under section 2.2.
> - We have also included the horizon term in page 1, under **Theoretical contributions**.
>
> Q1:
> - We have added a remark (Remark 1) on page 4 to include the sample-complexity of BC, as well as comparing BC with MTIL on page 4.
>
> Q2 & Q4:
> - We have provided a sentence in the caption of figures 13-15 on pages 30 and 31, as well as including paragraphs under **Inclusion of Target Data during Training Phase** and **Impact of Source Tasks and Source Data** on pages 28 and 29.
>
> Q5:
> - We have included the parameter ranges and described how we sample them on pages 24 and 25.
>
> Finally, we thank reviewer kWs2 for their constructive feedback. We invite reviewer kWs2 to comment on the other points as the feedback has been very helpful. We also would love reviewer kWs2 to once again reconsider their score.

---

> > ### Comment · Reviewer_kWs2 · 2023-11-19
> > **Re:**
> >
> > Thank-you for those updates!
> >
> > One last question: rather than comparing the KL-Divergence between the expert and the multi-task policies as your transfer metric, do you think it would make sense to, say, compare the $\ell_2$ norm of the difference in weights? I've seen this sort of metric used for task diversity in meta-learning (e.g. https://arxiv.org/pdf/1906.02717.pdf). Part of the reason it might be an interesting contrast to the methods you propose is that it is not a function of the expert state distribution (and therefore could correlate better with how well one has actually recovered the expert policy). Of course, one could not actually evaluate such a metric without full access to the expert policy.

---

> > > ### Author Response · Authors · 2023-11-19
> > >
> > > Thanks for bringing up the $\ell_2$-norm of the parameters as a metric. We have actually started considering this very recently---even the single-task setting will be interesting. In particular only recently there is result on a simple logistic regression [1]. As you have mentioned this somewhat decouples the metric from the state distribution of the expert policy might be beneficial.
> > >
> > > Thank you for the referenced paper, we have not seen this before but this appears to be very relevant to our future work.
> > >
> > > Reference:
> > > [1]: Hsu, D., & Mazumdar, A. (2023). On the sample complexity of estimation in logistic regression. arXiv preprint arXiv:2307.04191.

---

> > > > ### Comment · Reviewer_kWs2 · 2023-11-20
> > > > **Re:**
> > > >
> > > > If you still have the trained policies sitting around, would you consider adding it as another $\hat{\sigma}$ to the paper? I'd be quite curious as to how it works compared to the other candidates.

---

> > > > > ### Author Response · Authors · 2023-11-20
> > > > >
> > > > > We thank reviewer kWs2 for the extra suggestion. We have included using $\hat{\sigma} = \frac{1}{\lVert \phi - \hat{\phi} \rVert_2}$ as the task-diversity metric on discrete tasks. The preliminary result generally indicates a positive correlation between the L2 representation norm and the imitation performance. As reviewer kWs2 has suggested, however, this metric will require the representation of the expert policy which may not be accessible a priori.
> > > > >
> > > > > We have included this table (table 6) on page 32, with explanation on pages 30 and 31. Once again we thank reviewer kWs2 for their suggestions and would be delighted if they can reconsider our score.
> > > > >
> > > > > ## Pearson
> > > > > |               | Frozen Lake                      | Pendulum                         | Cheetah                          | Walker                           |
> > > > > |---------------|----------------------------------|----------------------------------|----------------------------------|----------------------------------|
> > > > > | L2            | $ - 0.008  \pm  0.023 $          | $ 0.059  \pm  0.069 $            | $ - 0.008  \pm  0.080 $          | $ - 0.016  \pm  0.096 $          |
> > > > > | Data Perf.    | $ 0.035  \pm  0.223 $            | $ - 0.027  \pm  0.023 $          | $ - 0.057  \pm  0.041 $          | $ 0.279  \pm  0.128 $            |
> > > > > | Bhattacharyya | $ - 0.358  \pm  0.098 $          | $  0.071  \pm  0.013 $ | $ 0.150  \pm  0.029 $            | $ - 0.041  \pm  0.011 $          |
> > > > > | L2 Repr. Norm | $0.169 \pm 0.040$ | $\mathbf{0.190 \pm 0.034}$ | $0.381 \pm 0.060$ | $0.048 \pm 0.015$          |
> > > > > | Approx. KL (Ours)    | $ \mathbf{ 0.276  \pm  0.118 } $ | $ - 0.099  \pm  0.017 $          | $ \mathbf{ 0.386  \pm  0.062 } $ | $ \mathbf{ 0.383  \pm  0.068 } $ |
> > > > >
> > > > > ## Spearman
> > > > > |               | Frozen Lake                      | Pendulum                         | Cheetah                          | Walker                           |
> > > > > |---------------|----------------------------------|----------------------------------|----------------------------------|----------------------------------|
> > > > > | L2            | $ - 0.019  \pm  0.010 $          | $ - 0.064  \pm  0.032 $          | $ 0.091  \pm  0.024 $            | $ - 0.008  \pm  0.155 $          |
> > > > > | Data Perf.    | $ \mathbf{ 0.035  \pm  0.028 } $ | $ 0.116  \pm  0.026 $            | $ 0.053  \pm  0.071 $            | $ - 0.159  \pm  0.083 $          |
> > > > > | Bhattacharyya | $ - 0.138  \pm  0.046 $          | $ 0.079  \pm  0.026 $            | $ 0.249  \pm  0.037 $            | $ 0.045  \pm  0.014 $            |
> > > > > | L2 Repr. Norm | $0.002 \pm 0.010$ | $\mathbf{0.288 \pm 0.048}$ | $\mathbf{0.404 \pm 0.064}$ | $0.125 \pm 0.027$          |
> > > > > | Approx. KL (Ours)   | $ 0.005  \pm  0.022 $            | $  0.223  \pm  0.036 $ | $ 0.363  \pm  0.054 $ | $ \mathbf{ 0.242  \pm  0.039 } $ |
> > > > >
> > > > > ## Kendall
> > > > > |               | Frozen Lake                      | Pendulum                         | Cheetah                          | Walker                           |
> > > > > |---------------|----------------------------------|----------------------------------|----------------------------------|----------------------------------|
> > > > > | L2            | $ 0.111  \pm  0.022 $            | $ 0.052  \pm  0.105 $            | $ - 0.082  \pm  0.059 $          | $ 0.010  \pm  0.044 $            |
> > > > > | Data Perf.    | $ - 0.033  \pm  0.006 $          | $ 0.137  \pm  0.068 $ | $ 0.046  \pm  0.023 $            | $ - 0.355  \pm  0.053 $          |
> > > > > | Bhattacharyya | $ - 0.203  \pm  0.030 $          | $ 0.136  \pm  0.043 $            | $ 0.422  \pm  0.064 $            | $ 0.022  \pm  0.026 $            |
> > > > > | L2 Repr. Norm | $\mathbf{0.206 \pm 0.033}$ | $\mathbf{0.185 \pm 0.031}$ | $\mathbf{0.490 \pm 0.070}$ | $0.168 \pm 0.028$         |
> > > > > | Approx. KL (Ours)   | $ 0.121  \pm  0.031 $ | $ 0.012  \pm  0.025 $            | $ 0.479  \pm  0.069 $ | $ \mathbf{ 0.220  \pm  0.039 } $ |

---

> > > > > > ### Comment · Reviewer_kWs2 · 2023-11-20
> > > > > > **Re:**
> > > > > >
> > > > > > Thanks! I have raised my score to a 5.

---

> > > > > > > ### Author Response · Authors · 2023-11-20
> > > > > > >
> > > > > > > We thank reviewer kWs2 for dedicating the time for the detailed discussions and invaluable feedback! We truly appreciate the amount of time and effort.

---

### Official Review · Reviewer_Ss6u · 2023-11-10

**Soundness:** 3 good
**Presentation:** 2 fair
**Contribution:** 2 fair
**Rating:** 5
**Confidence:** 3

**Summary:**

The paper posits that there exists a shared representation across a variety of tasks. It trains behavior cloning from a set of source tasks, learns a representation, and then learns a policy for a target task. It proposes that, the policy error is bounded by the diversity of the source tasks. It suggests that training in this paradigm will have better performance and uses less target task data than vanilla behavior cloning.


-----  Edit -----

I thank the author for writing the paper and their efforts in the rebuttal. They cleared some of my questions. But one of my major concern remains: How to define the shared representation? Since the objective of the paper is to provide a bound regarding learning a shared representation, it would be useful to define such representation clearly and setup appropriate experiment to learn said representation and validate the proposal.

It would be helpful to explain, on the current experiments with low-dimensional state space like Pendulum, what does the learned shared representation capture?

Further, it would be really insightful to:
- (1) include environments commonly studied in multi-task setting that have different task rewards: e.g., MetaWorld of FrankaKitchen as mentioned by Reviewer Eicc. (Given the paper considers source tasks with varying rewards in Sec 3).
- (2) include environments on visual inputs: e.g., the Pendulum visual version mentioned by the author. (Given the author clarification response).

I really appreciate the problem studied in the paper and the efforts of the author, looking forward to seeing more empirical validation of this paper in the future.

**Strengths:**

- Innovative Concept: The paper introduces an interesting hypothesis about the benefits of learning shared representations from diverse source tasks to improve policy learning in behavior cloning.
- Theoretical Contribution: It provides a theoretical framework that bounds policy error with respect to the source task diversity, offering a new perspective on the potential for generalization in behavior cloning.

**Weaknesses:**

- Lack of Clarity: The paper does not sufficiently describe the "shared representation" it aims to learn. A more detailed exposition, possibly including visualizations or analysis of the learned representation, is needed.
- Theoretical Bound Practicality: The paper presents an order bound on policy error but does not provide a comprehensive discussion on its tightness or practical applicability, leaving its usefulness in question.

**Questions:**

Section 4 requires further detail on the nature of the source tasks for each target task investigated. The concept of "shared representation" is pivotal yet remains vague within the paper. Is this representation a transformation from a visual image to a latent space, or something else? A detailed analysis or visual depiction of this shared representation would greatly enhance the clarity, perhaps focusing on a single task as an example.

The paper introduces a theoretical bound but does not elucidate on its tightness or practical applicability. It is essential to quantify or provide conditions under which the bound holds with a fixed constant, thereby ensuring utility in policy improvement with additional data.

How do you define the “policy error” on Page 1? How can it be measured? Is the measurement done in the task reward, action space, or divergence perspective? Besides, you mentioned that f-divergence imitation learning states that a learned policy is minimizing the divergence between expert and learner trajectory. How does the policy error fit / contrast with such existing framework?

Given the focus on multi-task imitation learning, it is imperative to benchmark against state-of-the-art Meta Learning methods for a comprehensive comparison.

“However, current methods require thousands of demonstrations even in simple tasks (Mandlekar et al., 2022; Jang et al., 2021; Ablett et al., 2023)” => *Thousands* of demonstrations seems to misrepresent the SOTA of IL? Refer to https://www.roboticsproceedings.org/rss19/p009.pdf  https://medium.com/toyotaresearch/tris-robots-learn-new-skills-in-an-afternoon-here-s-how-2c30b1a8c573 https://deepmind.google/discover/blog/robocat-a-self-improving-robotic-agent/  Please clarify.

For all experiments detailed in Section 4, please define the state and action space, including their dimensions.

Can the derived bound be applied to low-dimensional imitation learning that does not learn a representation? Assuming that the learned representation is the identity matrix, can we extend the bound to low-dimensional state space? Is the result implying that, adding more data into pre-training, will result in smaller policy error?  However, given what we saw in https://ieeexplore.ieee.org/abstract/document/10161474/ it seems that more data is not guaranteed to help imitation learning performance on the real robot. Can the author provide some insight?


Our result is due to the objective of behavioral cloning, where the method aims to minimize the Kullback–Leibler (KL) divergence between the expert and the learner (Ghasemipour et al., 2019; Xu et al., 2020). => Could you elaborate how the findings from Xu et al., 2020 is related to this statement?

Regarding Equation 4 and the "under some assumptions" qualifier, a more intuitive explanation of these assumptions and their impact on the model's generalizability would be essential.

---- Edit ----

I had a question on "what does this paper imply for low-dimensional state representation IL". I listed a paper that stated (Offline RL) "algorithms are not guaranteed to increase performance by including more data." It would be enlightening to extend this proposal to Offline RL. However, in retrospective, this request is probably out of scope for this proposal. I thank the author for clarification and sharing insights.

---

> ### Author Response · Authors · 2023-11-16
>
> We thank reviewer Ss6u for their relevant feedback. We hope our response encourages further discussions.
>
> **W1:** We acknowledge reviewer Ss6u's concern on the structure of the shared representation. The benefit of our theoretical result is that the shared representation has a minimal structure (e.g. in $\mathbb{R}^d$ and bounded). The sample complexity bound holds with high probability. Indeed, we believe that imposing a particular structure on the representation will yield a potentially better result and will further enable practitioners to understand the problem. The state-estimation problems in robotics are great examples for imposing structures on the representation. For example, consider a image-based pendulum task where we have no true readings of the joint information. If we are given the joint information we can solve it using linear controllers. Thus our representation can simply be enforced to be in the same space.
>
> **W2:** We thank reviewer Ss6u for highlighting the importance of lower bounds---we agree that a lower bound will provide significant insight to our result, specifically to understand whether our result is minimax optimal. We first emphasize that this result means that it is hopeful to obtain a near-expert policy as we increase the number of samples, at the rate of $O(1/\sqrt{NT} + 1/\sqrt{M})$. Regarding the lower bound, there are existing results on general Lipschitz and smooth loss functions (see table 1 of [1]). In the single-task BC setting, indeed the $\Omega(1/\sqrt{M})$ is tight and can perhaps be tightened to $\Omega(1/M)$ with specific function classes (i.e. $L^*$ shrinks to 0 thus removing the $\Omega(1/\sqrt{M})$ dependence). While this bound is not established on the multi-task setting, Our intuition is that following similar proof techniques we will arrive at similar $\Omega(1/\sqrt{NT} + 1/\sqrt{M})$ result. We thank reviewer Ss6u again for the very helpful comment and we plan to clearly address this in our future work due to time limitation of the rebuttal period.
>
> **Q1:** We acknowledge reviewer Ss6u's comment on the potential vagueness of the shared representation's structure. Generally, the representation can be a transformation from a visual image to a latent space, texts to a latent space, or a stacked sequence of past robot information into a latent space---our theoretical result imposes minimal latent space structure (other than $\mathbb{R}^d$ for example). An intuitive example is a visual pendulum task where the input space is a stack of images, which allows the representation to capture positional, velocity, acceleration, and higher-order derivatives.
>
> **Q2:** We thank reviewer Ss6u for pointing out the lack of constants, conditions and assumptions made in the main paper. We provide the technical details and exact theorem result with constants in appendix C with a high-level explanation on each assumptions made.
>
> Regarding practicality, we believe we can address this through whether the assumptions are reasonable and whether the bound can be used in practice. The former is provided in appendix C.1. For the latter, we believe a concern is whether certain assumptions hold---if not, the bound is hopeless and is uninformative. However, albeit tedious, we can still validate these assumptions by increasing the complexity of the models. Suppose we make the correct assumptions, then there are multiple facets that are practical. First, we can expect the policy performance to shrink at a square-root rate, with respect to the source and target data, meaning that it is hopeful to obtain near-expert performance. Second, we know exactly how much target data to collect in order for the imitation gap to shrink below the desired value as it is independent of the task-diversity constant. Third, the fact that the representation is more complex, we are free to gather more source, perhaps even include the target data in the pretraining as suggested by reviewer kWs2. The benefit here is that it is possible for some task data to be cheaper to collect, thus we can focus on exploiting the cheaper tasks.

---

> > ### Author Response · Authors · 2023-11-16
> >
> > **Q3:** We thank reviewer Ss6u for raising an undefined term in the paper. Based on reviewer kWs2's feedback, we have modified our terminology to imitation gap. The imitation gap is measured through the max-norm of the difference between the value functions of the learner and expert policy.
> >
> > **Q4:** Thank you for bringing up the meta-learning direction. As pointed out in the conclusion of our paper, understanding meta-learning approaches is very important and we aim to analyze their sample complexity bounds in the future. In particular, the benefit of some meta-learning approaches including MAML [2] and ANIL [3] lies in the capability to adjust the shared representation during test time. There is indeed initial work [4] on theoretically understanding these methods.
> >
> > **Q5:** We appreciate reviewer Ss6u for pointing out the potential misrepresentation of the sample complexity. We note that in our paper a demonstration corresponds to a state-action pair, known as the transition. On the other hand, the references use trajectories, a sequence of state-action pairs, as the definition of a demonstration. When taking into account the horizon/length of the trajectories, we will get similar number of transitions. For conciseness, we wish to keep the term "demonstration" as we have defined this explicitly in section 2.2.
> >
> > **Q6:** We have provided the environment description in appendix F.1. We will also describe the environments here for completeness:
> >
> > *Discrete environments*
> > |                             | Frozen Lake | Pendulum | Cheetah | Walker |
> > |-----------------------------|-------------|----------|---------|--------|
> > | dim($\mathcal{S}$)          | 64          | 3        | 17      | 24     |
> > | $\lvert \mathcal{A} \rvert$ | 5           | 11       | 64      | 64     |
> >
> > *Continuous environments*
> > |                    | Cartpole | Pendulum | Cheetah | Walker |
> > |--------------------|----------|----------|---------|--------|
> > | dim($\mathcal{S})$ | 5        | 3        | 17      | 24     |
> > | dim($\mathcal{A})$ | 1        | 1        | 6       | 6      |
> >
> > **Q7:** We thank reviewer Ss6u for bringing up the surprising result in real-life robotics experiments. After reading the referenced paper, we would like reviewer Ss6u to clarify the statement, "more data is not guaranteed to help IL performance on the real robot." In particular, section III-d of the paper states that:
> > ```
> > Dataset: Our Real-ORL dataset consists of around 3000 trajectories and the characteristics of our dataset is shown in Table. II. To validate the quality of our datasets, we use the Top-K% portion of dataset and trained behavior cloning for each task (K ∈ {100, 90}), observing that using the full datasets allowed behavior cloning to achieve the best performance on all tasks. To avoid biases favoring task/algorithm, we froze the dataset ahead of time.
> > ```
> > which seems to agree with our sample complexity bound.
> >
> > Regarding the derived bound being applied to low-dimensional IL, our theory suggests that the learner will pay a constant factor of the state-space dimension, which is captured within the Rademacher complexity.
> >
> > **Q8:** The result (Theorem 1) from Xu et al., 2020 indicates that the policy error (imitation gap) can be upper bounded by the KL-divergence between policy $\pi$ and expert $\pi^E$. In standard behavioural cloning, we often aim to maximize the log-likelihood of the policy based on data generated by the expert (i.e. $\max_\pi \mathbb{E}_{s \sim \rho^*} \log \pi(a \vert s)$). Consequently, this objective is equivalent to the log loss and corresponds to the KL-divergence when the expert is deterministic.
> >
> > **Q9:** Thank you for raising the challenge in understanding Eq. 4 and the assumptions. As mentioned in the response of Q2, we provide the exact theorem result with constants in appendix C with a high-level explanation on each assumptions made. We have further made modifications to our draft to explain the high-level assumptions made and the intuition. See paragraph before theorem 1 on page 4.
> >
> > Finally, thank you again for your insightful feedback. Please let us know if we have properly addressed your questions and what we can do to encourage you to increase our score.
> >
> > References:
> > [1]: Srebro, N., Sridharan, K., & Tewari, A. (2010). Smoothness, low noise and fast rates. Advances in neural information processing systems, 23.
> > [2]: Finn, C., Abbeel, P., & Levine, S. (2017, July). Model-agnostic meta-learning for fast adaptation of deep networks. In International conference on machine learning (pp. 1126-1135). PMLR.
> > [3]: Raghu, A., Raghu, M., Bengio, S., & Vinyals, O. (2019, September). Rapid Learning or Feature Reuse? Towards Understanding the Effectiveness of MAML. In International Conference on Learning Representations.
> > [4]: Collins, L., Mokhtari, A., Oh, S., & Shakkottai, S. (2022, June). Maml and anil provably learn representations. In International Conference on Machine Learning (pp. 4238-4310). PMLR.

---

> > > ### Author Response · Authors · 2023-11-21
> > >
> > > Dear reviewer Ss6u, since the rebuttal time is almost over, we are curious if our response addressed your questions and concerns on the lower bound and the practicality. Nevertheless, thank you once again for your feedback!

---

### Official Review · Reviewer_StV9 · 2023-11-10

**Soundness:** 2 fair
**Presentation:** 3 good
**Contribution:** 2 fair
**Rating:** 5
**Confidence:** 3

**Summary:**

The paper provides a tighter statistical guarantee in the sample-complexity of transferring what is learned from source tasks to target tasks and empirical evaluates this on some simple control tasks.

**Strengths:**

**Originality**
- The paper uses Rademacher complexity instead of Gaussian complexity, as used heavily by work that the paper references, in order to derive a tighter bound for a sample-complexity bound of the benefits of a representation in transfer learning in multi-task imitation learning (MTIL)

**Quality**
- Good to create and evaluate algorithms on discrete action space variants of continuous action environments while also evaluating on these continuous spaces to see whether theory that the paper proposes is actually empirically supported in both spaces for the same type of problem domain.
- The paper includes multiple correlation values in Tables 2 and 3 to cover certain limitations of individual ones.

**Clarity**
- The paper does a good job throughout explaining technical details and its experimental design.
- The paper provides intuitive explanations to accompany well-written rigorous definitions, which can help the reader better understand the concept being explained. For example, on page 3, the paper states "Intuitively, the Rademacher complexity of F measures the expressiveness of F over all datasets X through fitting random noise." after providing a rigorous definition of Rademacher complexity in its problem setup.

**Significance**
- The significance is potentially large, but I'm unsure how well it generalizes, especially due to the limitations of using only the KL-divergence and no other measure of dissimilarity between probability distributions.

**Weaknesses:**

1. The paper uses only KL-divergence to measure task-diversity for source and target tasks.

2. The paper doesn't compare using $D_{KL}$ to using other statistical measures of similarity between probability distributions, such as Bhattacharyya distance, which seem much more appropriate to do than the measures that the paper does compare $D_{KL}$ against.

3. **Generally when it comes to Rademacher Complexity in this context of this work, my concerns (really just an overarching single concern) are detailed in the paragraphs below.** However, I welcome thoughts on others from whether these are valid here or out of scope. If out of scope, then I also welcome discussion on how significant is the paper context, really?
- In the context of bounding sample-complexity in transfer learning, particularly for evaluating the richness of representation classes I wouldn't use Rademacher Complexity because of the importance of nuance in transfer learning on sequential tasks, which this measure avoids accounting for.
- Brief descriptions of Rademacher Complexity usefulness, main advantage, and main disadvantage for context:
  - Utility: Measures the ability of a function class to fit random noise, providing a general sense of the capacity of the function class.
  - Pro: Provides a general and well-understood measure of complexity that is applicable across various learning scenarios.
  - Con: It may not be as directly relevant to transfer learning scenarios because it doesn't specifically account for the nuances of transferring knowledge from a source to a target task.
- Given the specific requirements of transfer learning, which often involve understanding the relationship and distributional differences between source and target tasks, measures like Maximum Mean Discrepancy (MMD), Discrepancy Distance, and Task Similarity Measures become more pertinent. These measures are more directly aligned with the challenges of assessing how well a learned representation from one task can be applied to another, which is at the heart of transfer learning.
- Rademacher complexity, while powerful in many learning theory contexts, does not explicitly address these transfer-specific concerns. Therefore, it's more suited to general learning scenarios rather than the specific complexities of transfer learning.
- In fact, Gaussian complexity is somewhat similar but might be more suitable in certain contexts, especially in which approximately Gaussian assumptions naturally occur. Therefore, I again question the significance of this finding with Rademacher complexity in practice.

4. Evaluations do not include baselines using Gaussian Complexity in-place of Rademacher Complexity even though Gaussian Complexity may empirically be more useful on some tasks here.

**Questions:**

1. What insights do you have as to what causes the issue brought up in "We note that equation 5 is asymmetrical (i.e.  swapping τ with one of t ∈ [T] can yield different diversity estimate.)   This is a desirable property since model transfer generally is not symmetrical (Sugiyama et al., 2007). Suppose we have the expert policies π,π′ respectively for environments τ,τ′.  while π may stay performant in both τ,τ′, π′ may degrade when transferred to τ.  We demonstrate this in appendix D where the expert from each environment variation exhibits different robustness—one can stay performant in the target environment while another can degrade in performance."

2. On Page 2, the paper states "The consequence is that we can connect our result with deep-learning
theory, where the commonly used neural networks are quantified directly with Rademacher complexity (Bartlett et al., 2021)."
  a. Is this the best way to quantify neural network complexity here?
  b. Could you expound on your answer to 2a?
  c. What other options are available?

3. Thoughts on using maximum mean discrepancy instead of Rademacher complexity?
4. Thoughts on using discrepancy distances, as these are directly applicable in assessing transfer learning effectiveness.
5. Thoughts on using local Rademacher complexities? This would be more nuanced and data-dependent though, and the Rademacher complexity avoids this nuance.

**Details Of Ethics Concerns:**

I do not have ethics concerns with this work.

---

> ### Author Response · Authors · 2023-11-16
>
> We thank reviewer StV9 for their feedback. We appreciate that you find this work to be potentially large.
>
> **W1:** We thank the reviewer for their comment on the limitation of our task-diversity metric. We note that our task-diversity metric is based on the observations that (1) the source tasks likely capture the target task information if there is a source policy that performs well in the target task, and (2) the KL-divergence is asymmetrical which is a desirable property in task transfer. However, we agree that there can be other metrics, as you have mentioned in the later points. We aim to consider these metrics in future work.
>
> **W2:** Thank you for pointing this out, we will include a comparison against Bhattacharyya distance shortly.
>
> **W3:** Thank you for initiating this discussion. While we do believe this may be out of scope of our paper, we are happy to discuss this. The concern with discrepancy measure is that usually it relates only one source task to one target task. Perhaps what can happen in this case is to consider the source task distribution as the distribution of mixture policies in various MDPs.
>
> **W4:** Perhaps we misunderstood the comment, could you please rephrase it? The (empirical) evaluations did not involve any complexity measure, rather we focused on minimizing the loss through KL-divergence.
>
> **Q1:** Thank you for the question. This can be answered through the pendulum example. Suppose the pendulum can vary in its maximum torque. The lower-torque variant includes less policies that can keep the link upright while the higher-torque variant includes more policies. As a result, the lower-torque variant can be used in the higher-torque environment (arguably without finetuning) to keep the link upright, while this fails in the other case.
>
> **Q2:** Thank you for the interesting question. Generally, we do not believe that Rademacher complexity is the best way to quantify neural network complexity. In fact many analyses consider Gaussian complexities due to its mathematical properties, as demonstrated in prior work outlined in our paper. Another approach for analyzing neural networks can be using algorithmic stability along with metric entropy [3,4]. It is also common to consider norm-based complexity which is used to upper bound Rademacher complexity [1,5].
>
> **Q3-Q5:** They are all very great suggestions that we believe can be considered generally. We have not considered discrepancy measure but it might be interesting, as you have discussed in W3. Our goal is to directly quantify the representations learned from the source tasks and the target task. The difficulty may lie in including the true shared representation to the empirical representations from the source tasks and the target task, and finally connecting it to the imitation gap. We can alternatively consider the MMD of the policies, taking the expectation over their respective occupancy measures. Localized Rademacher complexity may give us $O(1/M)$ rate in some situations, as you have mentioned. But, it may still be difficult to improve upon $O(1/\sqrt{NT})$ generally, as we have discussed with reviewer Ss6u.
>
> Finally, thank you again for your insightful feedback. Please let us know if we have properly addressed your questions and what we can do to encourage you to increase our score.
>
> References:
> [1] Liang, T., Poggio, T., Rakhlin, A., & Stokes, J. (2019, April). Fisher-rao metric, geometry, and complexity of neural networks. In The 22nd international conference on artificial intelligence and statistics (pp. 888-896). PMLR.
> [2] Bartlett, P. L., Montanari, A., & Rakhlin, A. (2021). Deep learning: a statistical viewpoint. Acta numerica, 30, 87-201.
> [3] Bousquet, O., & Elisseeff, A. (2000). Algorithmic stability and generalization performance. Advances in Neural Information Processing Systems, 13.
> [4] Li, Y., Ildiz, M. E., Papailiopoulos, D., & Oymak, S. (2023). Transformers as algorithms: Generalization and stability in in-context learning.
> [5] Neyshabur, B., Tomioka, R., & Srebro, N. (2015, June). Norm-based capacity control in neural networks. In Conference on learning theory (pp. 1376-1401). PMLR.

---

> ### Author Response · Authors · 2023-11-16
>
> We thank reviewer StV9 for suggesting another metric. We have included a metric using the Bhattacharyya distance, replacing the KL-divergence. As a preliminary step we computed the metrics only for the discrete environments. We can see that Bhattacharyya distance may negatively correlate to the imitation performance in Frozen Lake and have weaker correlation in Walker compared to KL. We plan to investigate further in the future different task diversity metrics and how their theoretical properties may correlate to the imitation performance.
>
> EDIT:
> We have included this table (table 6) on page 32, with explanation on pages 30 and 31. we further included the L2 representation norm as suggested by reviewer kWs2
>
> ## Pearson
> |               | Frozen Lake                      | Pendulum                         | Cheetah                          | Walker                           |
> |---------------|----------------------------------|----------------------------------|----------------------------------|----------------------------------|
> | L2            | $ - 0.008  \pm  0.023 $          | $ 0.059  \pm  0.069 $            | $ - 0.008  \pm  0.080 $          | $ - 0.016  \pm  0.096 $          |
> | Data Perf.    | $ 0.035  \pm  0.223 $            | $ - 0.027  \pm  0.023 $          | $ - 0.057  \pm  0.041 $          | $ 0.279  \pm  0.128 $            |
> | Bhattacharyya | $ - 0.358  \pm  0.098 $          | $  0.071  \pm  0.013 $ | $ 0.150  \pm  0.029 $            | $ - 0.041  \pm  0.011 $          |
> | L2 Repr. Norm | $0.169 \pm 0.040$ | $\mathbf{0.190 \pm 0.034}$ | $0.381 \pm 0.060$ | $0.048 \pm 0.015$          |
> | Approx. KL (Ours)    | $ \mathbf{ 0.276  \pm  0.118 } $ | $ - 0.099  \pm  0.017 $          | $ \mathbf{ 0.386  \pm  0.062 } $ | $ \mathbf{ 0.383  \pm  0.068 } $ |
>
> ## Spearman
> |               | Frozen Lake                      | Pendulum                         | Cheetah                          | Walker                           |
> |---------------|----------------------------------|----------------------------------|----------------------------------|----------------------------------|
> | L2            | $ - 0.019  \pm  0.010 $          | $ - 0.064  \pm  0.032 $          | $ 0.091  \pm  0.024 $            | $ - 0.008  \pm  0.155 $          |
> | Data Perf.    | $ \mathbf{ 0.035  \pm  0.028 } $ | $ 0.116  \pm  0.026 $            | $ 0.053  \pm  0.071 $            | $ - 0.159  \pm  0.083 $          |
> | Bhattacharyya | $ - 0.138  \pm  0.046 $          | $ 0.079  \pm  0.026 $            | $ 0.249  \pm  0.037 $            | $ 0.045  \pm  0.014 $            |
> | L2 Repr. Norm | $0.002 \pm 0.010$ | $\mathbf{0.288 \pm 0.048}$ | $\mathbf{0.404 \pm 0.064}$ | $0.125 \pm 0.027$          |
> | Approx. KL (Ours)   | $ 0.005  \pm  0.022 $            | $  0.223  \pm  0.036 $ | $ 0.363  \pm  0.054 $ | $ \mathbf{ 0.242  \pm  0.039 } $ |
>
> ## Kendall
> |               | Frozen Lake                      | Pendulum                         | Cheetah                          | Walker                           |
> |---------------|----------------------------------|----------------------------------|----------------------------------|----------------------------------|
> | L2            | $ 0.111  \pm  0.022 $            | $ 0.052  \pm  0.105 $            | $ - 0.082  \pm  0.059 $          | $ 0.010  \pm  0.044 $            |
> | Data Perf.    | $ - 0.033  \pm  0.006 $          | $ 0.137  \pm  0.068 $ | $ 0.046  \pm  0.023 $            | $ - 0.355  \pm  0.053 $          |
> | Bhattacharyya | $ - 0.203  \pm  0.030 $          | $ 0.136  \pm  0.043 $            | $ 0.422  \pm  0.064 $            | $ 0.022  \pm  0.026 $            |
> | L2 Repr. Norm | $\mathbf{0.206 \pm 0.033}$ | $\mathbf{0.185 \pm 0.031}$ | $\mathbf{0.490 \pm 0.070}$ | $0.168 \pm 0.028$         |
> | Approx. KL (Ours)   | $ 0.121  \pm  0.031 $ | $ 0.012  \pm  0.025 $            | $ 0.479  \pm  0.069 $ | $ \mathbf{ 0.220  \pm  0.039 } $ |

---

> > ### Author Response · Authors · 2023-11-21
> >
> > Dear reviewer StV9, since the rebuttal time is almost over, we are curious if our response, in particular, a metric using Bhattacharyya, has addressed your questions and concerns. Nevertheless, thank you once again for your feedback!

---

> > > ### Comment · Reviewer_StV9 · 2023-11-22
> > > **Update**
> > >
> > > Hi Authors,
> > >
> > > Thank you for your response. I understand the importance of your effort in this work and the rebuttal, so I want to let you know that I will finish reviewing this + other Reviewers' reviews and your discussions with them thus far and reply by EoD today so that you have some time to address my response to your rebuttal before the deadline. I am mindful that we are nearing the deadline, so do not plan to ask you to do anything time-consuming, such as long experiment runs.
> > >
> > > Best,
> > > Reviewer StV9

---

> > > > ### Comment · Reviewer_StV9 · 2023-11-22
> > > > **Rebuttal Response Part 1**
> > > >
> > > > **W1:** It is stated in the paper somewhat often that the asymmetry of $D_{KL}$ is desirable/crucial/etc. alongside many references. For example in Appendix B, the paper states "We note that our estimate is asymmetrical, which is a critical property for transferring (Sugiyama et al., 2007; Mansour et al., 2009; Hanneke & Kpotufe, 2019)."
> > > >
> > > > Because of the asymmetry's importance, I think that it would make the paper much more readable and credible towards readers if at least one or two reasons for the asymmetry's criticality were included.
> > > >
> > > > The explanation in Page 5 is a possibility and not a certainty, and it is easy to come up with intentionally simple, or toy, examples in which the asymmetry leads to degradation of both expert policies or in which symmetry leads to improvement in both policies.
> > > >
> > > > **W3:** Isn't $D_{KL}$ limited in the same way as you reason: "The concern with discrepancy measure is that usually it relates only one source task to one target task.", as other discrepancy or dissimilarity measures, such as the Maximum Mean Discrepancy (MMD)?
> > > >
> > > > $D_{KL}(P || Q)$ measures the divergence of a single distribution $P$ from a a single distribution $Q$. In the paper's case $P$ is the "expert", which is equivalent to "optimal" according to the paper, policy for a single task and $Q$ is a learned target task composition policy.
> > > >
> > > > This seems to be why you sum each task-specific $D_{KL}$ over all of the tasks $t$ in your primary proposed measure **Approx. KL** (Eq. 6).
> > > >
> > > > Wouldn't you just do the same for other discrepancy measures, such as the MMD, Bhattacharyya Distance, Wasserstein Distance, and Deep CORAL [1], if you were to use them?
> > > >
> > > > **W4:** I believe I previously misunderstood this. It seems like Rademacher complexity is used to show the theoretical results but is not used in the algorithms proposed. Page 2 states "building upon our theoretical contribution, we propose a new metric that measures task diversity using the KL- divergence  of  the  expert  and  the  trained  policies" but I'm confused how the theoretical results lead to the new metric. Could you make this connection clear? Apologies if I missed it somewhere in the paper.
> > > >
> > > > ---
> > > >
> > > > [1] Sun, B., & Saenko, K. (2016). Deep coral: Correlation alignment for deep domain adaptation. In Computer Vision–ECCV 2016 Workshops: Amsterdam, The Netherlands, October 8-10 and 15-16, 2016, Proceedings, Part III 14 (pp. 443-450). Springer International Publishing.

---

> ### Author Response · Authors · 2023-11-22
>
> We thank reviewer StV9 for their feedback. We hope our response addresses your questions and concerns.
>
> **W1:** Thank you for your question. Our statement on environment asymmetry is a motivation for why we cannot simply transfer one policy from one environment to another. Indeed, there can be cases where you can simply transfer but this is not the case **generally**, another concrete example can be found under section 3.1, example 3 of Hanneke & Kpotufe, 2020. Consequently, we want to ensure our metric to be able to capture asymmetry when we swap order of the environments.
>
> We also thank you for pointing the paragraph out, we notice that the reference was incorrect and is now corrected to be Hanneke & Kpotufe, 2020 and have pointed the readers to the example for more intuition.
>
> **W3:** Thank you for raising this question. This depends on how MMD will be used. Suppose we compare the MMD of the representations, then it is not immediately clear because we will need to consider what is the distribution of the learned representation. On the other hand, when computing MMD of the source policies and target policy, then we agree that we can try using MMD. We note that, however, one consideration to make is what kernel to use when we compute the empirical estimation of MMD.
>
> **W4:** Thank you for indicating the possible confusion. The intuition comes from definition 4 on page 14, where $\sigma$-diversity is defined to be upper bounded by the ratio of the worst-case representation difference (definition 3) and the task-average representation difference (definition 2).
> Both representation differences in this case are actually KL-divergences due to the fact that we are doing behavioural cloning.
> Consequently, we approximate this ratio via our approximate KL metric. More details can be found in appendix B on page 15.
>
> We once again thank reviewer StV9 for their thoughtful discussions!

---

> > ### Author Response · Authors · 2023-11-22
> >
> > Dear reviewer StV9, we are mindful of your valuable time and truly appreciate your feedback. As there is very little time remaining in the discussion period, we hope that our previous response has addressed your concerns and questions. If not, we want to put our final efforts in addressing them. We sincerely thank you for engaging in the discussions, leading to the improvement of our work.

---

> > ### Comment · Reviewer_StV9 · 2023-11-23
> > **Rebuttal Response Part 2 (continued from Part 1 earlier in this thread)**
> >
> > **Q1:** Thank you for succinct explanation of the clear, concrete, and relatable pendulum example. I encourage you to include such an example in the main body of the paper, potentially with a visual, if you are able to find the space to do so and think it is important enough. I recognize you have some examples in the Appendix, but as you may agree, many readers tend to not visit the Appendix unless they find it vital to do so
> >  - Note: I recognize this is a presumption. I specifically stated "tend to" to not imply that all or even most readers don't visit the Appendix when they don't find it vital to do so. However there is friction in visiting the Appendix, so I'm certain that not all readers will visit it.
> >
> > Since transfer learning and asymmetry are central to the work, I think such an example will make the paper much more attractive, and more importantly, understandable and motivating to readers.
> >
> > I'm well versed in transfer learning, but I intentionally left my question open-ended so that I wouldn't induce bias into your response. I think a specific-concrete example is nice, as a general insights into this question will tend to be abstract, as they are in your paper at the moment. This isn't a bad thing. In fact, I think it's a good thing to have this generality, but I think including a specific-example in the main body would compound the strength of the correct general claim I quoted from your paper as part of my Question 1.
> >
> > This is just a suggestion in my attempt to make the paper as strong as possible, and consequently, if your paper is accepted, the conference as strong as possible. I will not reduce my rating if you decide not to include a concrete example in the main body.
> >
> > **Q2:** Thank you for sharing this information and references from which this information is supported!
> >
> > **Q3-Q5:** Thank you for providing insightful answers to my questions. There is indeed overlap between **W3** and **Q3-Q5.** In case it is unclear, I listed MMD simply as an example discrepancy measure and not to suggest or imply that you use MMD specifically. In fact MMD might be the option I'd try the least out of the bunch that I listed.
> >
> > My intent was to highlight that other discrepancy measures could, and arguably should, be used alongside $D_{KL}$ so that your algorithm is more robust to pitfalls of any one discrepancy measure. I understand a large benefit to using $D_{KL}$ is that minimizing $D_{KL}$ is equivalent to maximizing the log likelihood of the empirically seen (or "true") data distribution, which is what behavior cloning is trying to do, ASSUMING the function class has the capacity to represent the true data distribution. However, using $D_{KL}$ has often has serious downsides, or even pitfalls, by inducing a learned, generally continuous, policy that is spread over **every** seen data point in a "convex hull" type fashion (not exactly, usually ends up being Gaussian distribution, but the intuition is similar to capturing an approximate convex hull of the seen data points", which generally puts much support over areas that may not have it in the true data distribution.
> >
> > I'm simply making explicit a trade-off with using $D_{KL}$.
> >
> > I will continue with a brief Part 3 on your Bhattacharyya results and more. Please do share your thoughts in the meanwhile, and thank you for your effort!

---

> ### Author Response · Authors · 2023-11-23
>
> We thank reviewer StV9 for the continued discussions, we appreciate the amount of effort you have put in so far.
>
> **Q1:** This is a good point, we have included a pendulum example in the updated draft, please see page 5, figure 1. Generally, the result agrees with the intuition we have provided in the previous response, where the expert policy from the lower-torque variant is more robust to the higher-torque variants, but not the other way around. We hope that this addition would provide better intuition on the asymmetry of policy transfer.
>
> **Q3-Q5:** Thank you for pointing out the pitfalls of KL-divergence in the continuous setting. We agree that in practice the continuous setting is more difficult, and KL-divergence requires distribution realizability---often intractable to compute. Even in remark 3 (now moved to appendix A on page 15), our suggestion is to discretize the continuous space at the cost of the approximation error. We agree that this perhaps may be a limitation in practice, but we note that this limitation is very common in various areas of machine learning.
>
> We once again thank reviewer StV9 for their insightful discussions. We hope they can reconsider our score after our adjustments.

---

### Author Response · Authors · 2023-11-16
**General Response**

We would like to thank all the reviewers for their time to provide the thoughtful reviews, insights, and constructive feedback. We believe their ideas will further improve our work and clarity.
We are heartened that the reviewers found our work to have potential significance in the field (StV9, Eicc, F18A, Fs7F, Ss6u), technically sound (StV9, kWs2, F18A, Fs7F), and easy to follow (StV9, kWs2, Eicc, F18A).

Here, we have gathered a common feedback and provide a general response and updated our manuscript.
Additionally, we have replied to each reviewer individually.
1. Regarding the term "demonstrations": We thank reviewers Ss6u and kWs2 for pointing out the confusing usage of "demonstrations". We agree that a demonstration is often referred to a trajectory. In our paper, we define a demonstration to be a state-action tuple generated from the expert policy, as specified in section 2.2.
2. Regarding the term "policy error": We thank reviewers Ss6u and kWs2 for pointing out that "policy error" is not properly defined. While there are existing works [1], we agree with reviewer kWs2 that imitation gap/performance difference is more suitable. We have replaced "policy error" with "imitation gap" in our manuscript and clarified its meaning.
3. Difficulty in separating lines in Figures 1-4: We thank reviewers F18A and Eicc for indicating the difficulty in reading figures 1-4. We have updated the plot in the manuscript.
4. We aim to provide various analyses as suggested by all reviewers during this rebuttal period. Please stay tuned!

Finally, we thank all the reviewers again and welcome everyone to continue the insightful discussions!

References:
[1]: Pires, B. Á., & Szepesvári, C. (2016, June). Policy error bounds for model-based reinforcement learning with factored linear models. In Conference on Learning Theory (pp. 121-151). PMLR.

---

### Author Response · Authors · 2023-11-23

As the discussion period is coming to an end, we would like to thank all the reviewers for the productive and constructive discussions.

Best regards,
Authors of Submission 3065

---

### Meta-Review · Area_Chair_1mma · 2023-12-06

**Metareview:**

(a) Summary

The paper presents a theoretical and empirical study on the benefits of representation transfer in multitask imitation learning. The authors claim that leveraging representations trained on diverse source tasks can significantly improve sample efficiency when learning new tasks. In theory, they prove a high-probability bound on the performance difference between BC learned on top of the shared representation and the expert data.  Empirically, they show that on a variety of discrete and continuous control problems, pre-training on multi-task data allows for effective policy learning with limited target-task data.

(b) Strengths:
(+) Theoretical Analysis is Clear and Easy to Follow (Reviewer kWs2): The paper uses Rademacher complexity to derive novel sample complexity bounds for multitask imitation learning. The theoretical analysis was well-structured and easy to follow.

(+)  Thorough Empirical Evaluations (Reviewer Fs7Fm):  The authors provide a comprehensive set of evaluations in both discrete and continuous domains. They also provide additional empirical analysis on diversity measures and provide intuitive examples of the asymmetry of policy transfer.

(c) Weaknesses:
(-) Incremental Novelty (Reviewer StV9): Concerns were raised about the novelty of the theoretical insights, particularly regarding the Rademacher complexity. The equivalence up to a logarithmic factor between Rademacher complexity and Gaussian complexity is well-documented, potentially reducing the novelty of the theoretical contribution.

(-) Limited Practicality of Theoretical Results (Reviewer StV9): The practical usefulness of the theoretical results, especially the Rademacher complexity in the context of multitask imitation learning, is questioned. The complexity may be too general for practical applications, and its connection to transfer learning scenarios is not directly addressed.

(-) Empirical Evaluation on Toy Environments (Reviewer StV9): The empirical evaluation primarily involves toy environments. The generalizability of the findings to more complex, real-world scenarios might be limited.

(-) Writing and Clarity (Reviewer Fs7F): The paper's claims, particularly regarding the application of its findings to neural network architectures like convolutional networks, are suggested to be overstated or not sufficiently substantiated with empirical evidence.

(-) Comparison with Baselines (Reviewer kWs2): The comparison with the behavioral cloning (BC) baseline is seen as relatively straightforward, with a suggestion to show comparative results where BC has more data, to provide a fairer and more comprehensive comparison.

**Justification For Why Not Higher Score:**

* The theoretical contributions of the paper are limited, relying on a standard analysis from supervised learning.
* The empirical evaluation is also limited compared to existing multi-task imitation learning papers

**Justification For Why Not Lower Score:**

N/A

---

### Decision · Program_Chairs · 2024-01-16

Reject